



# Coccolithophore productivity at the western Iberian Margin during the middle Pleistocene (310 – 455 ka) – evidence from coccolith Sr/Ca data

Catarina Cavaleiro[1, 2, 3], Antje H. L. Voelker[2, 3], Heather Stoll[4*], Karl-Heinz Baumann[5] and Michal Kucera[1]

[1]University of Bremen, MARUM - Center for Marine and Environmental Sciences, Leobener Straße 8, 28359 Bremen, Germany.
[2]IPMA – Instituto Português do Mar e da Atmosfera, Divisão de Geologia Marinha e Georecursos Marinhos, Avenida Doutor Alfredo Magalhães Ramalho 6, 1495-165 Alges, Portugal.
[3]CCMAR, Centro de Ciências do Mar, Universidade do Algarve, Campus de Gambelas, 8005-139 Faro, Portugal.
[4]Geology Department, University of Oviedo, C/. Jesús Arias de Velasco s/n, 33005, Oviedo, Spain. *now at: Department of Earth Sciences, ETH Zürich, Sonneggstrasse 5, 8092 Zürich, Switzerland
[5]University of Bremen, Geosciences Department, Klagenfurter Straße 2-4, 28359 Bremen, Germany.

*Correspondence to*: Catarina Cavaleiro (cdcavaleiro@gmail.com)

**Abstract.** Coccolithophores contribute significantly to the marine primary productivity and play a unique role in ocean biogeochemistry by using carbon for photosynthesis (biological pump) and also for calcification (carbonate pump). Despite the importance of including coccolithophores in global climate models to allow better predictions of the climate system's responses to planetary change, highly uncertain coccolithophore paleoproductivity past reconstructions mostly relied on proxies dependent on accumulation and sedimentation rates, and preservation conditions. In this study we used an independent proxy, based on the coccolith fraction (CF) Sr/Ca ratio, to reconstruct coccolithophore productivity. We used the marine sediment core MD03-2699 from the western Iberian margin (IbM), spanning the glacial/interglacial cycles of Marine Isotope Stage (MIS) 12 to MIS 9. We found that IbM coccolithophore productivity was controlled by changes in the oceanographic conditions, such as in SST, the competition for nutrients with other phytoplankton groups and insolation. Long-term coccolithophore productivity was primarily affected by variations in the dominant water mass. Polar and subpolar surface waters during glacial substages were associated with decreased coccolithophore productivity, with strongest productivity minima being concomitant with Heinrich-type events (HtE). Subtropical, nutrient-poorer waters during interglacial substages, i.e. MIS 11c, might have lead to intensified competition for nutrients with diatoms resulting in intermediate levels of coccolithophore productivity. The transition from interglacial to glacial substages was likely associated with increasing presence of nutrient-richer waters, possibly with lower silica content than riverine discharges and mostly fed by either upwelling or surface waters of northern origin. This minimized the competition with diatoms and coccolithophores reached their productivity maxima. Climatic conditions during colder periods forced coccolithophores to change their phenology contributing to the dissonance between the CF Sr/Ca derived coccolithophore productivity and





nannofossil accumulation rate and total alkenone flux, which is interpreted as a consequence of the narrowing yearly time-window for coccolithophore productivity.

## 35 1 Introduction

Coccolithophores are major contributors to the marine primary production and play a unique role in ocean biogeochemistry because they use carbon for photosynthesis (biological pump) as well as for calcification (carbonate pump) (e.g. Rost and Riebesell, 2004; Westbroek et al., 1993). They are the most important unicellular primary producer producing calcite (Brand, 1994), contributing up to 60 % to the total oceanic calcium carbonate (Flores and Sierro, 2007) with peak contribution of

>80 %  in the interval of Marine Isotope Stage (MIS) 15 to MIS 9, when the assemblages were by far dominated by gephyrocapsids (Baumann and Freitag, 2004; Saavedra-Pellitero et al., 2017) are sensitive to rapid fluctuations in temperature, salinity, nutrients, and turbidity of surface waters (Baumann et al., 2005; McIntyre and Bé, 1967). Hence, coccoliths, the minute and remarkably abundant calcite plates produced by coccolithophores, have been collected from marine sediments and extensively used as tracers in paleoceanography to reconstruct the hydrological characteristics of the

photic zone as well as their productivity (Amore et al., 2012; Baumann et al., 2005; Beaufort et al., 2001; Flores et al., 1997; Maiorano et al., 2015; Marino et al., 2014; McIntyre and Molfino, 1996; Saavedra-Pellitero et al., 2017). However, coccolithophores' role in the climate system, via the carbon cycle, is still under debate (e.g. McClelland et al., 2016; Omta et al., 2013; Saavedra-Pellitero et al., 2017), although it would be important to accurately include them in global climate models (Ridgwell and Zeebe, 2005). The latter would allow better predictions of the climate system's responses to planetary

change, such as atmospheric carbon dioxide and coupled temperature rise as well as ocean acidification (e.g. Kohfeld and Ridgwell, 2013). Still, coccolithophore paleoproductivity reconstruction has been tentative and mostly relied on proxies dependent not only on the extent of the supply but also on dilution by mineral matter, changes in sedimentation or accumulation rates, as well as preservation conditions (Rullkötter, 2006). Beaufort et al. (1997) proposed a proxy to quantitatively reconstruct coccolithophore productivity, but its applicability is unfortunately limited to latitudes between

30°N and 30°S (Hernández-Almeida et al., 2019).

A widely used alternative proxy is the coccolith fraction Sr/Ca (CF Sr/Ca) ratio that is independent of accumulation rate (e.g. Cavaleiro et al., 2018; Mejía et al., 2014; Saavedra-Pellitero et al., 2017). Coccolithophores construct coccoliths internally within their cell and several studies show the direct and proportional relationship between the Sr/Ca ratio of the coccolith and the coccolith calcification rate. Calcification rate is a function of growth rate and therefore of coccolithophore

productivity (e.g. Rickaby et al., 2007; Stoll and Schrag, 2000). The faster coccolithophores grow, the faster they calcify and more Sr is incorporated into the calcite lattice of their coccoliths (Stoll et al., 2002b, 2002a; Stoll and Schrag, 2000). Culture studies also found a temperature dependence of coccolith Sr/Ca with a 0.03 mmol/mol increase per degree Celsius rise (Müller et al., 2014; Stoll et al., 2002a). This signal can, however, be removed from the Sr/Ca record using an independent sea-surface temperature (SST) reconstruction, so that the component of variation due to growth rate remains as residual (e.g.



Cavaleiro et al., 2018; Mejía et al., 2014; (e.g. Cavaleiro et al., 2018; Mejía et al., 2014; Saavedra-Pellitero et al., 2017). Also, despite the variations among species in the amount of Sr introduced into its calcite (with larger and more heavily calcified coccoliths generally having a higher Sr content than smaller and lighter coccoliths (Fink et al., 2010; Stoll et al., 2007)), it has been demonstrated that CF Sr/Ca ratios are not primarily controlled by variations in coccolith assemblage in the modern ocean (Barker et al., 2006; Stoll and Schrag, 2000). It is thus reasonable to conclude that most CF Sr/Ca ratio

records reflect long-term changes in coccolithophore production and calcification. With temperature and assemblage effects considered, the SST corrected CF Sr/Ca curve (or residual curve) can be expected to mostly reflect coccolithophores' growth rate and thus their productivity qualitatively (Müller et al., 2014; Stoll et al., 2002a).

      This study reconstructs coccolithophore productivity from the CF Sr/Ca ratio record at Site MD03-2699, retrieved from the western Iberian margin (IbM) (Fig. 1), in an eastern boundary upwelling system. We aim to characterise long-term changes

in coccolithophore productivity in such a system, where their behaviour in the past remains unknown. We explore different climatic scenarios during the Mid Brunhes interval (Barker et al., 2006; Baumann and Freitag, 2004; Jansen et al., 1986), focusing on the interval from Marine Isotope Stage (MIS) 12 to MIS 9, and evaluate the main factors influencing coccolithophore productivity. This interval was a critical time of important global climate change when, after the mid-Pleistocene transition, glacial-interglacial cyclicity became more stable at the periodicity of ca. 100 kyr (e.g. Berger and

Wefer, 2003). Our study period includes harsh glacial periods such as MIS 12 and the prolonged MIS 11c interglacial period. The MIS 11c interglacial is considered one of the best analogues for the current MIS 1 interglacial (Berger et al., 2015; Candy et al., 2014; Loutre and Berger, 2003; Oliveira et al., 2018) and termination (T) V is characterized by the longest and highest glacial-interglacial amplitude change of the last one million years (Lisiecki and Raymo, 2005). The IbM is particularly sensitive to global climate change (Hodell et al., 2015; Martrat et al., 2007; Oliveira et al., 2016; Rodrigues et

al., 2017) and an optimal area to evaluate this phytoplankton group's behaviour and gain a better understanding of its response to climate conditions during glacials, interglacials, deglaciations and the transition from interglacial to glacial conditions, at both orbital and sub-orbital time scales. We further compare our proxy with classical coccolithophore productivity proxies, such as nannofossil accumulation rate (NAR) and total alkenone fluxes.

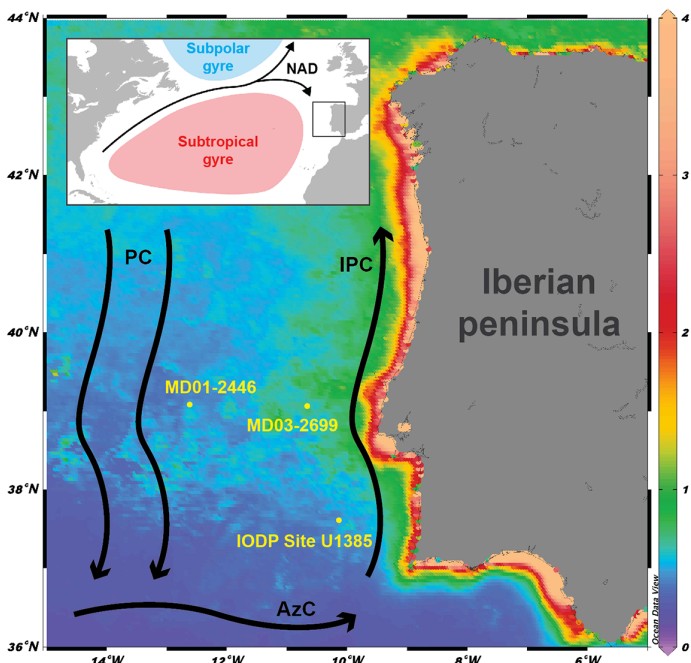

**Figure 1** Core location and major currents in the area: NAD = North Atlantic Drift; PC = Portugal Current; IPC = Iberian Poleward Current; AzC = Azores Current. Background: chlorophyll *a* concentration (mg.m$^{-3}$; March, April and May average, 2003-2018) derived from MODISA satellite data available at http://disc.sci.gsfc.nasa.gov/giovanni.

## 2 Regional setting

### 2.1 Present hydrography of the IbM

The study area is located at the north-eastern edge of the subtropical gyre and influenced by a southward flowing branch of the North Atlantic Current forming the Portugal Current system (Fig. 1). The Portugal Current system is mainly characterized by a slow, southerly flow poorly defined due to the intricate interaction between coastal and offshore currents (Peliz et al., 2005; Relvas et al., 2007), bottom topography and water mass convergence (Bischof et al., 2003). One of the most important features of the IbM is the surface circulation's seasonality as a response to the combined position of the

Azores high and the Icelandic low-pressure systems (Barton, 2001; Haynes and Barton, 2018; Relvas et al., 2007). During summer, the migration of the Azores high to the central Atlantic exposes the IbM to northerly Trade Winds, strengthening the Portugal Current and forcing the upper layer (150 to 200 m) to flow towards the equator. The induced offshore Ekman





transport allows colder, less salty and nutrient-rich subsurface water to rise to the surface. This upwelling (Alvarez et al., 2011; Fiúza, 1983) leads to high primary production (Figueiras et al., 2002; Fraga, 1981; Tenore et al., 2018). The upwelled

water is the Eastern North Atlantic Central Water (ENACW) that has two different origins: a less saline, colder, nutrient richer water mass of subpolar origin (ENACWsp) formed along with the Subpolar Mode Water in the Rockall Plateau region, and a warmer, saltier, nutrient poorer subtropical branch (ENACWst) formed along the Azores Front (Fiúza et al., 1984). Depending on the wind strength intensity either type can be upwelled (Fiúza et al., 1984). The upwelling season typically lasts from May to September and upwelling filaments are mostly observed off the most prominent capes (Fiúza et

al., 1982). Upwelling plumes and other mesoscale features can spread zonally offshore to distances of 200 km or more (Sousa and Bricaud, 1992). During winter, the Azores high is located further south and the Icelandic low intensifies, forcing the wind regime to become more southerly and enabling the near shore Iberian Poleward Current, which brings warmer and more oligotrophic waters from the Azores Current to the shelf and upper slope areas along the western IbM (Fiúza, 1983; Peliz et al., 2005; Vitorino et al., 2002). Downwelling can occur during this period but winter-time fresh-water discharges

from the major rivers are also a relevant feature, namely from the Tagus river that crosses the Iberian Peninsula, carrying a major load of suspended organic-rich sediments onto the continental shelf off Lisbon (Cabeçadas and Brogueira, 1998)

## 2.2 Present coccolithophore productivity patterns in the IbM

Several studies on coccolithophore abundance and distribution (e.g, Abrantes and Moita, 1999; Cachão et al., 2000; Guerreiro et al., 2013; Moita, 2001) have been performed along the IbM but only two were able to assess annual and

seasonal productive cycles (Ausín et al., 2018; Silva, 2008; Silva et al., 2009). Maximum coccolithophores were found during spring and summer, and associated with high irradiance levels (Ausín et al., 2017) and the convergence of warmer, more oligotrophic waters with relaxation of northerly winds (Silva, 2008; Silva et al., 2009). Minimum numbers, on the other hand, were observed during winter.

The competitive dynamics between coccolithophores and diatoms are still not completely understood (Cermeño et al.,

2011). Yet, these two main phytoplankton groups dominate the carbon dioxide mediation between the atmosphere and the ocean making them important contributors to the global carbon cycle. Diatoms are known to require more silica and iron than coccolithophores (e.g., Merico et al., 2004) and Tyrrell and Young (2009) suggested that coccolithophore blooms might be associated with silica and iron surface water depletion. Cermeño et al. (2011) suggested that coccolithophores could outcompete diatoms under steady-state nitrate limitation but under dynamical conditions (introducing nitrate pulses) diatoms

outcompeted coccolithophores. They further found that the more frequent the pulses, the more rapidly diatoms outcompeted coccolithophores. The coastal upwelling conditions resemble the dynamical conditions, when pulses of nutrients are brought to the ocean surface. This could explain why Abrantes and Moita (1999) found a phytoplankton dominance of diatoms during upwelling events and a clear dominance of coccolithophores (~ 90 %) during non-upwelling events. Also, Ausín et al. (2017) describe decreasing abundance of diatoms with increasing distance to shore whereas coccolithophores showed higher

abundances offshore than in coastal areas. This distribution pattern was maintained during the upwelling season (May to





September), when coccolithophores are outcompeted by diatoms, in particular during the longer lasting and more intense upwelling events (Moita, 2001; Silva et. al., 2008).

Despite the general acceptance of coccolithophores as a single functional phytoplankton group associated with low turbulence, low nutrient and high irradiance environments, different species show varying life strategies. Several studies

describe increased cell densities, mostly due to blooms of Emiliania huxleyi and Gephyrocapsa oceanica in the IbM (Moita, 2001; Guerreiro et al., 2013; Silva, 2008; Silva et al., 2009), together with other fast-growing opportunistic phytoplankton genera, such as the diatoms *Chaetoceros* s.l., *Thalassiosira* s.l and *Skeletonema* s.l.. This corroborates coccolithophores' role among early succession taxa, or at least some species, capable of rapid growth in a nutrient-rich environment and most likely explains why Abrantes and Moita (1999) found coccolithophores' distribution in recent sediments to reflect their water

column distribution during an upwelling situation.

## 3 Material and methods

### 3.1 Sediment sampling and coccolith fraction separation

In this study we used sediments of core MD03-2699 (39º02.20'N, 10º39.63'W, 1895 m water depth), retrieved from a sediment drift, located ca. 100 km offshore, on the Estremadura promontory (Fig. 1). Sediments were collected using a giant

CALYPSO piston corer on board the R/V Marion Dufresne II (PICABIA Cruise, 2003). The sedimentary record is mainly composed of hemipelagic silty clays. For our research, we use the age model published by Voelker et al. (2010).

For the CF Sr/Ca ratio, a total of 183 samples were analysed. Samples were taken at 4 cm spacing from the ~7 meters long section (from 1190 to 1898 cm core depth) corresponding to MIS 12 to MIS 9 (~ 455 to 310 ka). This resulted in a temporal resolution of ~775 years, although the oldest part has lower resolution due to the lower sedimentation rates (Rodrigues et al.,

2011). To obtain the CF Sr/Ca record, ~250 mg of freeze-dried sample was collected and suspended in 2% ammonia (to avoid carbonate dissolution) and sieved through a 20 μm mesh (to separate the coccolith fraction (<20 μm) from mostly foraminifera and their fragments, and other larger microfossils or sediment components). All sieving material was carefully washed with running tap water and rinsed with distilled water in between samples to avoid cross contamination.

### 3.2 Sample preparation and Sr/Ca analysis

We followed a three step protocol to clean the sediment samples based on Stoll and Ziveri (2002): (1) addition of 15 ml of MNX reagent (75 mg of hydroxylamine hydrochloride, 6 ml of concentrated ammonia and 9 ml of ultrapure water) for a 12 hours reaction in an automatic shaker. This step reduces Fe and Mn oxyhydroxides that scavenge metals from seawater and contain non-carbonate Sr, (2) addition of 2% ammonia to remove any non-carbonate Sr, e.g., from clays, by exchanging cations ($Sr_2^+$) with the excess of $NH_4^+$, (3) three ultrapure water rinses to extract the ammonia. A weak buffered acid (6 g

glacial acetic acid, 7 g ammonium acetate in 1 l of Milli-Q water) was then used to dissolve the coccoliths, minimizing the contribution of ions from non-carbonates phases. The samples were left in acid during 12 hours and the obtained solution



was centrifuged, extracted and kept in acid-cleaned centrifuge tubes. A first ICP-AES measurement of Ca was performed by diluting 100 μl of the original sample (2 ml) into 2 ml of ultrapure Millipore water. The samples were subsequently diluted to Ca concentrations similar to the standard solutions. Calibration was conducted following the method described by de

Villiers et al. (2002) using standards with constant Ca concentrations and different Sr concentrations to provide Sr/Ca ratios ranging from 0.75 to 4 mmol/mol. All measurements were conducted using the ICP-AES (Thermo ICAP DUO 6300) in the Geology Department at the University of Oviedo with reproducibility better than 0.02 mmol/mol. To infer any possible contamination other metals, such as Fe and Mg, were also measured together with Sr.

**3.3 Extraction of paleoproductivity record from the CF Sr/Ca ratios**

As previously mentioned, to correctly interpret the CF Sr/Ca ratio proxy it is important to account for changes in coccolith assemblage and SST through time.

Our study is placed within the *Gephyrocapsa caribbeanica* acme (Baumann and Freitag, 2004; Bollmann et al., 1998) between MIS 14/13 and MIS 8 when this species dominated ($\geq$ 50 %) the coccolithophore community globally. This species' governance minimizes the bias of the CF Sr/Ca data due to significant coccolith assemblage changes. Moreover, coccoliths

belonging to the genus *Gephyrocapsa* dominated the assemblage in core MD03-2699 during the entire studied period with an average of 97% (Fig. 2; Amore et al., 2012). The sum of the remaining coccoliths, containing the larger and Sr-rich coccoliths *Calcidiscus leptoporus*, *Coccolithus pelagicus* and *Helicosphaera carteri* averaged 2.8 % (maximum 8.7%, standard deviation 1.6%). Their contribution to the total Sr of the coccolith fraction is therefore negligible when compared to the Gephyrocapsids' contribution.





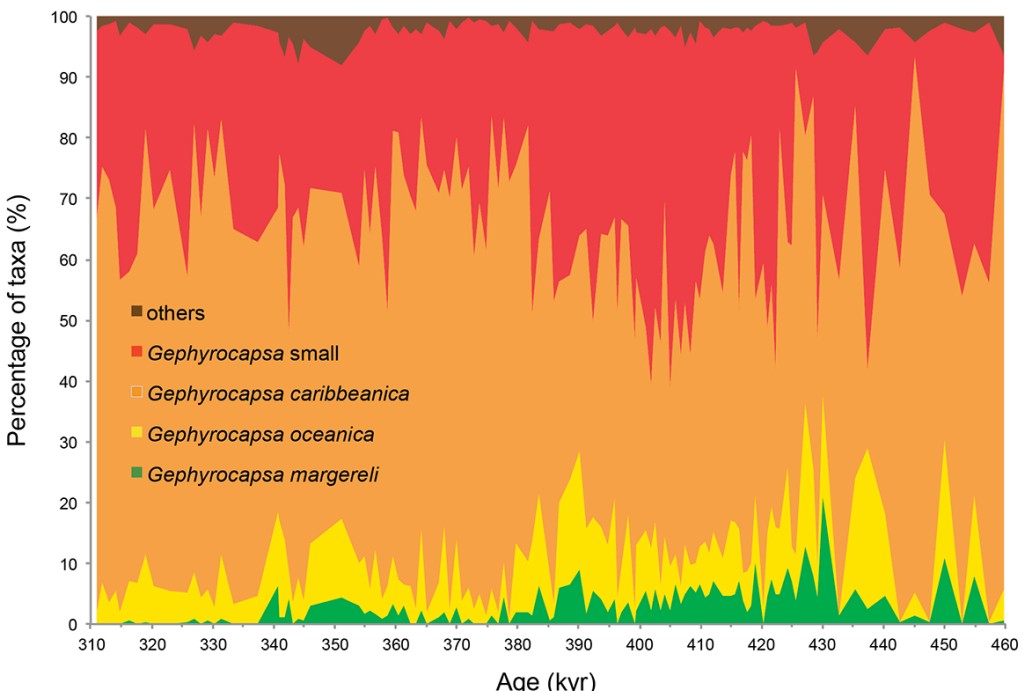


**Figure 2** Composition of coccolith assemblages in core MD03-2699 based on Amore et al. (2012). Note that only a minor percentage of coccoliths belong to groups other than the dominant Gephyrocapsids. The chronology is based on the age model of Voelker et al. (2010).

The temperature effect was extracted from the CF Sr/Ca ratios record following Mejia et al. (2014) and Cavaleiro et al. (2018). We used the multi-species temperature dependence to calculate a SST-predicted Sr/Ca curve and the extraction of the temperature effect consisted in subtracting the SST-predicted Sr/Ca values from the initial CF Sr/Ca ratios, Eq. 1:

$$CF\frac{Sr}{Ca} - SSTpredicted\frac{Sr}{Ca} = CF\frac{Sr}{Ca}residual$$                                                                        ,

(1)

We calculated the temperature dependence using the $U^{K'}_{37}$-based SST record of core MD03-2699 for the same period (Rodrigues et al., 2011). The alkenones present in the sediments are today mostly produced by a few species of the class Prymnesiophyceae, mainly by the coccolithophores *Emiliania huxleyi* and *Gephyrocapsa oceanica* (Prahl et al., 1988; Prahl and Wakeham, 1987), and the $U^{K'}_{37}$-based SST is therefore the best estimator of coccolithophores' habitat temperature. After extraction of the temperature influence, the resultant residual mainly represents coccolithophore calcification and growth rate

and consequently coccolithophore long-term productivity fluctuations. We refer to the Sr/Ca residual as coccolithophore productivity because after correcting for the temperature changes, we expect the data to mostly reflect coccolithophore growth rate and thus their paleoproductivity qualitatively. This coccolithophore productivity record reflects relative productivity change, representing the productivity deviation around the average productivity of the time series.

An uncertainty envelope/confidence interval for the CF Sr/Ca residual estimation (shown as a grey envelope in Fig. 3) was 205 calculated using Astrochron (Meyers, 2014) where the upper and lower limits correspond to the Monte Carlo 20 and 80 % confidence interval, respectively. One hundred Monte Carlo simulations were run for each of the 183 data points carrying out propagation of errors accounting for uncertainties measurement of: (1) temperature (with $\sigma = 1.5°C$); (2) Sr concentration (with $\sigma = 0.02$ mmol/mol); and (3) the linear regression of temperature versus Sr ($\sigma = 0.12$ mmol/mol).

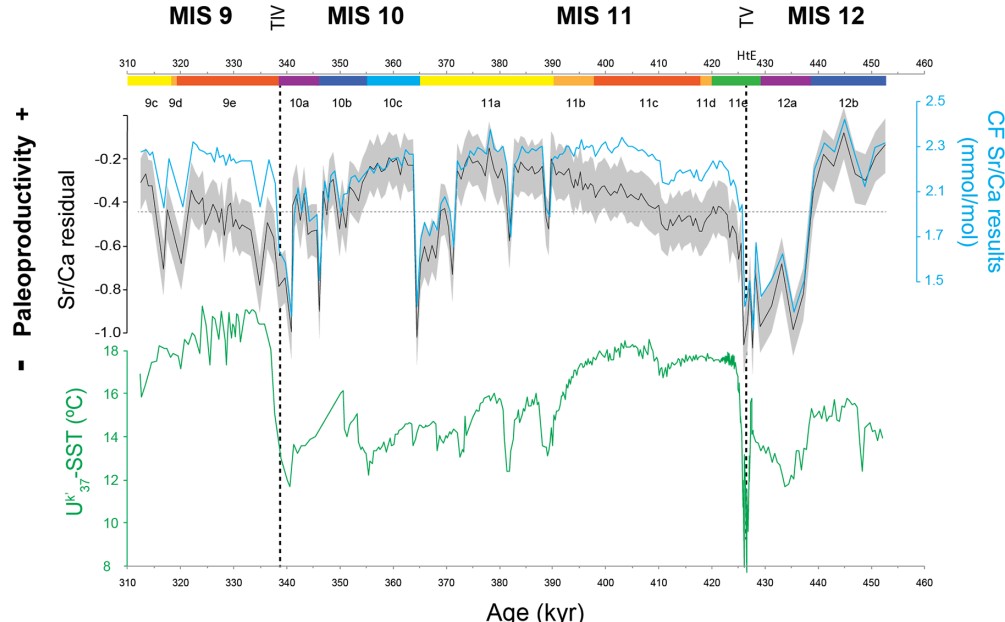


**Figure 3** Coccolithophore paleoproductivity results from core MD03-2699: coccolith fraction (CF) Sr/Ca ratio and the resulting CF Sr/Ca residual with confidence interval (grey shading represents Monte Carlo 20-80% interval), $U^{k'}_{37}$-based reconstruction of sea surface temperature (Rodrigues et al., 2011) The horizontal dashed line marks the average value for CF Sr/Ca and CF Sr/Ca residual. The chronology is based on the age model of Voelker et al. (2010), Marine isotope stages 215 (MIS) and substages marked according to Railsback et al. (2015), vertical dashed lines highlight Terminations V and IV.





### 3.4 Time-series analysis

Using the outcome of the harmonic analysis of the CF Sr/Ca residual record in the SPECTRUM program (Schulz and Stattegger, 1997), we extracted frequencies lower than 37 kyr from the coccolithophore productivity record (i.e. residuals of

eccentricity and obliquity whose occurrence in the time series is not frequent enough for reliable results). The filtered record was then used for the spectral and cross-spectral analysis in the programs REDFIT and SPECTRUM, respectively (Schulz and Mudelsee, 2002, Schulz and Stattegger, 1997), to establish statistically significant leads/lags between the coccolithophore productivity record and seasonal insolation.

### 4 Results

### 4.1 Potential influence of detrital carbonate on the CF Sr/Ca data

If present, detrital carbonate could negatively bias our CF Sr/Ca ratio (and our coccolithophore productivity record) by increasing the background carbonate in relation to the coccolith carbonate. In our study, we considered potential sources of detrital carbonate biasing our CF Sr/Ca results: (1) ice-rafted debris (IRD) discharged by melting icebergs; (2) aeolian input and (3) riverine discharge. Deposition of carbonate/dolomite rich IRD is linked to extremely cold and lower salinity

conditions (Fig. 4), namely Heinrich-type events (Andrews and Voelker, 2018; Hodell et al., 2008; Rodrigues et al., 2011; Salgueiro et al., 2010; Stein et al., 2009). Ice-rafted material, especially the fine-grained material, also contains reworked coccoliths, which can be Tertiary or Cretaceous in age (Marino et al., 2011, 2014; Rahman, 1995). Increased aridity and wind conditions enhance aeolian transport (e.g., Bozzano et al., 2002), which can transport detrital carbonate material, coming mostly from North Africa (e.g., Negral et al., 2012; Stumpf et al., 2010) to the Iberian margin. On the other hand,

increased moisture and precipitation intensifies rock weathering and riverine discharge, raising fluvial transport of detrital carbonate onto the Iberian margin (Hodell et al., 2017; Hodell and Channell, 2016). In the Iberian margin, increased detrital carbonate presence is prevalent during rapid millennial-scale related low sea-level stands and IRD events, as documented by Lebreiro et al. (2009), Marino et al. (2011), Hodell et al. (2013, 2017) and Hodell and Channell (2016).

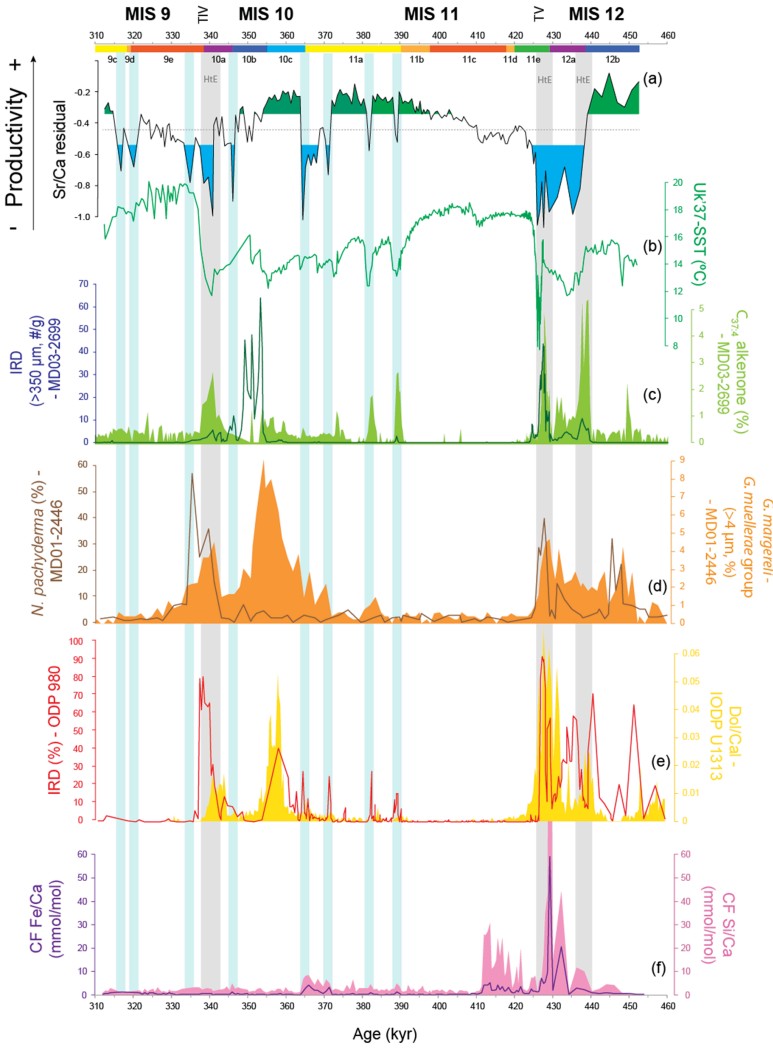

**Figure 4** Coccolithophore paleoproductivity reconstruction with "high" and "low" coccolithophore productivity levels highlighted in green and blue shading, respectively (a) compared to: b) $U^{k'}_{37}$-based reconstruction of sea surface temperature from core MD03-2699 (Rodrigues et al., 2011); c) ice-rafted debris (IRD) abundance (Voelker et al., 2010) and heptatriatetraenone ($C_{37:4}$ alkenone) abundance (Rodrigues et al., 2011), both from core MD03-2699; d) *Neogloboquadrina pachyderma* relative abundance and *Gephyrocapsa margereli* and *G. muellerae* group relative abundance, both from core





MD01-2446 (Marino et al., 2014); e) percentage ice-rafted debris (IRD) from ODP Site 980 (McManus et al., 1999) and Dolomite/Calcite ratio from IODP Site U1313 (Stein et al., 2009); and f) coccolith fraction (CF) Fe/Ca and Si/Ca ratios. Note that the MD03-2699 lithic fragments were counted in a coarser size fraction (>315 μm) than the standard size fraction (>150 μm; Hemming, 2004) and thus most likely only record major ice-rafting events at the IbM, as suggested by Marino et al. (2014). Chronology as in Fig. 3. Vertical bars: grey bars correspond to Heinrich-type events and blue bars to short-lived

events of decreased coccolithophore productivity.

In order to evaluate this potential biasing effect we defined a CF Sr/Ca ratio threshold at 1.8 mmol/mol based on the CF Sr/Ca and CF Mg/Ca cross-plot. (Fig. 5a) We used CF Mg/Ca because Mg/Ca peaks can be attributed to the presence of increased detrital carbonate, such as dolomite $(CaMg(CO_3)_2)$. The threshold was defined at 1.8 mmol/mol of Sr/Ca because

values at or below that level coincided with high Mg/Ca results (Fig. 5a). Also, consistently low CF Sr/Ca combined with high CF Mg/Ca started at CF Mg/Ca levels higher than 50 mmol/mol (Fig. 5a). And, samples with CF Sr/Ca results lower than 1.8 mmol/mol and CF Mg/Ca above 50 mmol/mol (Fig. 5b) were considered likely contaminated by detrital carbonate. We estimate that (i) 10% of the samples might have been biased by the presence of detrital carbonate and confirm that (ii) all of those samples are linked to rapid millennial-scale and low sea-level stands, namely during substages MIS 10a, MIS 12a

and the respective subsequent termination (Fig. 4). Although a possible bias effect exists in those samples, it does not prove that coccolithophore productivity did not decrease significantly during those intervals (see Discussion). Bias due to reworked coccoliths contribution is negligible because reworked coccolith abundance was generally below 2% not only in our study site (Amore et al., 2012) but also in other IbM sites for the same time interval (Maiorano et al., 2015; Marino et al., 2014).

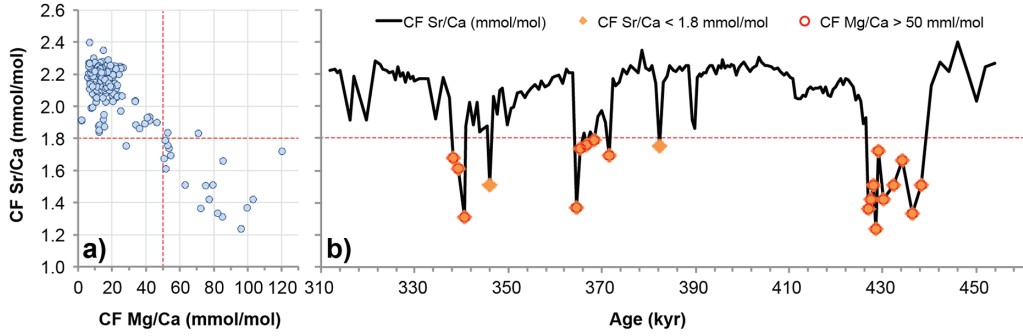


**Figure 5** Detrital carbonate analysis: a) coccolith fraction (CF) Sr/Ca results and CF Mg/Ca cross-plot. Vertical and horizontal red lines delimit thresholds at 1.8 mol/mol of Sr/Ca and 50 mmol/mol of Mg/Ca, respectively; b) CF Sr/Ca samples possibly subjected to detrital carbonate contamination (red makers), notably associated with cold and abrupt events.





### 4.2 Coccolithophore productivity results


The CF Sr/Ca ratios varied between 1.2 and 2.4 mmol/mol (Fig. 3), with 85 % of the data falling within the range of 1.8 to 2.3 mmol/mol. This interval represents 39 % of the whole sampling variation, with a total sampling range of 1.16 mmol/mol. The Sr/Ca residual, from now on referred to as coccolithophore productivity, generally mimics the CF Sr/Ca ratio record, and shows minima and maxima at the same levels (Fig. 3). The high CF Sr/Ca ratios found in the record do not coincide with


increased abundance of large/Sr-rich coccoliths (Fig. 2) or with higher temperatures (Fig. 3). This supports the statement that our CF Sr/Ca variation reliably records coccolithophore calcification rate and productivity changes. Both MIS 12 and MIS 10 are characterized by higher coccolithophore productivity during the first half and lower productivity during the second half. Large and rapid increases in coccolithophore productivity characterize the deglaciations at the end of MIS 12 and MIS 10.


To better visualize and interpret the CF Sr/Ca results, the coccolithophore productivity data were divided into three relative intervals. An "intermediate" interval is defined by the mean of the data plus and minus half of the standard deviation. The "high" and "low" coccolithophore productivity levels are above and below the intermediate level of coccolithophore productivity, respectively (highlighted by green and blue shading in figures, respectively). We would like to stress that our study focuses on the qualitative characteristics of the coccolithophore paleoproductivity record, rather than quantitatively


estimating the productivity of coccolithophores.

A preliminary visual inspection of the record already allows the identification of higher amplitude events during glacial substages, MIS 12a and 12b, and MIS 10a and 10b, in comparison to the less variable interglacial substages, MIS 11c and MIS 9e. The spectral analysis of the filtered coccolithophore productivity record revealed precession (~21 and 17 ka) and higher (~6 ka) quasi-periodicities (Fig. 6).


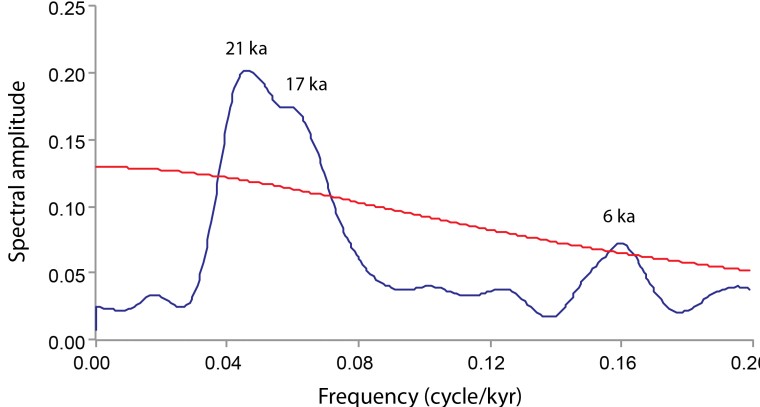



**Figure 6** Coccolithophore paleoproductivity time-series analysis showing significant peaks at 21, 17 and 6 ka. The red line defines the significance level of 80% (bandwidth is 0.02), indicating that the peaks at 21 and 17 ka are not independent (may represent the same frequency given the uncertainty of dating and resolution).

**5 Discussion**

Coccolithophore productivity maxima happened during the first half of MIS 12 and MIS 10 and during the transition from interglacial MIS 11c to glacial MIS 10 (Fig. 3). Productivity minima, on the other hand, occurred during glacial maximum substages MIS 12a and MIS 10a and at the end of MIS 11a. Marine Isotope Stages 11c and 9e, commonly known as the interglacial substages (Past Interglacials Working Group of PAGES, 2016), mainly show periods of lower amplitude changes

within intermediate level of coccolithophore productivity when compared to the rest of the record. Rodrigues et al. (2011) mentioned a similar SST pattern, on the same time frame and at the same site, which could imply SST as an important factor affecting coccolithophore productivity.

We therefore firstly evaluated the relationship between coccolithophore productivity and SST changes. The relationship between SST and coccolithophore productivity is characterised by a weak negative correlation ($r=-0.4$, $p<0.01$). However,

the cross-plot seems to show an inflexion around 15.5 ºC (Fig. 7) and, by separating the different sets of samples under the different temperature ranges (below and above 15.5 ºC), we obtained very different correlation results. Increasing temperatures until 15.5 ºC seem to have a positive effect on coccolithophore productivity (though with a low positive correlation; $r=0.27$, $p<0.05$), whereas increasing temperatures above 15.5 ºC seem to have a negative effect on coccolithophore productivity (with a medium negative correlation; $r=-0.54$, $p<0.01$). Though it is accepted that increased

temperature generally enhances coccolithophores growth (e.g. Sett et al., 2014), our results demonstrate that the relationship between coccolithophores productivity and SST is not straightforward and should be explored further in the future and that other factors, besides temperature, must have affected coccolithophore productivity in the western IbM. In an upwelling region like the IbM a potential factor could be nutrient concentrations because the upwelled waters in general have temperatures between 13 and 16 °C (e.g., Fiúza et al., 1998).




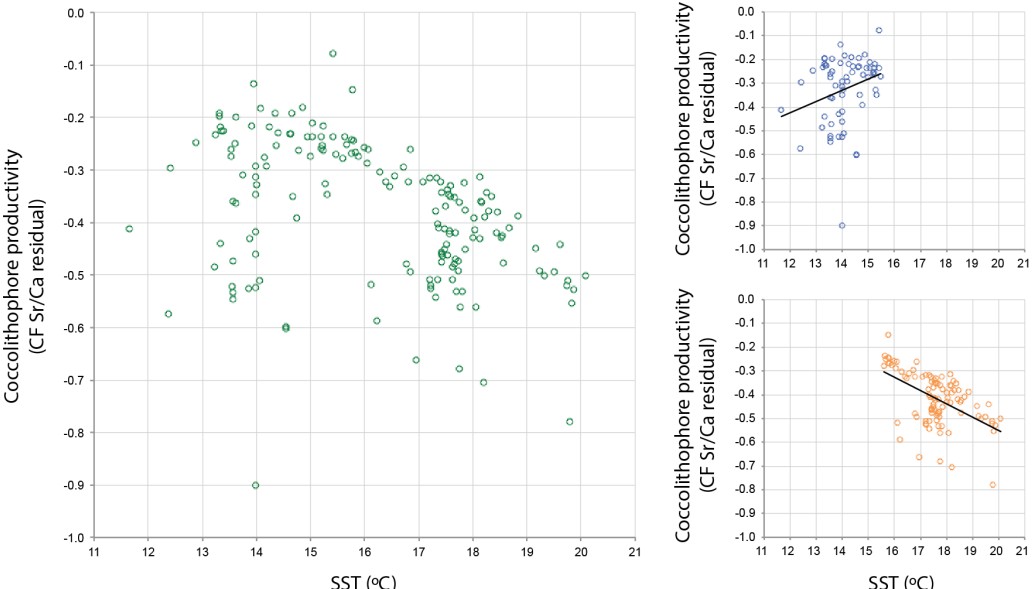

| Correlation Coefficients - Coccolithophore productivity (CF Sr/Ca residual) vs SST (ºC) | | | |
|---|---|---|---|
| | Sr/Ca residual vs. SST (ºC) | Sr/Ca residual vs. SST (ºC) | Sr/Ca residual vs. SST (ºC) |
| **R** | **-0.4** | **0.27** | **-0.54** |
| R Standard Error | 5.17E-3 | 0.01 | 7.3E-3 |
| t | -5.52 | 2.22 | -6.32 |
| p-value | 1.29E-7 | 0.03 | 7.8E-9 |
| H0 (5%) | rejected | rejected | rejected |
| No# of valid cases | 165 (all valid samples) | 66 (samples SST < 16 ºC) | 99 (samples > 16 ºC) |

**Figure 7** Coccolithophore paleoproductivity and SST Pearson correlation results: all valid samples (green), samples below 15.5 ºC (blue) and samples above 15.5 ºC (orange). Person correlation computed with StatPlus:mac, AnalystSoft Inc. - statistical analysis program for macOS®. Version v7. See http://www.analystsoft.com/en/


**5.1 Causes for coccolithophore productivity change**

**5.1.1 Intermediate coccolithophore productivity levels associated with terminations and interglacial periods**

The abrupt SST rise during the deglaciations is accompanied by rapid increases in coccolithophore productivity (TIV and TV; Fig. 3). The increasing SST are linked to the acceleration of the Atlantic Meridional Overturning Circulation (AMOC),
strengthened North Atlantic Current and to the northward migration of the North Atlantic frontal systems (e.g., McManus et



al., 2004; Voelker et al., 2010). Off western Iberia, in core MD03-2699, the deglacial SST increase and the subsequent particularly stable SSTs during MIS 11c were linked to a dominant influence of the subtropical waters of the Azores and Iberian Poleward Currents (Rodrigues et al., 2011; Voelker et al., 2010). Accordingly, during MIS 9e and MIS 11c, the abundance of warm coccolith taxa increased (Amore et al., 2012; Maiorano et al., 2015; Marino et al., 2014; Palumbo et al.,

2013a) as well as the abundance of tropical planktonic foraminifera species in cores MD01-2443 (de Abreu et al., 2005) and MD03-2699 (presence of *Globorotalia menardii* and *Sphaeroidinella dehiscens;* Voelker et al., 2010). These evidences point to the prevailing presence of subtropical oligotrophic waters during the later phase of the deglaciation and across interglacial MIS 11c. As a consequence of the limited nutrient supply, coccolithophore productivity, especially the opportunistic and fast-growing species, could have been limited to intermediate levels during MIS 11c.

Besides changes in the nutrient availability promoted by changes in the advection of the dominant water masses in the IbM, fluctuations in the upwelling regime, precipitation and arid plus windy conditions could also have played a role in determining coccolithophore productivity.

The addition of iron into the surface ocean positively affects the overall phytoplankton community (Blain et al., 2004; Martin et al., 1990). However, diatoms require more silica and iron (e.g., Merico et al., 2004), whereas coccolithophore blooms

might be related with silica and iron depletion (Merico et al., 2004; Tyrrell and Merico, 2004). This leads to the assumption that in an increased nutrient availability environment, associated with increased bioavailability of silica (Capellacci et al., 2013) and iron, diatoms could outcompete coccolithophores via nutrient competition, limiting coccolithophores to intermediate levels of productivity.

We here consider two main sources of additional silica and iron input to the Iberian margin: aeolian input and riverine

discharge. Periods of strengthened wind regime and aridity are generally limited to glacial substages and stadials (Desprat et al., 2009, 2017; Hodell et al., 2013; Oliveira et al., 2016; Sánchez Goñi et al., 2016). However, Rodrigues et al. (2011) interpreted the synchrony between high terrigenous input in core MD03-2699 and intensified dust export from North Africa (e.g. Helmke et al., 2008) as indicative of a strong wind regime in the IbM region during the deglaciation and early phase of MIS 11c. Such strong winds have been sufficient to replenish the surface waters with silica and iron, allowing diatoms to

bloom (Abrantes, 2000; Capellacci et al., 2013) and outcompete coccolithophores.

Also, based on evidence from the late Pleistocene (e.g. Margari et al., 2014), the transition from MIS 12 to MIS 11 is likely also marked by a transition from very cold winters and windy conditions in the western and southern Iberian margin to warmer conditions with increased winter precipitation (Desprat et al., 2009; Tzedakis et al., 2009). Higher humidity and precipitation would increase continental weathering and transport of silica and iron to coastal areas. And indeed the coccolith

fraction, in which the Sr/Ca ratio was measured, also shows increased silica and iron contents when compared to calcium (CF Si/Ca and CF Fe/Ca; Fig. 4), especially during deglaciation and the beginning of MIS 11c. Also, CF Si/Ca and CF Fe/Ca are intricately connected (r=0.95, p<0.01), and are both negatively correlated with coccolithophore productivity (r=-0.44, p<0.01 and r= -0.49, p<0.01, respectively). This could indicate that even if iron, likely brought by dust originated in the Sahara desert, might have played an important fertilizing effect for the overall phytoplankton community offshore the IbM





(Blain et al., 2004) in the past, diatoms most likely outcompeted coccolithophores, limiting their productivity due to increased competition for nutrients, when silica was in sufficient amount not to limit diatom blooms. All of these evidences suggest that irrespective of a moderate to strong upwelling regime, which increased the overall nutrients availability and phytoplanktonic productivity, only sufficiently higher the amounts of silica and iron introduced by increased wind and aridity or by increased precipitation, would lead to diatom blooms and coccolithophores to be outcompeted, most likely
explaining the intermediate level of coccolithophore productivity.

**5.1.2 High coccolithophore productivity levels associated with the transition from interglacial to glacial periods**

Coccolithophore productivity increases steadily from mid MIS 11c until the end of MIS 11b and remained generally high from late MIS 11c to MIS 10c. Although the increasing trend in coccolithophore productivity coincides with a decreasing trend in SST from 403 to 390 kyr (which returned a high negative correlation, r=-0.88, p<0.01), the high productivity
interval, from 398 kyr to 354 kyr, seems disconnected from SST influence (r=0.16, p=0.2).

The cooling of the SST during the transition from interglacial substages to glacial substages is related to the build-up of ice sheets on the continents and the subsequent changes in atmospheric and oceanic circulation, which subsequently resulted in a strength decline of the AMOC (e.g., McManus et al., 1999; Voelker et al., 2010), also evidenced by the decreasing trend in bottom water ventilation (e.g., Martrat et al., 2007). During interglacial substages, such as MIS 11c and MIS 9e, the wind
stress associated to the upwelling events is hypothesized to be lower than during glacial substages (e.g. Salgueiro et al., 2010). Whereas, during the transitions from interglacial to glacial substages, the narrower latitudinal temperature gradient caused by the expansion of the northern continental ice sheets and the associated location shifts in the North Atlantic frontal system, is hypothesized to have increased the northerly winds and lead to more intense upwelling or wind related mixing of the upper water column. As a consequence, higher turbulence would replenish the ocean surface with nutrients.

Because of the differences in the planktonic oxygen isotope records between cores MD03-2699 and MD01-2446 (more offshore) Voelker et al. (2010) suggested that the high variability in the closer to shore core MD03-2699 could reflect variations in upwelling of deeper waters into the thermocline. Indeed, Fiúza (1984) had already proposed that variations in wind stress could lead to the upwelling of different water masses. Stronger winds would favour upwelling intensification, which could lead to the upwelling of the deeper, nutrient-richer ENACWsp on the IbM (Fiúza, 1984), as already suggested
for some late Pleistocene periods (Salgueiro et al., 2010, 2014). Enhanced northerly wind stress could have intensified the upwelling, both in strength (upwelling deeper and nutrient-richer waters) and in distance to shore (reaching further offshore than today). Increased nutrient availability would thus support the whole phytoplankton community and decrease diatoms' and coccolithophores' competition for nutrients, especially for those coccolithophores species capable of rapid growth in a nutrient-rich environment, such as the *Gephyrocapsa caribbeanica* is thought to have been, given its cosmopolitan
distribution and dominance in the sediments (Baumann and Freitag, 2004; Bollmann et al., 1998). Note that this would only be possible given that bioavailable silica and iron were not sufficiently high to allow diatoms to outcompete coccolithophores, as seen before, and as the CF Si/Ca seem to show, with quite low and constant levels during the



coccolithophore productivity maxima, only interrupted by a period of increased CF Si/Ca and abrupt decrease in coccolithophore productivity, from ~375 to 365 ka (Fig. 4). The coccolithophore productivity maxima during the transition

from interglacial MIS 11c to glacial MIS 10a conditions indicate the fastest growth and calcification rates of the record, only comparable to mid-MIS 12 levels, in the beginning of our record, and preceding full glacial conditions. This supports the idea that the coccolithophore community was able to better perform under these transitional conditions, despite colder SST, and under windier and more turbulent settings and more intense upwelling.

### 5.1.3 Low coccolithophore productivity associated with abrupt cooling events and glacial MIS 12a

Coccolithophore productivity shows consistent minima along the record that, with the exception of glacial maximum stage MIS 12a and the ~10 kyr interval at the transition from MIS 11a to MIS 10c, are of short duration and related to abrupt climate change events. Thus, most of the minima coincide with the presence of colder and less saline surface waters that resulted from meltwater incursions onto the IbM evidenced by the increased IRD content, higher percentages of $C_{37:4}$ alkenones as well as higher percentages of coccoliths in the sediment belonging to the species *Gephyrocapsa margereli-G.*

*muellerae* (>4 µm) (Fig. 4; Marino et al., 2014; Rodrigues et al., 2011). The most extreme of these events are the Heinrich-type events (HtE; Stein et al., 2009; Fig. 4), with the ones during TIV and TV corresponding to the terminal stadial events of Hodell et al. (2015). The associated cooling brought the SSTs nearly to local glacial levels (Fig. 4; Rodrigues et al., 2011). Similar to their counterparts during the last glacial cycle, the meltwater events resulted in an accentuated decrease or nearly elimination of the AMOC and in a southward displacement of the North Atlantic frontal system with the subpolar/arctic front

moving into the latitudes of the IbM (Alonso-Garcia et al., 2011; Hodell et al., 2008; McManus et al., 1999; Rodrigues et al., 2011, 2017). This lead to the advection of polar and subpolar water masses onto the IbM evidenced by the increased percentage of polar planktonic foraminifera *Neogloboquadrina pachyderma* in the nearby core MD01-2446 (Fig. 4; Marino et al., 2014) and in cores off Galicia (Desprat et al., 2007, 2009) and in accordance with evidence from the last glacial cycle (e.g., Salgueiro et al., 2010, 2014).

The abrupt coccolithophore productivity drops at 438 ka and at 340 ka prior to and during TV and TIV, respectively, are both related to HtE (Fig. 4). As previously mentioned, during HtE detrital carbonate might have negatively biased our coccolithophore productivity. However, detrital carbonate might have acted as a low productivity signal amplifier instead of attributing false low coccolithophore productivity in such intervals. An extreme decline in coccolithophore productivity during the TV HtE/terminal stadial event was also observed at IODP Site U1313 in the central North Atlantic basin

(Cavaleiro et al., 2018). During those events the abrupt SST decrease coupled with increased turbidity and blockage of sunlight (due to melting icebergs and/or sea ice cover) would have decreased the coccolithophores' ability to survive. These conditions resemble those in the present-day polar domain of the Norwegian-Greenland Sea (Baumann et al., 2000). Here, a less diverse coccolithophore community was observed with a slightly diminished coccolithophore population, which could well support the abrupt decreases detected in our coccolithophore productivity record.

Additional coccolithophore productivity minima are detected at 389 ka, 382 ka, 371 ka, 364 ka, the latter dropping to HtE



levels (Fig. 4), 346 ka, 334 ka, 319 ka and 316 ka. Despite no significant concomitant IRD peaks in core MD03-2699, these decreases are associated with increases in the abundance of the $C_{37:4}$ alkenone and colder SST (Fig. 4; Rodrigues et al., 2011) and the presence of *N. pachyderma* and *G. margereli-G. muellerae* (>4 μm) in core MD01-2446 (Fig. 4; Marino et al., 2014), testifying a rapid change to colder and less saline conditions at this latitude. The four older drops in coccolithophore

productivity happened concomitantly with abrupt increases in *Neogloboquadrina pachyderma* abundance at ODP Site 980 (55°29'N 14°42'W) in the eastern North Atlantic (Fig. 4; McManus et al., 1999; Oppo et al., 1998), placing them in a broader spatial scale of climate deterioration. Thus, we conclude that the IbM was subjected to drastic surface water changes that led to a coccolithophore community decrease in the IbM and generated abrupt decreases in our coccolithophore productivity record.

Finally, the reduced productivity during MIS 12a at the location of core MD03-2699 is in stark contrast to the open ocean record of IODP Site U1313 (Cavaleiro et al., 2018) and evidence from the western Iberian margin for the glacial maxima of MIS 2 and MIS 6 (Salgueiro et al., 2010, 2014), all of which point to increased productivity during the glacial maxima. Such a prolonged decrease in coccolithophore productivity is also not observed during glacial maximum MIS 10a, when productivity at site MD03-2699 remained at intermediate levels between the abrupt cold events (Fig. 4). The IRD record of

ODP Site 980 and persistent high percentages of *N. pachyderma* at ODP Site 980 (Oppo et al., 1998; McManus el al., 1999) and in core MD01-2448 in the Bay of Biscay (Toucanne et al., 2009) point to the prolonged presence of (sub)polar waters in the eastern North Atlantic throughout MIS 12a. These waters appear to also have affected the IbM, by hampering upwelling (enhanced upwelling is seen as cause for increased productivity at the IbM during glacial maxima; Abrantes et al., 2000; Salgueiro et al., 2010, 2014) and thus coccolithophore productivity in core MD03-2699.

**5.2 Coccolithophore productivity, NAR and alkenone flux: the mismatch**

Previous studies in the IbM used coccolith-derived proxies, such as coccolith assemblages, nannofossil accumulation rates (NAR), and total alkenone fluxes (produced mostly by coccolithophores) to reconstruct changes in coccolithophore paleoproductivity (Amore et al., 2012; Maiorano et al., 2015; Marino et al., 2014; Palumbo et al., 2013a). The traditional proxies depend not only on the supply of coccoliths or organic compounds but also on dilution by minerals and other

sediment constituents and on changing preservation conditions (Rullkötter, 2006). As the CF Sr/Ca ratio is an independent proxy, i.e. independent of sedimentation and accumulation rates, it offers a new perspective into coccolithophore productivity dynamics. We thus assess how well the different productivity proxy records compare with our coccolithophore productivity data.

A visual comparison of our coccolithophore productivity record with the NAR and alkenone flux records already suggests

that their relationship is not straightforward (Fig. 8). While in some intervals coccolithophore productivity, NAR and alkenone flux show similar trends, in others, they diverge. Correlation analysis between coccolithophore productivity record and the NAR and the alkenone flux return positive but weak relationships (less than r=0.3, statistical analysis not shown).



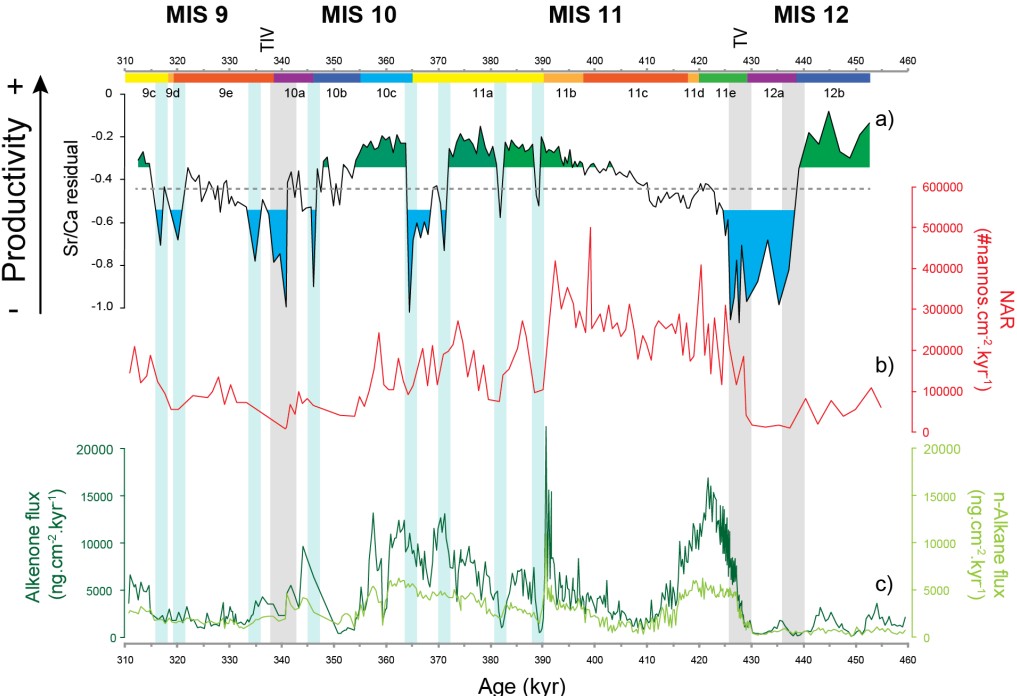

**Figure 8** Coccolithophore paleoproductivity reconstruction with "high" and "low" coccolithophore productivity levels
highlighted in green and blue shading, respectively (a) compared to: b) NAR (Amore et al., 2012) and c) total alkenone flux
and n-alkane flux (Rodrigues et al., 2011). Chronology as in Fig. 3. Vertical bars: grey bars correspond to Heinrich-type
events and blue bars to short-lived events of decreased coccolithophore productivity.

### 5.2.1 Coccolithophore productivity and alkenone flux

A synchronous high amplitude increase in the alkenone and the n-alkane fluxes during the deglaciation and beginning of
MIS 11 (Fig. 8) made Rodrigues et al. (2011) suggest that coccolithophores had been mostly nourished by a terrestrial input
of nutrients either by winds or pluvial discharges. Our coccolithophore productivity record also indicates a synchronous,
high amplitude rise. However, in early MIS 11, after the previously mentioned synchronous increase during the deglaciation,
the alkenone and n-alkane fluxes show a relatively fast and high amplitude decrease whereas the coccolithophore
productivity keeps increasing. Such a mismatch between coccolithophore productivity and alkenone/n-alkane records is also
reflected by the very low correlation coefficient (r=0.08, p=0.31). Moreover, the high positive correlation between the
alkenone and n-alkanes fluxes (r=0.81, p<0.01) suggests that the conditions that allowed a higher continental input, i.e.

accumulation and preservation of the n-alkanes (originating from terrestrial plants), also favoured the accumulation and preservation of the alkenone compounds. The discrepancy between our coccolithophore productivity record and the alkenone and n-alkane fluxes suggests that the changes of the latter two mostly reflect fluctuations in the conditions that led to higher

export, accumulation and/or preservation of those organic compounds rather than coccolithophore growth, calcification rate and productivity.

We further ran cross-spectral analysis between coccolithophore productivity and alkenone records to check if there were similar semi-periodicities in both time series and respective leads and lags (Fig. 9). In fact, cross-spectral analysis confirms no significant periodicities between the two time series at Milankovitch frequencies. Millennial scale changes are, however,

coherent in both time series at ~ 7.8 and ~ 4.4 kyr and the alkenone record leads the coccolithophore productivity by 0.82 and 0.43 kyr, respectively (associated errors of 0.4 and 0.2 kyr, respectively). Coccolithophore productivity leads the alkenone record by 0.9 kyr at quasi-periodicities of ~ 3.7 kyr. Since coccolithophores are the only known organisms producing enough alkenones to get exported and preserved in the sediments in significant amounts, the spectral analysis results corroborate our suggestion that the alkenone record is actually mostly offering a signal of preservation in the

sediments.

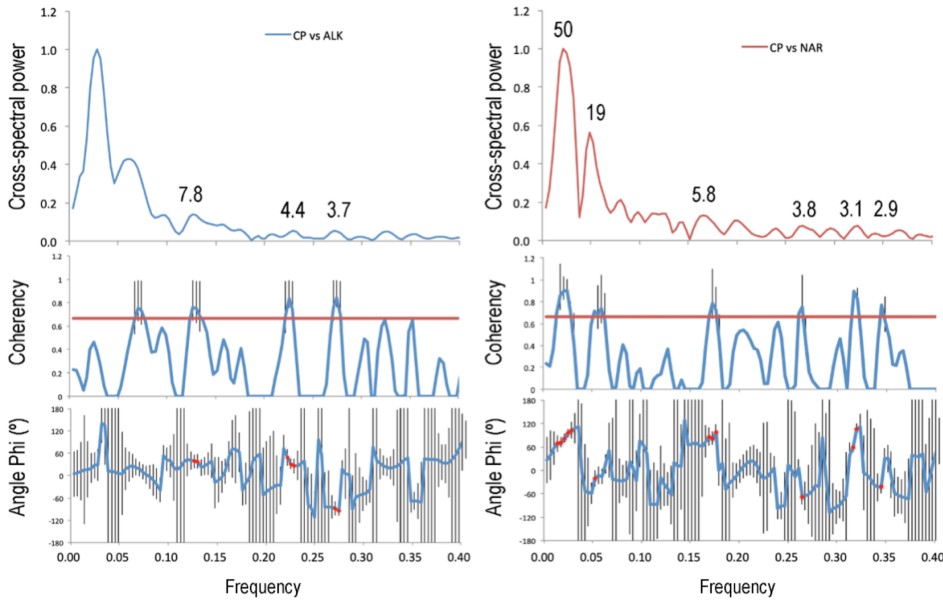

**Figure 9** Cross-spectral analysis results for Coccolithophore paleoproductivity (CP) and alkenone flux (ALK – blue line; Rodrigues et al., 2011) and nannofossil accumulation rate (NAR – red line; Amore et al., 2012). The quasi-periodicities are



given on top each significant peak on the cross-spectral results. The red line in coherency results defines the significance level of 80% (bandwidth is 0.02). All estimation errors are shown as vertical lines. Red dots highlight the angles for which significant quasi-periodicities exist.

### 5.2.2 Coccolithophore productivity and NAR

Amore et al. (2012) and Palumbo et al., (2013b) calculated the NAR for core MD03-2699 and suggested generally increased
coccolithophore productivity during interglacials compared to glacial substages (Fig. 8). Indeed, both NAR and our coccolithophore productivity record show minima during MIS 12a, MIS 10a and several of the abrupt (HtE) climate events. However, NAR does not reflect increased accumulation during periods of higher coccolithophore productivity, e.g. from 395 kyr to 355 kyr. Contrarily, the NAR reveals large amplitude shifts during times of high coccolithophore productivity.

The coccolithophore productivity and NAR correlation analysis also returned a low correspondence, but higher than with the
alkenone flux record (r=0.28, p<0.01). And, despite coccolithophore productivity and NAR records showing synchronous increasing trends during MIS 11c to MIS 11b, the relationship between the two is not straightforward.

The cross-spectral analysis between the coccolithophore productivity and NAR time series detected that at the millennial time-scales of ~ 5.8 kyr and ~3.1 kyr (Fig. 9) NAR is leading coccolithophore productivity by 1.4 kyr and 0.7 kyr, respectively (associated errors of 0.3 and 0.1 kyr, respectively). Since NAR is calculated according to the total number of
coccoliths found in the sediments these findings again support that some specific export or preservation conditions might have happened prior to the increased productivity of coccolithophores. Interestingly, we found coccolithophore productivity leading the NAR at the precession frequency (~ 19 kyr) by 0.1 kyr, as well as at the higher frequencies of 3.8 kyr and 2.9 kyr by 0.72 kyr and 0.33 kyr, respectively. This means that despite some noise within the NAR signal, it does carry our coccolithophore productivity signal, which was not evidenced with the alkenone record.

On glacial to interglacial time scales and during abrupt climate events, the North Atlantic has been subjected to significant changes in the depth of the lysocline (depth below which carbonate starts to dissolve) and in the carbonate compensation depth (depth below which no carbonate is found due to dissolution). We further studied the possibility of dissolution causing the asynchrony between coccolithophore productivity and the NAR. Short-lived dissolution events are likely to have occurred in the deeper IbM area during MIS 12 and MIS 10 glaciations, as suggested by Marino et al. (2014) for core MD01-
2446, as a consequence of the northward and upward progression of the southern sourced waters/Antarctic Bottom Water (e.g., Voelker et al., 2010). At a water depth of 1895 m, site MD03-2699 would have been located above the Antarctic Bottom Water influence (placed near 2500 m in the western North Atlantic; Thunell et al., 2002). Two lines of evidence confirm that corrosive southern sourced waters did not affect the coccolithophore assemblages in core MD03-2699: a) there is no evidence for increased abundance of larger, more calcified coccoliths, more resilient to dissolution, in detriment of
smaller and less calcified coccoliths in the coccolithophore flora (Amore et al., 2012); b) at IODP Site U1385, located about 500 m deeper than our site, Maiorano et al. (2015) found low coccolith dissolution indices, varying between 1 and 0.9, against the values below 0.25 (MIS 12a) and 0.5 (MIS 10a) for core MD01-2446 at 3570 m (Marino et al., 2014). Since



carbonate dissolution does not seem to have been relevant in core MD03-2699, the asynchrony between the NAR and the coccolithophore productivity record must result from other processes.

### 5.2.3 Why such a mismatch?

One possibility is linked to changes in the phenology of coccolithophore productivity, i.e. changes in the yearly timing when coccolithophores are more productive. Nowadays, coccolithophores in the IbM start to bloom in late winter and spring. Blooms can occur for 6 months (from late winter into summer months) with peak bloom months during spring (April, May; Hopkins et al., 2015). If this phenology was maintained in the past, it would be expected that the coccolithophore productivity record would show a precessional cyclicity, following either spring or summer insolation maxima along with a ~6 kyr cyclicity, when productivity would be high at both seasons, such as today, or when a shift to the summer months would occur, most likely associated to colder climatic conditions (e.g., Cavaleiro et al., 2018). Indeed our coccolithophore productivity record shows precession (~21 kyr and 17 kyr) as well as ~6 kyr quasi-periodicities (harmonics of precession, Fig. 6). This suggests precession, through insolation modulation, as a controlling factor of coccolithophore phenology and thus their productivity. We suggest that coccolithophore phenology in the IbM either remained similar to today or shifted to a predominant summer productive peak during cold substages and stadials, and as already suggested to the mid North Atlantic (Cavaleiro et al., 2018).

To check for the relationship between coccolithophore productivity and insolation at precession and precession harmonics we ran cross-spectral analysis with equinoxes and solstices peak daily insolation (Fig. 10). The results returned a non-significant peak around 6 kyr for spring, whereas summer solstice insolation led coccolithophore productivity by 1.5 kyr (± 0.34 kyr) and autumn equinox and winter solstice lagged coccolithophore productivity by 0.9 kyr (± 0.36 kyr) and 1.7 kyr (± 0.28 kyr), respectively. This corroborates the theory that coccolithophore phenology might have changed from spring to summer during colder periods while autumn and winter insolation changes had a diminished effect on coccolithophore productivity.





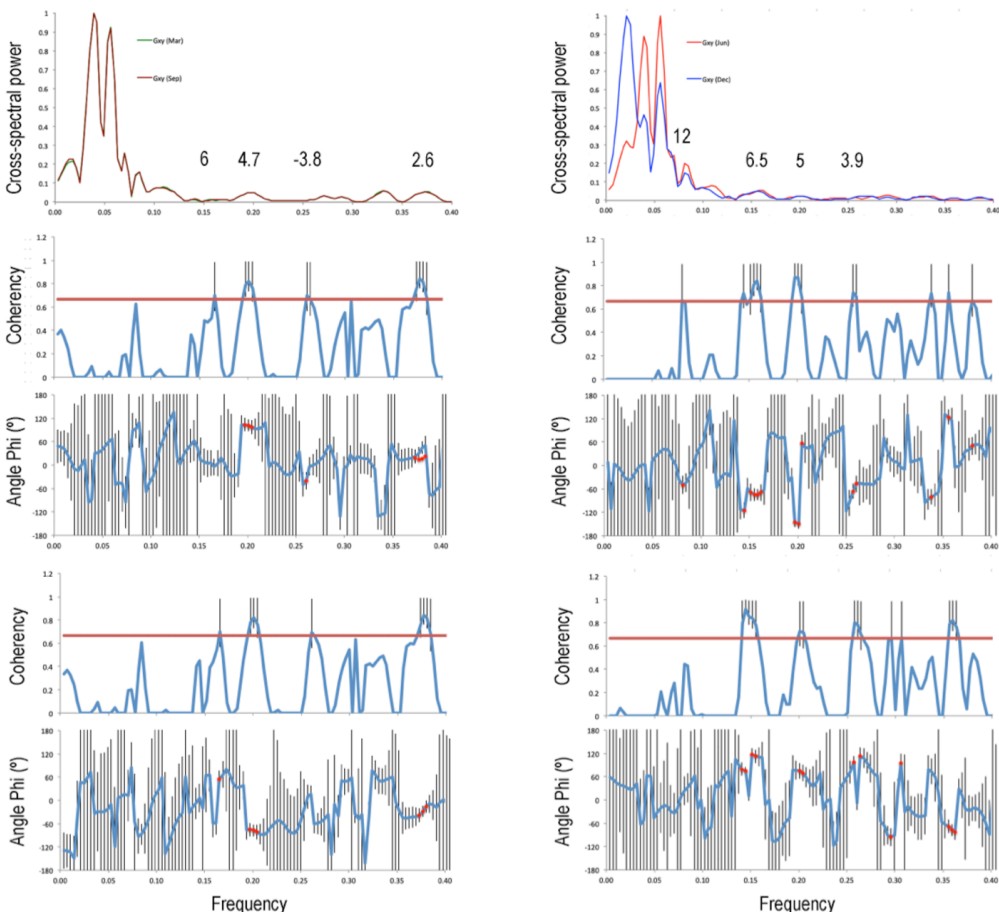


**Figure 10** Cross-spectral analysis results for Coccolithophore paleoproductivity (PC) and insolation curves: March and September cross spectral analysis results (left panel), June and December cross spectral analysis results (right panel; insolation data from Laskar et al., 2011). The quasi-periodicities are given on top the first four significant peaks on the cross-spectral results. The red line in coherency results defines the significance level of 80% (bandwidth is 0.02). All estimation 550 errors are shown as vertical lines. Red dots highlight the angles for which significant quasi-periodicities exist.

If such changes in phenology did happen, then, during periods of colder climate in the IbM, coccolithophores would be forced into a narrower time window to flourish (i.e. increased insolation, higher nutrient availability and less competition



with diatoms). Coccolithophore productivity would thus rise when nutrients and insolation increase but the yearly time-
window for higher productivity would become narrower as SST would decrease as a consequence of the more frequent
arrival of (sub)polar waters. This could ultimately lead to lower NAR and decreased accumulation of alkenones as well, even
at times when coccolithophores were calcifying faster and more efficiently because the yearly time window for export to
happen would be narrower.

**6 Conclusions**

With this study we discuss the different factors controlling coccolithophore productivity in the IbM. Coccolithophore
productivity is primarily affected by climatic changes, namely changes in temperature and ocean circulation that led to the
predominance of different water masses in the IbM. Presence of (sub)polar surface water masses during glacial substages
and stadials resulted in lower coccolithophore productivity, in particular during HtE stadials. On the other hand, the
prevalence of nutrient-poorer subtropical waters during interglacial substages and increased riverine input of iron and silica
increased coccolithophores' and diatoms' competition for nutrients leading to relative intermediate coccolithophore
productivity levels. The transitions between interglacial and glacial substages were characterized by coccolithophore
productivity maxima due to the emerging presence of nutrient-rich waters brought by strong northerly wind forced
upwelling. These factors allowed coccolithophores to bloom and attain their relative calcification rate maxima and ultimately
their productivity maxima. During the transition from interglacial to glacial substages coccolithophores were nonetheless
forced to change their phenology, and their most productive season changed according to the best conditions provided, most
likely from spring to summer because the SSTs during winter and spring were still too cold. Ultimately, this contributed to
the dissonance between coccolithophore productivity, NAR and alkenone flux records, as a consequence of the yearly time-
window narrowing of coccolithophore productivity.

**Data availability.** The data used will be available from the Pangaea data repository https://www.pangaea.de

**Author contribution.** CC and AV designed the study. CC prepared the samples and analysed them under the supervision of
HS. CC drafted the paper and all authors contributed for the discussion and to the final version.

**Competing interests.** The authors declare that they have no conflict of interest.

**Acknowledgements.** The study was financially supported by the Fundação para a Ciência e a Tecnologia (FCT; Portugal)
through projects PORTO (PDCT/MAR/58282/2004), INTER-TRACE (PTDC/CLI/70772/2006) and CCMAR
(UID/Multi/04326/2019). C.C. acknowledge her doctoral fellowship (SFRH/BD/84187/2012) and A.V. her FCT
Investigador contract (IF/01500/2014). C.C. also expresses gratitude for the laboratorial and analytical support at the
Geosciences Department, University of Oviedo, and at MARUM and Geosciences Department, University of Bremen. CC



further acknowledges the effort and precious help of Ana Méndez-Vicente while measuring the coccolith fraction samples at the University of Oviedo. CC would also like to thank Dulce Oliveira, Fátima Abrantes, Filipa Naughton and Teresa
Rodrigues for the helpful comments to the manuscript.

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





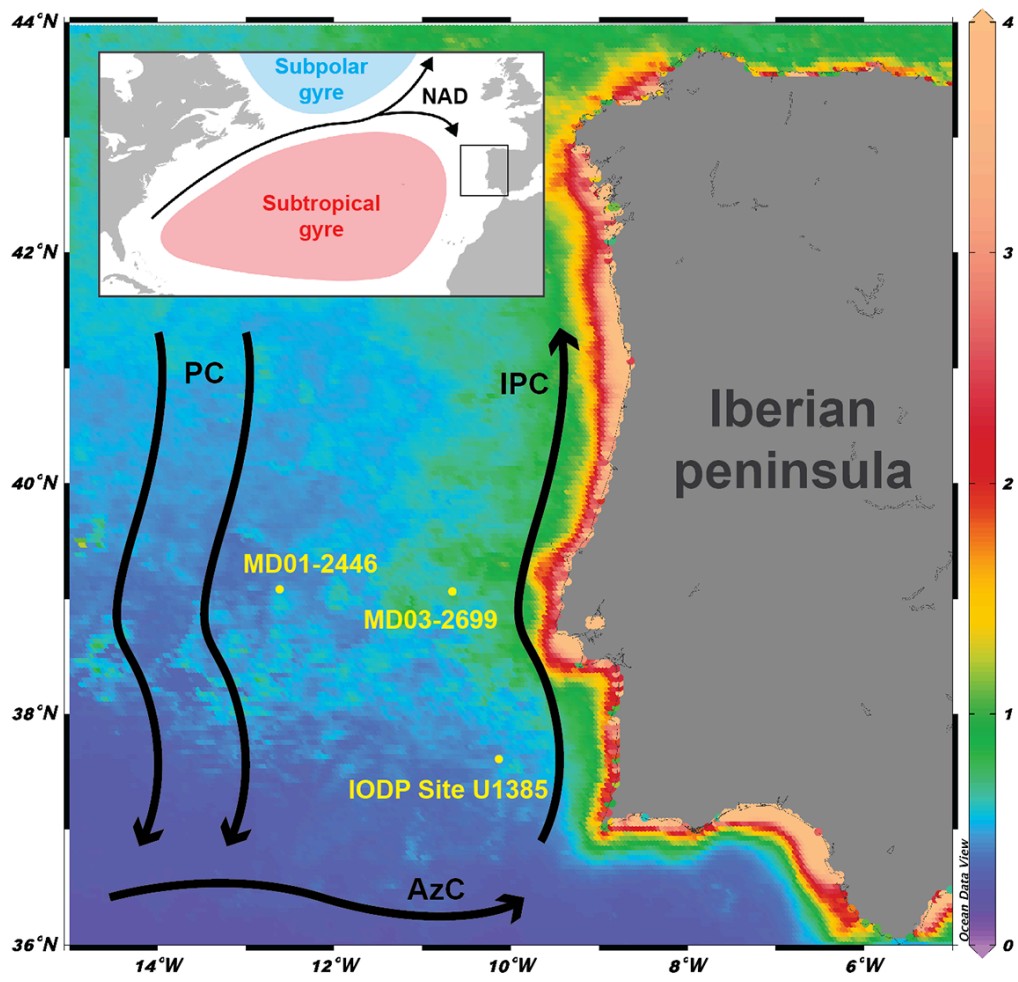


**Figure 1: Core location and major currents in the area: NAD = North Atlantic Drift; PC = Portugal Current; IPC = Iberian Poleward Current; AzC = Azores Current. Background: chlorophyll *a* concentration (mg.m⁻³; March, April and May average, 2003-2018) derived from MODISA satellite data available at http://disc.sci.gsfc.nasa.gov/giovanni.**





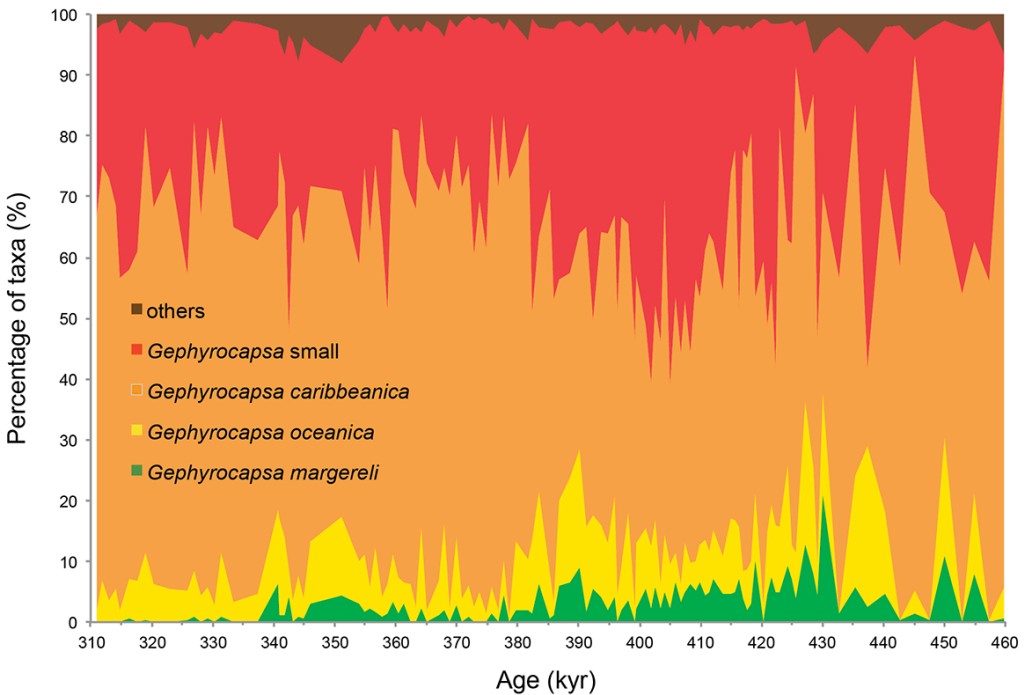

**Figure 2: Composition of coccolith assemblages in core MD03-2699 based on Amore et al. (2012). Note that only a minor percentage of coccoliths belong to groups other than the dominant Gephyrocapsids. The chronology is based on the age model of Voelker et al. (2010).**

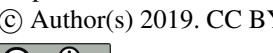



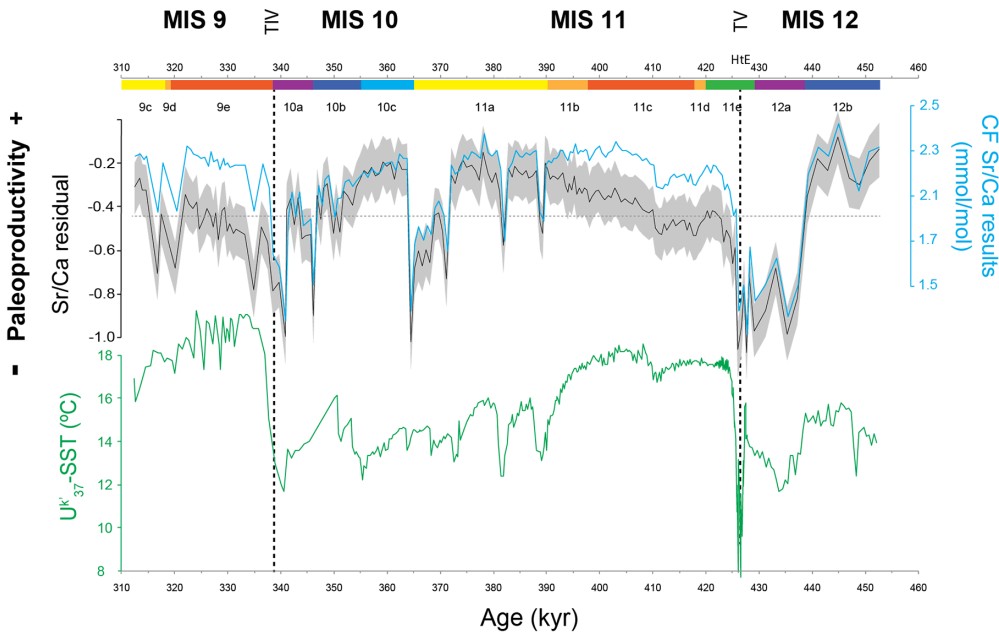

**Figure 3: Coccolithophore paleoproductivity results from core MD03-2699: coccolith fraction (CF) Sr/Ca ratio and**
**the resulting CF Sr/Ca residual with confidence interval (grey shading represents Monte Carlo 20-80% interval),**
**$U^{k'}_{37}$-based reconstruction of sea surface temperature (Rodrigues et al., 2011) The horizontal dashed line marks the**
**average value for CF Sr/Ca and CF Sr/Ca residual. The chronology is based on the age model of Voelker et al. (2010),**
**Marine isotope stages (MIS) and substages marked according to Railsback et al. (2015), vertical dashed lines**
**highlight Terminations V and IV.**





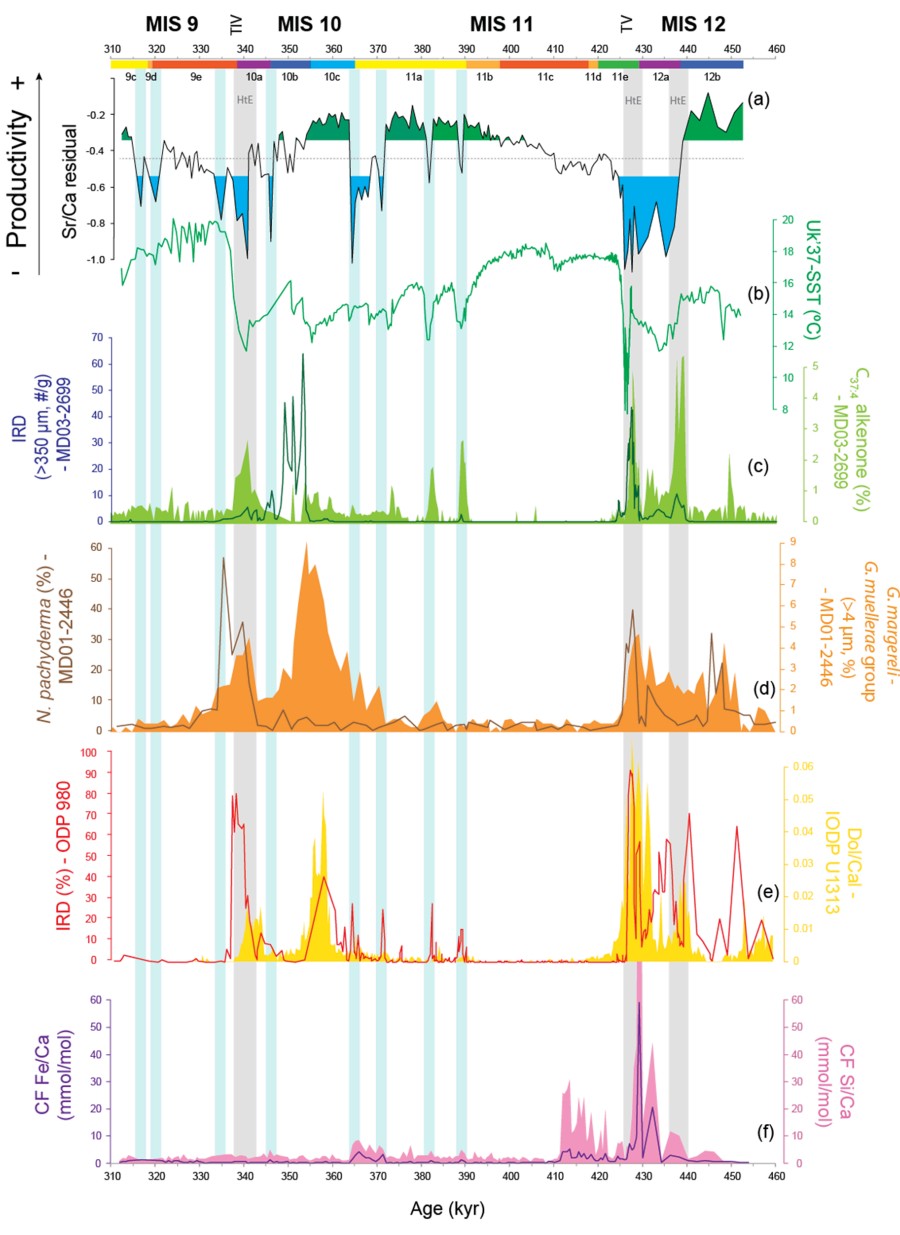






**Figure 4: Coccolithophore paleoproductivity reconstruction with "high" and "low" coccolithophore productivity levels highlighted in green and blue shading, respectively (a) compared to: b) $U^{k'}_{37}$-based reconstruction of sea surface temperature from core MD03-2699 (Rodrigues et al., 2011); c) ice-rafted debris (IRD) abundance (Voelker et al., 2010) and heptatriatetraenone ($C_{37:4}$ alkenone) abundance (Rodrigues et al., 2011), both from core MD03-2699; d)**

***Neogloboquadrina pachyderma* relative abundance and *Gephyrocapsa margereli* and *G. muellerae* group relative abundance, both from core MD01-2446 (Marino et al., 2014); e) percentage ice-rafted debris (IRD) from ODP Site 980 (McManus et al., 1999) and Dolomite/Calcite ratio from IODP Site U1313 (Stein et al., 2009); and f) coccolith fraction (CF) Fe/Ca and Si/Ca ratios. Note that the MD03-2699 lithic fragments were counted in a coarser size fraction (>315 µm) than the standard size fraction (>150 µm; Hemming et al., 2004) and thus most likely only record**

**major ice-rafting events at the IbM, as suggested by Marino et al. (2014). Chronology as in Fig. 3. Vertical bars: grey bars correspond to Heinrich-type events and blue bars to short-lived events of decreased coccolithophore productivity.**

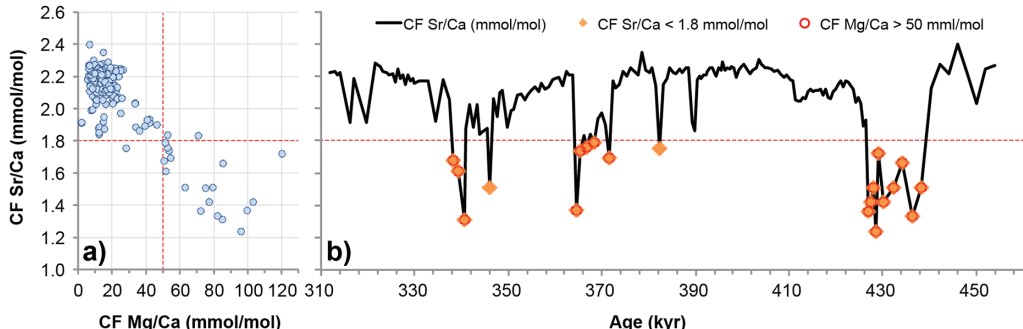

**Figure 5: Detrital carbonate analysis: a) coccolith fraction (CF) Sr/Ca results and CF Mg/Ca cross-plot. Vertical and horizontal red lines delimit thresholds at 1.8 mol/mol of Sr/Ca and 50 mmol/mol of Mg/Ca, respectively; b) CF Sr/Ca samples possibly subjected to detrital carbonate contamination (red makers), notably associated with cold and abrupt events.**





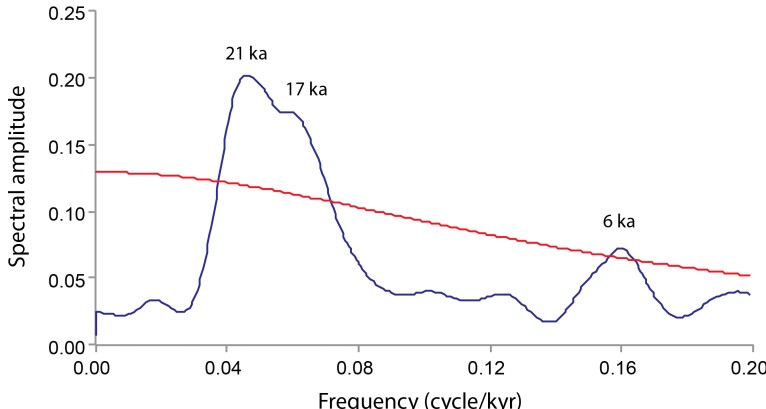

**Figure 6: Coccolithophore paleoproductivity time-series analysis showing significant peaks at 21, 17 and 6 ka. The red line defines the significance level of 80% (bandwidth is 0.02), indicating that the peaks at 21 and 17 ka are not independent (may represent the same frequency given the uncertainty of dating and resolution).**





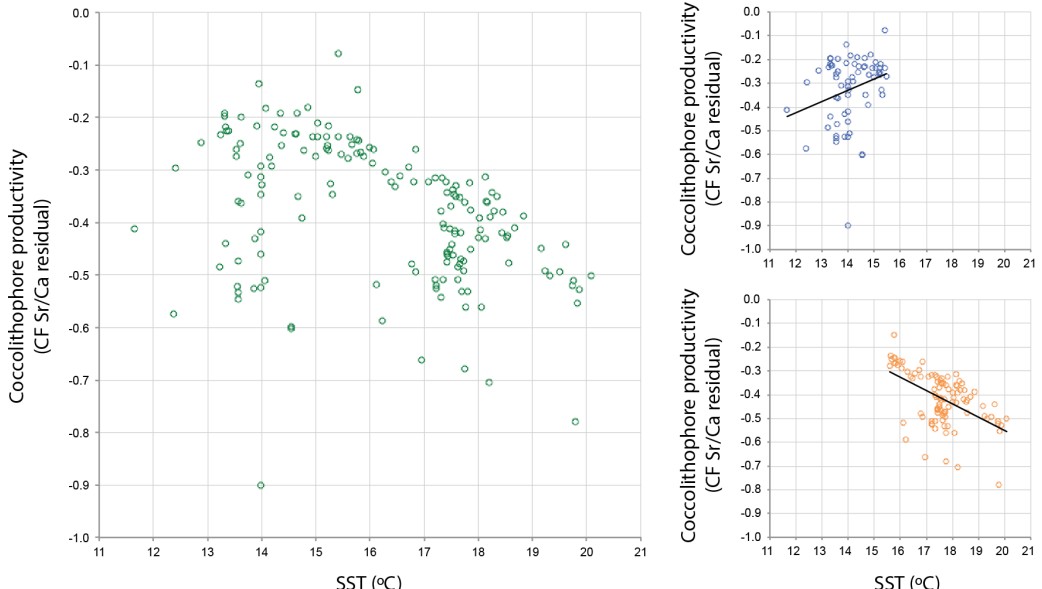

| Correlation Coefficients - Coccolithophore productivity (CF Sr/Ca residual) vs SST (ºC) | | | |
|---|---|---|---|
| | Sr/Ca residual vs. SST (ºC) | Sr/Ca residual vs. SST (ºC) | Sr/Ca residual vs. SST (ºC) |
| **R** | **-0.4** | **0.27** | **-0.54** |
| R Standard Error | 5.17E-3 | 0.01 | 7.3E-3 |
| t | -5.52 | 2.22 | -6.32 |
| p-value | 1.29E-7 | 0.03 | 7.8E-9 |
| H0 (5%) | rejected | rejected | rejected |
| No# of valid cases | 165 (all valid samples) | 66 (samples SST < 16 ºC) | 99 (samples > 16 ºC) |

**Figure 7: Coccolithophore paleoproductivity and SST Pearson correlation results: all valid samples (green), samples**
**below 15.5 ºC (blue) and samples above 15.5 ºC (orange). Person correlation computed with StatPlus:mac, AnalystSoft Inc. - statistical analysis program for macOS®. Version v7. See http://www.analystsoft.com/en/**

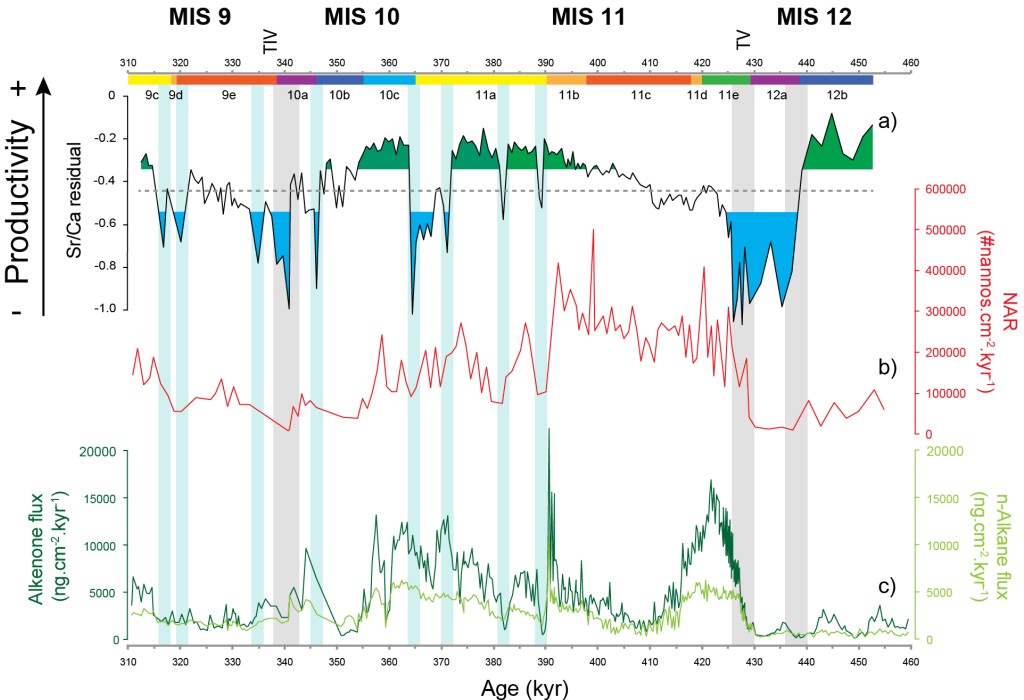

**Figure 8: Coccolithophore paleoproductivity reconstruction with "high" and "low" coccolithophore productivity levels highlighted in green and blue shading, respectively (a) compared to: b) NAR (Amore et al., 2012) and c) total alkenone flux and n-alkane flux (Rodrigues et al., 2011). Chronology as in Fig. 3. Vertical bars: grey bars correspond to Heinrich-type events and blue bars to short-lived events of decreased coccolithophore productivity.**





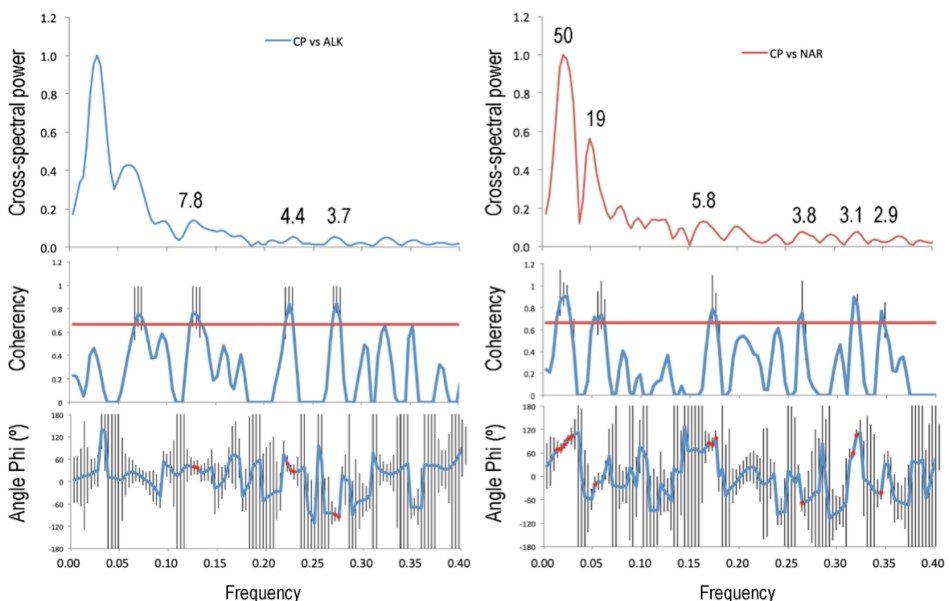

**Figure 9: Cross-spectral analysis results for Coccolithophore paleoproductivity (CP) and alkenone flux (ALK – blue line; Rodrigues et al., 2011) and nannofossil accumulation rate (NAR – red line; Amore et al., 2012). The quasi-periodicities are given on top each significant peak on the cross-spectral results. The red line in coherency results defines the significance level of 80% (bandwidth is 0.02). All estimation errors are shown as vertical lines. Red dots highlight the angles for which significant quasi-periodicities exist.**






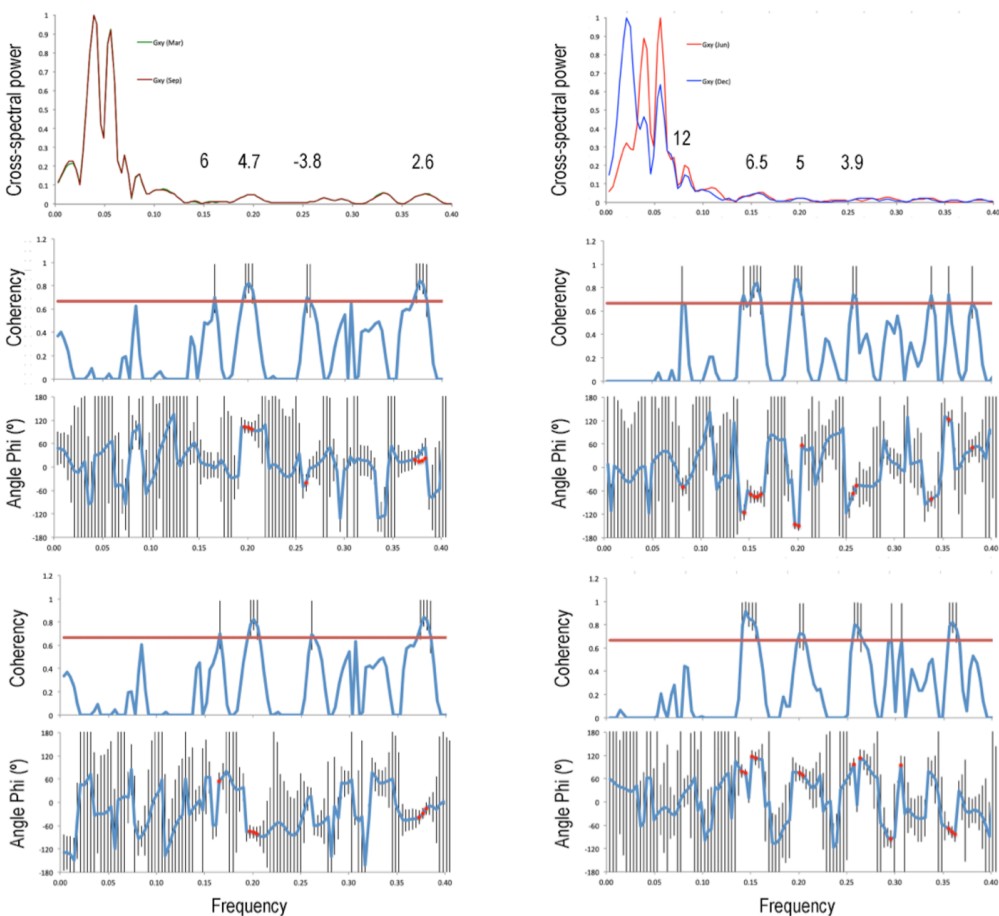

**Figure 10: Cross-spectral analysis results for Coccolithophore paleoproductivity (PC) and insolation curves: March and September cross spectral analysis results (left panel), June and December cross spectral analysis results (right panel; insolation data from Laskar et al., 2011). The quasi-periodicities are given on top the first four significant peaks on the cross-spectral results. The red line in coherency results defines the significance level of 80% (bandwidth is 0.02). All estimation errors are shown as vertical lines. Red dots highlight the angles for which significant quasi-periodicities exist.**
