# Peer review of "Coccolithophore productivity at the western Iberian Margin during the middle Pleistocene $(310-455~\mathrm{ka})$ – evidence from coccolith Sr/Ca data"

_Climate of the Past, 2019_

## Referee Comment (RC1) · Tom Dunkley Jones (Referee) · 26 Nov 2019

This is a very nice piece of work, presenting new high-resolution coccolith Sr/Ca records from the mid Pleistocene of the Iberian Margin. Coccolith Sr/Ca is an under-used but potentially powerful proxy for understanding the cellular growth and calcification rates of the dominant marine calcifying phytoplankton, the coccolithophore algae. Of particular value in this manuscript, is the integration of these records of growth rates, with other records of export flux to sediments of both the organic (alkenones) and inorganic (coccolith) carbon fixed by these phytoplankton. Together these allow a nuanced

interpretation – including the coupling / decoupling - of surface ocean growth conditions and aggregated net export.

I would recommend publication with revisions. In particular I would recommend shortening the manuscript and focusing on the strongest signals within the data in order to generate the impact that this work deserves.

Comments (in order of the text):

Line 17 – instead of "climate models" better to say "Earth System Models" as the common understanding of a "climate model" is one that doesn't include biogeochemistry. And again at Line 48 (and anywhere else) – "climate models" including coccolithophore productivity – better to talk about Earth System and/or biogeochemical and/or carbon cycle models.

Line 18 – "coccolithophore paleoproductivity past reonconstructions" doesn't make sense

Line 23 – define SST at first usage

Lines 38-42 long sentence that jams together two concepts – split.

INTRODUCTION

In both the introduction and the discussion, I feel the absence of a clearly articulated question – what is the "knowledge gap" and how does this paper address that gap? There are suggestions of problems in the representation of coccolithophore production and export in biogeochemical models, but no sense of what the specifics of these are, or how they might be addressed by this study. When I first read this section, I was not convinced that models could be informed by new coccolith Sr/Ca records (but see below). Then there is the time period studied – there is a general overview of the significant changes going on through this interval, but the rationale for looking at the coccolithophore response is so broad that it loses meaning:

"to evaluate this phytoplankton group's behaviour and gain a better understanding of its response to climate conditions during glacials, interglacials, deglaciations and the transition from interglacial to glacial conditions, at both orbital and sub-orbital time scales." (line 85 & on). Or:

"We aim to characterise long-term changes in coccolithophore productivity in such a system, where their behaviour in the past remains unknown." (line 74) or:

"...and evaluate the main factors influencing coccolithophore productivity." (line 77)

I would really like specifics of: 1) the dynamics / processes that you seek to investigate and 2) why these intervals.

I think part of your struggle is related to: 1) setting up the Sr/Ca as something that is a "better" measure of coccolithophore "productivity" than other approaches – e.g. NAR / alkenone accumulation; and 2) equating coccolith Sr/Ca with quite a loose concept of "productivity". Together these end up setting off your "productivity" records "against" one another, rather than being mutually informative about different components of the growth – export – accumulation system. This seems to lead to a discussion which is phrased in terms of "mismatch" rather than one that allows the complexity of the system response to be seen, because, you've got the advantage of multi-proxy data covering different aspects of the same system.

My recommendation is that you recast this introduction somewhere along these lines:

1) There are significant uncertainties about the complex interactions between coccolithophore growth rates, nutrient dynamics, seasonality, export (carbonate / organic carbon), dissolution and final accumulation / burial rates. These uncertainties make modelling the responses of this system to modern environmental change problematic.

2) These questions can only be addressed with: a) records that test the dynamic response of various components of the system over a reasonable range of change (i.e. palaeo records); and b) through multi-proxy studies of growth environment (Sr/Ca) and

export of both organic (alkenones) and inorganic (NAR) carbon.

3) Then make the argument for the particular time period studied providing the chance to test a range of particular environmental conditions – and make your introduction to the time period outline what these might be – e.g. upwelling, seasonality, temperature...

With this set-up, hopefully you'll then be able to circle round in the discussion and answer these questions.

Sr/Ca VARIATIONS AND ASSEMBLAGE CHANGES

Line 68 – 69: I'd like some more justification for the assumption that "assemblage changes don't matter". My reading of the Fink et al. 2010 paper was that the abundance of the larger Calcidiscus leptoporus did have a significant impact on CF Sr/Ca. I'm also suspicious of using the logic that in some instances in the modern oceans CF Sr/Ca changes coincide with productivity changes, therefore, it must be productivity, when coincident assemblage changes haven't been properly considered. Given expertise of Baumann and Stoll, I would like to see some more justification of this point, that CF Sr/Ca are really dominated by changes in growth rate, rather than assemblages, especially with respect to Calcidiscus, which I suspect can contribute strongly to some records that show large Sr/Ca variations.

Later – Line 179 – you talk about being in the Gephyrocapsa acme and that this makes assemblage variability less of an issue for CF Sr/Ca, but this somewhat admits that assemblage change can be an issue under other circumstances. A clear delineation – even without hard and fast data / rules – between when assemblage changes are likely and not-likely a problem with CF Sr/Ca would be better than trying to imply there is no issue. Please be precise with the logic and transparent with the reader as to if and where assemblage change might be playing a significant role.

Line 184 – gephyrocapsids's – lower case I believe.

Lines 255- bias from other carbonate phases – from Figure 5 it looks like the trend starts before the cut-off used, from more like 30 mmol/mol Mg/Ca. And could you please clarify which "cut off" you are using, whether this is Sr/Ca below 1.8 mmol/mol or higher values of Mg/Ca? If tracing contamination from other carbonates, would it make more sense to use the Mg/Ca values for the cut-off? For example you could cut off more stringently, at ∼30 mmol/mol Mg/Ca and yet maintain what look like more robust / primary signals of lower Sr/Ca within data that would pass this criteria. You would lose a few more data points in total, but I think this would be a more defensible cut-off point and rationale.

Line – 272 – "This interval represents 39% of the whole sampling variation...", doesn't make sense to me, please clarify. Do you mean something like the clipped data represents 39% of the dynamic range in Sr/Ca of the full sample set?

Lines 288-289: is significance level of 80% really enough to be confident that the 6ka peak is real? I'm not really convinced. I don't think this spectral analysis reveals anything and is a distraction for the reader - you have a nice tight coupling to well-resolved climate records (Uk37) and a good age model for making 104-year correlations to N. Atlantic climate records, so I don't think there's a need to try to resolve periodicities independently within this record. I would leave this analysis out.

Lines 355 – 350: are all these paragraphs part of one argument? If so combine.

Line 356: Si/Ca fraction. With the preparation methods and uptake by acetic acid digestion, can the authors please clarify which phase / sedimentary component they think the Si is coming from? Is it likely dominated by biogenic Si?

Lines 360 – 365: I think I get what you're trying to say, but this could be expressed more clearly.

Lines 435 – 437 what has MIS12a got to do with MIS 2 and 6? Not clear what your point is here.

DISCUSSION:

In general, the discussion feels long and could have more focus. It feels like you are discussing every aspect of the record from a descriptive perspective, rather than focusing on what the data tell you about processes. I've got specific comments below, but I would focus on the broader longer-term trends and behaviour of the coccolithophore productivity and export system during times of distinct oceanographic conditions (i.e. between the stages) rather than the millennial scale lead and lags (dubious as to how robust these are). It would be great to use these different intervals to try frame clearly articulated conceptual models about how and why growth rate is coupled or decoupled from organic and inorganic carbon fluxes at different stages. Such conceptual models would have the potential to genuinely inform the thinking of biogeochemical modellers by providing clear patterns of change that should be reproducible by numerical biogeochemical models of these systems. But I think you need to clearly formulate these, in words and ideally schematics, in order for them to take notice of your data. This is also where the multi-proxy approach you take is a clear ADVANTAGE, it's not about "mismatches" in the data, it's about using multi-proxy data to represent the responses of different components of the primary production to export system.

Section 5.2 – looking at the records, I think it's a matter of scale at which they are interrogated. Yes, they are subtly different, but they also preserve some of the same features with reasonable fidelity – for example there are broad trends from MIS 12 to 11, to 10, to 9 that are conserved between the proxies. I would consider taking off the "Mismatch" from your title to this section, to give you space to consider both the agreements and the divergence. This would be more helpful in the communication of the key findings of the study – point out the agreements first and then suggest the mismatches.

Following on from this Line 474 – alkenone and n-alkane fluxes mostly reflect conditions of increased export / preservation of organics. Maybe, but I'm not sure that this can, or should, then be decoupled from "rather than coccolithophore growth". . . in line

475. The first order coupling with some features of the coccolith accumulation rates (NAR) and the alkenones, and also your Sr/Ca measure – e.g. the transition from MIS 12 to 11 – would suggest a coupled system change, and this could be the case if increase coccolithophore production was part of the driver of increased general MAR and organic export and accumulation rates? I just wouldn't be so quick to decouple these components.

Lines 477 – 485: leads and lags of the alkenone versus Sr/Ca records of <1000 ka – is this getting down to the resolution of sampling uncertainty / offset? Were the records based on the same sample set?

Figure 8 – useful to have the Uk37 temperatures on this figure as well for reference.

Line 498 – "the NAR reveals large amplitude shifts during times of high coccolithophore productivity" – this seems to be missing the point. The Sr/Ca also shows large amplitude shifts within this period, arguably larger than the NAR. The point is that the NAR steps down substantially form MIS 11b to 11a, whereas Sr/Ca remains high (as you say in preceding sentence). This to me is the interesting system change, and there's a reverse trend in alkenone MAR (gentle rise in values into MIS11a). Could this be a seasonality thing between Sr/Ca and NAR? I.e. growing faster (higher Sr/Ca) but for a shorter growth season (less coccoliths?)? Seasonality could (maybe?) also be coupled with more efficient export and preservation of organics (alkenone MARs)? Ah, yes you come to this in Section 5.2.3. But, (see comment below), I think you could condense, simplify and make your interpretation of the key points more clearly in the discussion; including integrating 5.2.3. with these discussions of the data.

Lines 505 – 509: again the question would be about sampling uncertainty between the records – can you confirm that these leads / lags are meaningful on the sub-millenial scale? I'm just pushing back, because, as a reader, they do not convey a strong argument about process or feature of the data that I would be confident in. This feeds into a general point about the discussion – I think this could be edited down quite

considerably, so that your key points are more clearly and forcefully stated (and more easily digested by the reader!). This cross-spectral analysis doesn't add anything for me.

---

## Referee Comment (RC2) · Anonymous Referee #2 · 5 Jan 2020

The paper by Catarina Cavaleiro and collaborators entitled 'Coccolithophore productivity at the western Iberian Margin during the middle Pleistocene (310 – 455 ka) – evidence from coccolith Sr/Ca data' examines the geochemical response (coccolith Sr/Ca elemental data) across the MIS12 – MIS9 time slice offshore Portugal.
Based on published coccolithophorid culture finding, the Authors use the abundance of strontium relative to calcite in fossil coccoliths measure by ICP-AES to derive a palaeoproductivity index during the rapid climatic oscillations of the Pleistocene. The region of interest typified by the Portugal Current System was previously documented

in terms of changes in the courantology, sea surface temperatures (among other key climate-sensitive data) in a bunch of publications (cited in the paper). The authors used this well-established framework to interpret fluctuations in Sr/Ca ratios and productivity in the sunlit waters. They also discuss their data at the level of the phytoplanktonic ecosystem as they argue that coccolithophorid growth (and productivity) is dictated by macro and micronutrient availability and the competition with diatoms. They mainly focus their biogeochemical discussion on MIS 12 10 showing higher productivity at the beginning of these climate transitions. Playing at different timescales, they ultimately compare their coccolithophorid productivity indices to the available *i)* alkenone fluxes and *ii)* nannofossils accumulation rates in published literature and found some coherencies and discrepancies.

I am generally supportive of publication of this work in Climate of the Past. I have, however, a number of comments and questions, which I hope the Authors will find fair and useful to prepare their revisions.

General comments

- It would be good to state what was measured exactly. 'Coccolith fraction' is not sufficient as the less that 20 micron filtrate may contain many non-coccolith particles. Some photos will be welcome from key samples to illustrate this.
- There was this nice paper by Omta *et al.)* that came out a few years back (On the potential role of marine calcifiers in glacial-interglacial dynamics - doi:10.1002/gbc.20060) in which an elegant model linking ocean alkalinity and the flourishment of coccolithophores at the inception of deglacial periods (with a possible role on the deglaciation). This paper has been omitted in the present study. I urge the Authors to explore such a control on their productivity data. Even if the periods are not necessarily the same, another useful related paper is that by Duchamp-Alphonse developing the carbonate counter-pump aspects (Enhanced ocean-atmosphere carbon partitioning via the carbonate counter pump during the last deglacial – doi:10.1038/s41467-018-

04625-7). What I am trying to say is that the Authors did a pretty good job in integrating local and regional data but quantitatively understanding pelagic calcification requires a bigger biogeochemical picture.

- Sentence line 283 '*We would like to stress that our study focuses on the qualitative characteristics of the coccolithophore paleoproductivity record, rather than quantitatively estimating the productivity of coccolithophores.*' is misleading and made me doubt about my understanding of the paper. If the Authors interpret Sr/Ca ratios, they intrinsically develop a quantitative approach pertaining productivity in the surface waters.

- Removing the temperature effect from Sr/Ca data to derive productivity component only. I am still debating with myself to be honest. When I read the paper for the first time, I found that it was a good idea. But the more I think, the more I believe that this is not. Both calcification rates and temperatures (and the control of the latter on the former) synergistically dictate Sr/Ca coccolith ratios. Thus dissecting the proxy may induce an artificial bias. I leave these thoughts to the Authors for their revisions. . .

- Emerging from the previous point, the heart of the Sr/Ca productivity proxy is poorly approached in this paper. The Authors mix cellular growth rate, primary productivity, and calcification rates. This is only calcification rates that control the substitution of Sr to Ca. Yet, culture data are unable to properly measure calcification rates, as they only document the bulk over the course of the batch experiments (See the Appendices in Stoll *et al.* 'Climate proxies from Sr/Ca of coccolith calcite: Calibrations from continuous culture of *Emiliania huxleyi*' published in 2002 in GCA). Thus, the generalisation of the proxy to productivity is far-fetched, as it implicitly means primary productivity in turn leading to the strength of the biological pump. I think that the Authors should clarify this.

- The Authors spent considerable effort (and space in the manuscript) to try and find a good match between their coccolithophore productivity and the sedimentation of *Point 1* coccolith-derived calcite (NAR) on one hand, and *Point 2* coccolith-derived compound-specific organic matter (alkenones) on the other hand. *Point 1* For the

reasons outlined above, the Sr/Ca has not to scale with the bulk production (-ity) of calcite. This geochemical proxy has to do with intracellular processes why the production of calcite is also related ecologically with the density of cells in seawater and cellular division rates. *Point 2* We know that alkenones are not only synthesised by the coccolithophores but also by other non-calcifying haptophytes (incl. naked coccolithophores). Furthermore, the export of calcite and organic matter from the top of the water column down to the seafloor obey to different processes (as their on the seafloor and during sedimentary burial diagenesis).

Therefore, I cannot see why all these parameters should scale. I am not aware of any sedimentary succession in which this is the case. I am happy to be wrong though.

- I personally disagree with the fact the Si and Fe concentrations relative to Ca are meaningful in such a sedimentary study nor that they reflect the palaeoconcentrations of these elements. Si and Fe are very tricky to measure and it is unlikely that the measurements reflect the composition of coccolith calcite. Even if it was the case, by which means (proxy) the coccolith Si/Ca ratios would reflect the concentration of silicic acid in ambient waters?

- The Authors have managed to lose me with the concept of phenology they are trying to introduce. This is a black box concept and this is very misleading or at least not clear at all. Could they elaborate?

- I found the statistics very poorly treated in the manuscript.

Specific comments

Pg 1 Line 16. Perhaps use Carbonate Counter-Pump instead?
Pg 1 Line 30. This what?
Pg 1 Line 33. Not clear to me.
Pg 2 Line 40. Circumvoluted sentence. Consider splitting it.
Pg 2 Line 59. I disagree with this statement (see General points).
Pg 3 Line 68. I wonder whether the changes in size of gephyrocapsid coccoliths could

influence the Sr/Ca ratio

Pg 4 Line 97. Poorly defined in terms of what?

Pg 5 Line 122. Minimum numbers. Do you mean absolute or relative abundances?

Pg 6 Line 156. The less than 20 micron fraction contain non coccolith particles. The Authors should do a better job in the characterization of the calcite / dolomite particles analysed. This is crucial.

Section 3.4. I don't understand what is the relevance of this.

Section 4.1. belongs to the discussion. Section 4.2 should come first noting that the description if the results is extremality skinny.

Figure 4. Please make the ages more legible.

Figure 5. What is the significance of the anti-correlation between Mg and Sr?

Figure 6 is unnecessary in my opinion.

Pg 14 Line 303. See my general comment on temperature and productivity on Sr/Ca ratios.

Pg 16 Line 333. What do you refer to with 'opportunistic and fast growing species' here?

Pg 16 Line 355. Methodologically unjustified even using 'weak' acid.

Pg 17 Line 361. Sentence not clear and too long.

Pg 17 Line 371. Decrease of the SST.

Pg 18 Line 421. I am not following the logic here. Are the Authors trying to say that the ice coverage reached the studied area?

Pg 19 Line 449. I don't understand the point that the Authors are trying to make here.

Pg 19 Line 454. Visual comparison of what?

Pg 19 Line 457. An illustration of the poor statistical approach here. . .

Pg 19 Line 457. An illustration of the poor statistical approach here. . .

Pg 21 Lines 477- 492 and figure 9 are not necessary.

---

## Author Comment (AC1) · 11 Feb 2020

*We greatly appreciate the referee's effort and comments and truly believe they will improve the quality of the paper. Please find our answers below (italicized).*

This is a very nice piece of work, presenting new high-resolution coccolith Sr/Ca records from the mid Pleistocene of the Iberian Margin. Coccolith Sr/Ca is an under-used but potentially powerful proxy for understanding the cellular growth and calcification rates of the dominant marine calcifying phytoplankton, the coccolithophore algae.

[Figure]

Of particular value in this manuscript, is the integration of these records of growth rates, with other records of export flux to sediments of both the organic (alkenones) and inorganic (coccolith) carbon fixed by these phytoplankton. Together these allow a nuanced interpretation – including the coupling / decoupling - of surface ocean growth conditions and aggregated net export. I would recommend publication with revisions. In particular I would recommend shortening the manuscript and focusing on the strongest signals within the data in order to generate the impact that this work deserves.
*We agree with the referee and we will shorten and re-structure the manuscript accordingly.*

Comments (in order of the text): Line 17 – instead of "climate models" better to say "Earth System Models" as the common understanding of a "climate model" is one that doesn't include biogeochemistry. And again at Line 48 (and anywhere else) – "climate models" including coccolithophore productivity – better to talk about Earth System and/or biogeochemical and/or carbon cycle models.
*This will be changed accordingly.*

Line 18 – "coccolithophore paleoproductivity past reconstructions" doesn't make sense
*It will be changed to "coccolithophore paleoproductivity reconstructions".*

Line 23 – define SST at first usage
*This will be changed accordingly.*

Lines 38-42 long sentence that jams together two concepts – split.
*We agree with the referee this will be changed to: "They are the most important unicellular primary producer producing calcite (Brand, 1994) contributing up to 60 % to the total oceanic calcium carbonate (Flores and Sierro, 2007) and sensitive to rapid fluctuations in temperature, salinity, nutrients, and turbidity of surface waters (Baumann et al., 2005; McIntyre and Bé, 1967). Coccolithophores had a peak contribution of >80 % in the interval of Marine Isotope Stage (MIS) 15 to MIS 9, when the assemblages were by far dominated by gephyrocapsids (Baumann and Freitag, 2004; Saavedra-Pellitero*

*et al., 2017)."*

INTRODUCTION In both the introduction and the discussion, I feel the absence of a clearly articulated question – what is the "knowledge gap" and how does this paper address that gap? There are suggestions of problems in the representation of coccol- ithophore production and export in biogeochemical models, but no sense of what the specifics of these are, or how they might be addressed by this study. When I first read this section, I was not convinced that models could be informed by new coccolith Sr/Ca records (but see below). Then there is the time period studied – there is a general overview of the significant changes going on through this interval, but the rationale for looking at the coccolithophore response is so broad that it loses meaning: "to evaluate this phytoplankton group's behaviour and gain a better understanding of its response to climate conditions during glacials, interglacials, deglaciations and the transition from interglacial to glacial conditions, at both orbital and sub-orbital time scales." (line 85 on). Or: "We aim to characterise long-term changes in coccolithophore productivity in such a system, where their behaviour in the past remains unknown." (line 74) or: "...and evaluate the main factors influencing coccolithophore productivity." (line 77) I would really like specifics of: 1) the dynamics / processes that you seek to investigate and 2) why these intervals.
*We agree with the referee's comments and we will further clarify and narrow the main purpose and goal of this research. We have chosen the Iberian margin because this area is subjected to seasonal upwelling but particularly sensitive to climate change. The behavior and long-term response of coccolithophores in an area subjected to seasonal upwelling remains unknown. Therefore, we aim to characterize coccol- ithophore's response in an upwelling area subjected to significant climatic changes, such as glacials, interglacials and the transitions between interglacial and glacial sub- stages.*

I think part of your struggle is related to: 1) setting up the Sr/Ca as something that is a "better" measure of coccolithophore "productivity" than other approaches – e.g.

NAR/alkenone accumulation; and 2) equating coccolith Sr/Ca with quite a loose concept of "productivity". Together these end up setting off your "productivity" records "against" one another, rather than being mutually informative about different components of the growth – export – accumulation system. This seems to lead to a discussion which is phrased in terms of "mismatch" rather than one that allows the complexity of the system response to be seen, because, you've got the advantage of multi-proxy data covering different aspects of the same system.

*We will narrow the purpose and aim of the paper and re-structure the manuscript focusing on the processes and on the advantages of the multiproxy approach.*

My recommendation is that you recast this introduction somewhere along these lines: 1) There are significant uncertainties about the complex interactions between coccolithophore growth rates, nutrient dynamics, seasonality, export (carbonate / organic carbon), dissolution and final accumulation / burial rates. These uncertainties make modelling the responses of this system to modern environmental change problematic. 2) These questions can only be addressed with: a) records that test the dynamic response of various components of the system over a reasonable range of change (i.e. palaeo records); and b) through multi-proxy studies of growth environment (Sr/Ca) and export of both organic (alkenones) and inorganic (NAR) carbon. 3) Then make the argument for the particular time period studied providing the chance to test a range of particular environmental conditions – and make your introduction to the time period outline what these might be – e.g. upwelling, seasonality, temperature... With this set-up, hopefully you'll then be able to circle round in the discussion and answer these questions.

*We greatly appreciate the referee's suggestions and we intend to include them in the introduction.*

Sr/Ca VARIATIONS AND ASSEMBLAGE CHANGES Line 68 – 69: I'd like some more justification for the assumption that "assemblage changes don't matter". My reading of the Fink et al. 2010 paper was that the abundance of the larger *Calcidiscus lep-*

*toporus* did have a significant impact on CF Sr/Ca. I'm also suspicious of using the logic that in some instances in the modern oceans CF Sr/Ca changes coincide with productivity changes, therefore, it must be productivity, when coincident assemblage changes haven't been properly considered. Given expertise of Baumann and Stoll, I would like to see some more justification of this point, that CF Sr/Ca are really dominated by changes in growth rate, rather than assemblages, especially with respect to *Calcidiscus*, which I suspect can contribute strongly to some records that show large Sr/Ca variations.

*We thank the referee's comment and we will make this clearer in the revised manuscript. In summary, in our research, given the vast majority of gephyrocapsids (97 % average, 60 % of* Gephyrocapsa caribbeanica*) and very low abundance of* Calcidiscus leptoporus *and* Helicosphaera *sp., it seems unlikely that changes in the range of their average relative or absolute abundance would have a significant effect on the coccolith fraction Sr/Ca ratio.*

Later – Line 179 – you talk about being in the Gephyrocapsa acme and that this makes assemblage variability less of an issue for CF Sr/Ca, but this somewhat admits that assemblage change can be an issue under other circumstances. A clear delineation – even without hard and fast data / rules – between when assemblage changes are likely and not-likely a problem with CF Sr/Ca would be better than trying to imply there is no issue. Please be precise with the logic and transparent with the reader as to if and where assemblage change might be playing a significant role.

*The referee is correct when stating that indeed changes in the coccolith assemblage might bias the interpretation of the coccolith fraction Sr/Ca proxy. However, we believe that to delineate a threshold above which changes in coccolith assemblage would significantly bias the coccolith fraction Sr/Ca ratio is out of the scope of this research. As mentioned previously, in our research it is very unlikely that the changes in* Calcidiscus leptoporus *abundance could bias the CF Sr/Ca results, given the Gephyrocapsids dominance. Plus, statistically, we find that there is no significant relationship between the relative abundance of* Calcidiscus *and our coccolith fraction Sr/Ca ratio or coccol-*

[Figure]

*ithophore productivity proxy (see correlation charts and Pearson's correlation results below – Figure 1 and Table 1).*

Line 184 – gephyrocapsids's – lower case I believe.
*Correct, this will be changed accordingly.*

Lines 255- bias from other carbonate phases – from Figure 5 it looks like the trend starts before the cut-off used, from more like 30 mmol/mol Mg/Ca. And could you please clarify which "cut off" you are using, whether this is Sr/Ca below 1.8 mmol/mol or higher values of Mg/Ca? If tracing contamination from other carbonates, would it make more sense to use the Mg/Ca values for the cut-off? For example you could cut off more stringently, at 30 mmol/mol Mg/Ca and yet maintain what look like more robust / primary signals of lower Sr/Ca within data that would pass this criteria. You would lose a few more data points in total, but I think this would be a more defensible cut-off point and rationale.
*We aim to clarify this in the revised version of the manuscript. By combining the data from CF Sr/Ca and CF Mg/Ca ratios we defined a threshold of 1.8 mmol/mol because until such level most of the samples showed quite low CF Mg/Ca results (on the same range of the major cluster of the data). However, as suggested by the reviewer, we will consider changing the CF Mg/Ca ratio from 50 mmol/mol to 30 mmol/mol, to more clearly highlight the samples where some biasing by other carbonate phases might have existed.*

Line – 272 – "This interval represents 39% of the whole sampling variation. . .", doesn't make sense to me, please clarify. Do you mean something like the clipped data represents 39% of the dynamic range in Sr/Ca of the full sample set?
*We agree with the referee and the wording will be changed in the revised manuscript to make sure that this information is clearly given to the reader. We will substitute that paragraph by the following one: "The coccolith fraction Sr/Ca ratio results varied between 1.2 and 2.4 mmol/mol (sampling range of 1.16 mmol/mol) but 85 % of the samples returned results between 1.8 and 2.3 mmol/mol. These 85 % of the sam-*

*ples only represent 39 % of the total sampling range." This means that only 15 % of the samples fall on 70% of the sampling range. And, most of them coincide with the samples with higher bias likelihood from other carbonate phases.*

Lines 288-289: is significance level of 80% really enough to be confident that the 6ka peak is real? I'm not really convinced. I don't think this spectral analysis reveals anything and is a distraction for the reader - you have a nice tight coupling to well-resolved climate records (Uk37) and a good age model for making 104-year correlations to N. Atlantic climate records, so I don't think there's a need to try to resolve periodicities independently within this record. I would leave this analysis out.
*We agree that we should focus the purpose of this research in long-term processes and not so much on shorter scale changes. Hence, we agree that this spectral analysis might deviate the reader from the most important aspects of our research. Therefore, it is very likely that this analysis will be left out of the revised manuscript after the re-structuring and re-focusing of the paper.*

Lines 355 – 350: are all these paragraphs part of one argument? If so combine.
*All paragraphs from current line 335 to line 350 will be one paragraph only, as suggested by the referee.*

Line 356: Si/Ca fraction. With the preparation methods and uptake by acetic acid digestion, can the authors please clarify which phase / sedimentary component they think the Si is coming from? Is it likely dominated by biogenic Si?
*We thank the referee for this comment and we will look carefully and clarify on the revised manuscript or delete the assumption that higher coccolith fraction Si/Ca and Fe/Ca could evidence higher competition with diatoms.*

Lines 360 – 365: I think I get what you're trying to say, but this could be expressed more clearly.
*We thank the referee's comment and we will make this clearer in the revised manuscript.*

Lines 435 – 437: what has MIS12a got to do with MIS 2 and 6? Not clear what your point is here.

*We thank the referee's comment and the reference to MIS 2 and 6 will be either deleted or clarified in the revised manuscript.*

DISCUSSION: In general, the discussion feels long and could have more focus. It feels like you are discussing every aspect of the record from a descriptive perspective, rather than focusing on what the data tell you about processes. I've got specific comments below, but I would focus on the broader longer-term trends and behaviour of the coccolithophore productivity and export system during times of distinct oceanographic conditions (i.e. between the stages) rather than the millennial scale lead and lags (dubious as to how robust these are). It would be great to use these different intervals to try frame clearly articulated conceptual models about how and why growth rate is coupled or decoupled from organic and inorganic carbon fluxes at different stages. Such conceptual models would have the potential to genuinely inform the thinking of biogeochemical modellers by providing clear patterns of change that should be reproducible by numerical biogeochemical models of these systems. But I think you need to clearly formulate these, in words and ideally schematics, in order for them to take notice of your data. This is also where the multi-proxy approach you take is a clear ADVANTAGE, it's not about "mismatches" in the data, it's about using multi-proxy data to represent the responses of different components of the primary production to export system. Section 5.2 – looking at the records, I think it's a matter of scale at which they are interrogated. Yes, they are subtly different, but they also preserve some of the same features with reasonable fidelity – for example there are broad trends from MIS 12 to 11, to 10, to 9 that are conserved between the proxies. I would consider taking off the "Mismatch" from your title to this section, to give you space to consider both the agreements and the divergence. This would be more helpful in the communication of the key findings of the study – point out the agreements first and then suggest the mismatches. Following on from this Line 474 – alkenone and n-alkane fluxes mostly reflect conditions of increased export / preservation of organics. Maybe, but I'm not sure that

this can, or should, then be decoupled from "rather than coccolithophore growth"...in line 475. The first order coupling with some features of the coccolith accumulation rates (NAR) and the alkenones, and also your Sr/Ca measure – e.g. the transition from MIS 12 to 11 – would suggest a coupled system change, and this could be the case if increase coccolithophore production was part of the driver of increased general MAR and organic export and accumulation rates? I just wouldn't be so quick to decouple these components. Lines 477 – 485: leads and lags of the alkenone versus Sr/Ca records of <1000 ka – is this getting down to the resolution of sampling uncertainty / offset? Were the records based on the same sample set?

*We thank the referee's comment and confirm that the samples where both CF Sr/Ca ration and alkenone content and accumulation (and nannofossil accumulation rate) were measured are from the same set of samples but the alkenone content and consequent sea surface temperature estimation had a higher sampling resolution.*

Figure 8 – useful to have the Uk37 temperatures on this figure as well for reference. *This will be changed accordingly.*

Line 498 – "the NAR reveals large amplitude shifts during times of high coccolithophore productivity" – this seems to be missing the point. The Sr/Ca also shows large amplitude shifts within this period, arguably larger than the NAR. The point is that the NAR steps down substantially form MIS 11b to 11a, whereas Sr/Ca remains high (as you say in preceding sentence). This to me is the interesting system change, and there's a reverse trend in alkenone MAR (gentle rise in values into MIS11a). Could this be a seasonality thing between Sr/Ca and NAR? I.e. growing faster (higher Sr/Ca) but for a shorter growth season (less coccoliths?)? Seasonality could (maybe?) also be coupled with more efficient export and preservation of organics (alkenone MARs)? Ah, yes you come to this in Section 5.2.3. But, (see comment below), I think you could condense, simplify and make your interpretation of the key points more clearly in the discussion; including integrating 5.2.3. with these discussions of the data. Lines 505 – 509: again the question would be about sampling uncertainty between the records –

can you confirm that these leads / lags are meaningful on the sub-millenial scale? I'm just pushing back, because, as a reader, they do not convey a strong argument about process or feature of the data that I would be confident in. This feeds into a general point about the discussion – I think this could be edited down quite considerably, so that your key points are more clearly and forcefully stated (and more easily digested by the reader!). This cross-spectral analysis doesn't add anything for me.

*We greatly appreciate the referee's comments on the Discussion and, accordingly, we will shorten and re-structure it, focusing on processes and on the advantages of the multiproxy approach.*

[Figure]

*Figure 1*          *Figure 2*

**Fig. 1.** Figures 1 and 2 showing cross-plots of Calcidiscus leptoporus with coccolithophore productivity proxy (Fig. 1) and the coccolith fraction (CF) Sr/Ca ratio (mmol/mol).

| Pearson's Correlation coef. | R | p-value | H0 (5%) |
|---|---|---|---|
| CF Sr/Ca ratio (mmol/mol) vs *C. leptoporus* (%) | -0.14 | 0.07 | accepted |
| Coccolithophore productivity proxy vs *C. leptoporus* (%) | -0.09 | 0.21 | accepted |

**Fig. 2.** Table 1 shows the Pearson's correlation analysis of Calcidiscus leptoporus abundance and both CF Sr/Ca ratios (mmol/mol) and coccolithophore productivity reconstruction

---

## Author Comment (AC2) · 11 Feb 2020

*We appreciate the referee's effort and comments and believe they will improve the quality of the paper. Please find our answers below (italicized).*

The paper by Catarina Cavaleiro and collaborators entitled 'Coccolithophore productivity at the western Iberian Margin during the middle Pleistocene (310 – 455 ka) – evidence from coccolith Sr/Ca data' examines the geochemical response (coccolith Sr/Ca elemental data) across the MIS12 – MIS9 time slice offshore Portugal. Based on pub-

[Figure]

lished coccolithophorid culture finding, the Authors use the abundance of strontium relative to calcite in fossil coccoliths measure by ICP-AES to derive a palaeoproductivity index during the rapid climatic oscillations of the Pleistocene. The region of interest typified by the Portugal Current System was previously documented in terms of changes in the courantology, sea surface temperatures (among other key climate-sensitive data) in a bunch of publications (cited in the paper). The authors used this well-established framework to interpret fluctuations in Sr/Ca ratios and productivity in the sunlit waters. They also discuss their data at the level of the phytoplanktonic ecosystem as they argue that coccolithophorid growth (and productivity) is dictated by macro and micronutrient availability and the competition with diatoms. They mainly focus their biogeochemical discussion on MIS 12 10 showing higher productivity at the beginning of these climate transitions. Playing at different timescales, they ultimately compare their coccolithophorid productivity indices to the available i) alkenone fluxes and ii) nannofossils accumulation rates in published literature and found some coherencies and discrepancies. I am generally supportive of publication of this work in Climate of the Past. I have, however, a number of comments and questions, which I hope the Authors will find fair and useful to prepare their revisions.

General comments - It would be good to state what was measured exactly. 'Coccolith fraction' is not sufficient as the less that 20 micron filtrate may contain many non-coccolith particles. Some photos will be welcome from key samples to illustrate this.
*We thank the referee's comment and we will further clarify the composition of the "coccolith fraction" in the revised manuscript because indeed, it may contain non-coccolith particles. However, as explained in the methods section, all samples were treated to avoid Sr contamination from non-carbonate particles (assuming that foraminifera and foraminifera fragments were extracted during sieving). Plus, the only existing picture (see below) of site MD03-2699 is from sample 1898 (referring to the depth in the core – 1898 cm) with a corresponding age of 485 kyr (not covered in our research). Note that this photo was taken during a master class exercise and consequently it tried to gather as many different coccoliths in picture as possible.*

- There was this nice paper by Omta et al.) that came out a few years back (On the potential role of marine calcifiers in glacial-interglacial dynamics - doi:10.1002/gbc.20060) in which an elegant model linking ocean alkalinity and the flourishment of coccolithophores at the inception of deglacial periods (with a possible role on the deglaciation). This paper has been omitted in the present study. I urge the Authors to explore such a control on their productivity data. Even if the periods are not necessarily the same, another useful related paper is that by Duchamp-Alphonse developing the carbonate counter-pump aspects (Enhanced ocean-atmosphere carbon partitioning via the carbonate counter pump during the last deglacial – doi:10.1038/s41467-018- 04625-7). What I am trying to say is that the Authors did a pretty good job in integrating local and regional data but quantitatively understanding pelagic calcification requires a bigger biogeochemical picture.

*We thank the referee for mentioning these papers, we will include them and take them into account, with an approach centered in the processes, also in the line of referee #1's comment.*

- Sentence line 283 'We would like to stress that our study focuses on the qualitative characteristics of the coccolithophore paleoproductivity record, rather than quantitatively estimating the productivity of coccolithophores.' is misleading and made me doubt about my understanding of the paper. If the Authors interpret Sr/Ca ratios, they intrinsically develop a quantitative approach pertaining productivity in the surface waters.

*We thank the referee's comment and we would like to stress that this sentence was included in the manuscript to clarify the reader that the CF Sr/Ca ratio is not an absolute productivity proxy neither does it allow for the calculation of absolute marine productivity in terms of production of organic carbon or calcium carbonate by coccolithophores.*

- Removing the temperature effect from Sr/Ca data to derive productivity component only. I am still debating with myself to be honest. When I read the paper for the first time, I found that it was a good idea. But the more I think, the more I believe that this

is not. Both calcification rates and temperatures (and the control of the latter on the former) synergistically dictate Sr/Ca coccolith ratios. Thus dissecting the proxy may induce an artificial bias. I leave these thoughts to the Authors for their revisions. . . - Emerging from the previous point, the heart of the Sr/Ca productivity proxy is poorly approached in this paper. The Authors mix cellular growth rate, primary productivity, and calcification rates. This is only calcification rates that control the substitution of Sr to Ca. Yet, culture data are unable to properly measure calcification rates, as they only document the bulk over the course of the batch experiments (See the Appendices in Stoll et al. 'Climate proxies from Sr/Ca of coccolith calcite: Calibrations from continuous culture of Emiliania huxleyi' published in 2002 in GCA). Thus, the generalisation of the proxy to productivity is far-fetched, as it implicitly means primary productivity in turn leading to the strength of the biological pump. I think that the Authors should clarify this.

*We thank the referee's comment and we will further clarify these points in the revised manuscript. We further acknowledge the extent work already done in correlating coccolith Sr/Ca ratio with coccolithophore productivity. Stoll and Schrag, 2000 initially suggested that the CF Sr/Ca ratios are strongly controlled by coccolithophorid growth and calcification rate. Stoll et al., 2002a (Potencial and limitations of proxy) and Stoll et al., 2002b (E. huxleyi cultures), Stoll et al., 2002c (multi species cultures), Stoll et al., 2007a (Arabian and Sargassum seas) and Stoll et al., 2007b (bay of Bengal) used culture records, sediment traps and sediment samples to confirm the relationship between coccolith Sr/Ca ratios and coccolithophore productivity (coccolithophore growth rate and coccosphere export). Furthermore, Stoll et al., 2002a and Mejia et al., 2013 clearly stated that the temperature effect on the CF Sr/Ca must be addressed when reconstructing past coccolithophore productivity. Indeed, in our research the extraction of the temperature effect does not represent a major change of the original curve. However, Cavaleiro et al., 2018, show a final coccolithophore productivity record notably different from the original coccolith fraction Sr/Ca curve due to the large influence of temperature in that area. Our temperature correction in the Iberian margin site re-*

*inforces that, contrary to the open ocean mid North Atlantic, the temperature changes in the Iberian margin do not seem to have had affected the coccolith fraction Sr/Ca and consequent coccolithophore productivity. Plus, the possibility to use a proxy that is independent of accumulation rates allows comparison with commonly used "coccolithophore productivity proxies" such as nannofossil accumulation ratios and alkenone export from which coccolithophore productivity, in the ancient photic layer, is commonly inferred from. Finally, the term productivity is thus in this research used as a coccolithophore productivity proxy directly associated with coccolith calcification rate and generally associated with increased cell division and growth of coccolithophores that could lead to increased particulate organic matter and calcium carbonate export.*

- The Authors spent considerable effort (and space in the manuscript) to try and find a good match between their coccolithophore productivity and the sedimentation of Point 1 coccolith-derived calcite (NAR) on one hand, and Point 2 coccolith-derived compound-specific organic matter (alkenones) on the other hand. Point 1 For the reasons outlined above, the Sr/Ca has not to scale with the bulk production (-ity) of calcite. This geochemical proxy has to do with intracellular processes why the production of calcite is also related ecologically with the density of cells in seawater and cellular division rates. Point 2 We know that alkenones are not only synthesized by the coccolithophores but also by other non-calcifying haptophytes (incl. naked coccolithophores). Furthermore, the export of calcite and organic matter from the top of the water column down to the seafloor obey to different processes (as their on the seafloor and during sedimentary burial diagenesis). Therefore, I cannot see why all these parameters should scale. I am not aware of any sedimentary succession in which this is the case. I am happy to be wrong though.

*We thank the referee's comment and we hope to clarify these doubts, also in line with some of referee #1's comments, by re-structuring the paper focusing on processes and on the advantages of the multiproxy approach.*

- I personally disagree with the fact the Si and Fe concentrations relative to Ca are

meaningful in such a sedimentary study nor that they reflect the palaeoconcentrations of these elements. Si and Fe are very tricky to measure and it is unlikely that the measurements reflect the composition of coccolith calcite. Even if it was the case, by which means (proxy) the coccolith Si/Ca ratios would reflect the concentration of silicic acid in ambient waters?

*We greatly appreciate the referee's comment and based on comments of both reviewers we will verify our assumptions and either clarify or delete them from the revised manuscript.*

- The Authors have managed to lose me with the concept of phenology they are trying to introduce. This is a black box concept and this is very misleading or at least not clear at all. Could they elaborate?

*We thank the referee's comment and we will further explain the definition of phenology and why changes in climate could represent changes in the coccolithophore phenology, i.e., how warmer or colder conditions (and consequent climatic changes) could actually lead to changes in the productive regime of coccolithophores throughout the year.*

- I found the statistics very poorly treated in the manuscript.

*We thank the referee's comment and, if referring to the spectral analysis, in line with the comments from referee #,1 we will decide whether to keep or delete the spectral and cross-spectral analysis.*

Specific comments Pg 1 Line 16. Perhaps use Carbonate Counter-Pump instead?
*This will be changed accordingly.*

Pg 1 Line 30. This what?
*"This" refers to the fact that more nutrient-rich waters decreased the competition with diatoms for nutrients. It will be written more clearly in the revised version of the manuscript.*

Pg 1 Line 33. Not clear to me.
*We thank the referee's comment and this will be addressed in the body of the paper*

*to clarify why changes in climate could represent changes in the coccolithophore phenology, i.e., how warmer or colder conditions (and consequent climatic changes) could actually lead to changes in the productive regime of coccolithophores throughout the year.*

Pg 2 Line 40. Circumvoluted sentence. Consider splitting it.
*We agree with the referee's comment, also in line with referee 1's comment, this will be changed to: "They are the most important unicellular primary producer producing calcite (Brand, 1994) contributing up to 60 % to the total oceanic calcium carbonate (Flores and Sierro, 2007) and sensitive to rapid fluctuations in temperature, salinity, nutrients, and turbidity of surface waters (Baumann et al., 2005; McIntyre and Bé, 1967). Coccolithophores had a peak contribution of >80 % in the interval of Marine Isotope Stage (MIS) 15 to MIS 9, when the assemblages were by far dominated by gephyrocapsids (Baumann and Freitag, 2004; Saavedra-Pellitero et al., 2017)."*

Pg 2 Line 59. I disagree with this statement (see General points).
*We thank the referee's comment and we will further address and clarify this (see also the reply given above, to the general points' replies).*

Pg 3 Line 68. I wonder whether the changes in size of gephyrocapsid coccoliths could influence the Sr/Ca ratio.
*We thank the referee's comment but believe this is out of the scope of our research since it is not our intention to better understand how coccolith Sr/Ca varies with the size of gephyrocapsa coccoliths.*

Pg 4 Line 97. Poorly defined in terms of what?
*We thank the referee's comment but believe this is out of the scope of our research since it is not our intention to better describe the Portugal current system. We can however substitute "poorly defined due to" by "with".*

Pg 5 Line 122. Minimum numbers. Do you mean absolute or relative abundances?
*We thank the referee's comment and we will address this by clearly stating absolute*

*abundances.*

Pg 6 Line 156. The less than 20 micron fraction contain non coccolith particles. The Authors should do a better job in the characterization of the calcite / dolomite particles analysed. This is crucial.
*We thank the referee's comment and this will be clarified in the revised version of the manuscript.*

Section 3.4. I don't understand what is the relevance of this.
*We agree that this spectral analysis might deviate the reader from the most important aspects of our research. Therefore, it is very likely that this analysis will be left out of the revised manuscript after the re-structuring and re-focusing of the paper.*

Section 4.1. belongs to the discussion. Section 4.2 should come first noting that the description if the results is extremality skinny.
*We thank the referee's comment and given the re-structuring of the paper, it is also likely that this section is moved to the discussion.*

Figure 4. Please make the ages more legible.
*This will be changed accordingly.*

Figure 5. What is the significance of the anti-correlation between Mg and Sr?
*We thank the referee's comment and this information will be added to the manuscript. The p-value of the Pearson correlation is 0, therefore the relationship between either the coccolith fraction and the coccolithophore productivity and the coccolith fraction Mg/Ca is highly significant, as expected. It is the outliers, or higher values of Mg/Ca that are associated with very low CF Sr/Ca ratios and coccolithophore productivity results, mostly associated to abrupt and cold millennial-scale events.*

Figure 6 is unnecessary in my opinion.
*We thank the referee's comment and it is very likely that this spectral analysis will be left out of the revised manuscript after the re-structuring and re-focusing of the paper.*

Pg 14 Line 303. See my general comment on temperature and productivity on Sr/Ca ratios.
*We thank the referee's comment and believe we have already commented this matter above.*

Pg 16 Line 333. What do you refer to with 'opportunistic and fast growing species' here?
*We thank the referee's comment and clarify that by "opportunistic and fast growing species" we refer to species r-selected. This will be re-written and clarified in the revised manuscript.*

Pg 16 Line 355. Methodologically unjustified even using 'weak' acid.
*We thank the referee for this comment and we will look carefully and clarify on the revised manuscript or delete the assumption that higher coccolith fraction Si/Ca and Fe/Ca could evidence higher competition with diatoms.*

Pg 17 Line 361. Sentence not clear and too long.
*We thank the referee for this comment and we will look carefully and clarify on the revised manuscript or delete the assumption that higher coccolith fraction Si/Ca and Fe/Ca could evidence higher competition with diatoms.*

Pg 17 Line 371. Decrease of the SST.
*This will be changed accordingly.*

Pg 18 Line 421. I am not following the logic here. Are the Authors trying to say that the ice coverage reached the studied area?
*We thank the referee for this comment and clarify that we have not stated or suggested that ice coverage reached the Iberian margin. We are stating research (Line 338) that has found evidences of the presence of melting icebergs in the western Iberian margin during rapid millennial-scale events.*

Pg 19 Line 449. I don't understand the point that the Authors are trying to make here.

*We thank the referee's comment and this will be written more clearly in the revised version of the manuscript.*

Pg 19 Line 454. Visual comparison of what?
*We thank the referee's comment and further clarify that the records of coccolithophore productivity, nannofossil accumulation rate and alkenone flux were compared visually.*

Pg 19 Line 457. An illustration of the poor statistical approach here...
*We thank the referee's comment and the statistical analysis will be provided in the supplementary material with the respective p-values.*

Pg 21 Lines 477- 492 and figure 9 are not necessary.
*We agree that this spectral analysis might deviate the reader from the most important aspects of our research. Therefore, it is very likely that this analysis will be left out of the revised manuscript after the re-structuring and re-focusing of the paper.*

References

Mejía, L.M., Ziveri, P., Cagnetti, M., Bolton, C., Zahn, R., Marino, G., Martínez-Méndez, G., Stoll, H., 2014. Effects of midlatitude westerlies on the paleoproductivity at the Agulhas Bank slope during the penultimate glacial cycle: Evidence from coccolith Sr/Ca ratios. Paleoceanography 29, 697–714. https://doi.org/10.1002/2013PA002589

Stoll, H.M., Arevalos, A., Burke, A., Ziveri, P., Mortyn, G., Shimizu, N., Unger, D., 2007b. Seasonal cycles in biogenic production and export in Northern Bay of Bengal sediment traps. Deep Sea Res. Part II Top. Stud. Oceanogr. 54, 558–580. https://doi.org/10.1016/j.dsr2.2007.01.002

Stoll, Heather M., Klaas, C.M., Probert, I., Encinar, J.R., Alonso, J.I.G., 2002c. Calcification rate and temperature effects on Sr partitioning in coccoliths of multiple species of coccolithophorids in culture. Glob. Planet. Change 34, 153–171. https://doi.org/10.1016/S0921-8181(02)00112-1

Stoll, Heather M, Rosenthal, Y., Falkowski, P., 2002b. Climate proxies from Sr/Ca of

coccolith calcite: calibrations from continuous culture of Emiliania huxleyi. Geochim. Cosmochim. Acta 66, 927–936. https://doi.org/https://doi.org/10.1016/S0016-7037(01)00836-5

Stoll, H.M., Schrag, D.P., 2000. Coccolith Sr/Ca as a new indicator of coccolithophorid calcification and growth rate. Geochemistry, Geophys. Geosystems 1. https://doi.org/10.1029/1999GC000015

Stoll, Heather M., Ziveri, P., M., G., Probert, I., Young, J.R., 2002a. Potential and limitations of Sr/Ca ratios in coccolith carbonate: new perspectives from cultures and monospecific samples from sediments. Philos. Trans. R. Soc. A Math. Phys. Eng. Sci. 360, 719–747. https://doi.org/10.1098/rsta.2001.0966

Stoll, H.M., Ziveri, P., Shimizu, N., Conte, M., Theroux, S., 2007a. Relationship between coccolith Sr/Ca ratios and coccolithophore production and export in the Arabian Sea and Sargasso Sea. Deep Sea Res. Part II Top. Stud. Oceanogr. 54, 581–600. https://doi.org/10.1016/j.dsr2.2007.01.003
* * *
[Figure]

**Fig. 1.** Figure 1 – SEM picture from site MD03-2699, at 1898 cm with a corresponding age of 485 kyr. The irregular surface background is the filter and several coccoliths from different species can be seen, na

---

## Author Response (AR1)

Dear Editor Luc Beaufort,

We are submitting the revised version of the manuscript " Coccolithophore productivity at the western Iberian Margin during the middle Pleistocene (310 – 455 ka) – evidence from coccolith Sr/Ca data " [cp-2019-131] in which we address comments of the two reviewers.
We greatly appreciate the reviewers' comments and have incorporated them to improve this latest version. Suggestions by reviewer Tom Dunkley Jones resulted to major changes in the discussion with subchapters 5.2 and 5.3 being mostly re-written or newly added. The spectral analyses results and related text and figures have been omitted from the revised manuscript.

You will find below a detailed reply (in green) to all the points raised along with the specification of the revisions that have been made and arguments that we used in considering the comments.
A tracked-changes version of this manuscript is also at the end of this rebuttal letter.

Once more, thank you very much for your understanding throughout the revision process and under such a challenging pandemic situation.

Best regards, on behalf of all co-authors,
Catarina Cavaleiro

**Response to comments from the editors and reviewers:**
**-Reviewer 1**
This is a very nice piece of work, presenting new high-resolution coccolith Sr/Ca records from the mid Pleistocene of the Iberian Margin. Coccolith Sr/Ca is an underused but potentially powerful proxy for understanding the cellular growth and calcification rates of the dominant marine calcifying phytoplankton, the coccolithophore algae.
Of particular value in this manuscript, is the integration of these records of growth rates, with other records of export flux to sediments of both the organic (alkenones) and inorganic (coccolith) carbon fixed by these phytoplankton. Together these allow a nuanced interpretation – including the coupling / decoupling - of surface ocean growth conditions and aggregated net export. I would recommend publication with revisions. In particular I would recommend shortening the manuscript and focusing on the strongest signals within the data in order to generate the impact that this work deserves.

**Comments (in order of the text):**
Line 17 – instead of "climate models" better to say "Earth System Models" as the common understanding of a "climate model" is one that doesn't include biogeochemistry.
And again at Line 48 (and anywhere else) – "climate models" including coccolithophore productivity – better to talk about Earth System and/or biogeochemical and/or carbon cycle models.
We thank the reviewer for this comment and changed this accordingly.

Line 18 – "coccolithophore paleoproductivity past reconstructions" doesn't make sense
We thank the reviewer for this comment and have changed the previous sentence to "the reconstruction of coccolithophore paleoproductivity mostly relied on proxies dependent on accumulation and sedimentation rates, and preservation conditions."

Line 23 – define SST at first usage
We thank the reviewer for this comment and this was changed accordingly.

Lines 38-42 long sentence that jams together two concepts – split.
We thank the reviewer for this comment and since the introduction was re-structured, this long sentence was deleted.

**INTRODUCTION**

In both the introduction and the discussion, I feel the absence of a clearly articulated question – what is the "knowledge gap" and how does this paper address that gap?

There are suggestions of problems in the representation of coccolithophore production and export in biogeochemical models, but no sense of what the specifics of these are, or how they might be addressed by this study. When I first read this section, I was not convinced that models could be informed by new coccolith Sr/Ca records (but see below). Then there is the time period studied – there is a general overview of the significant changes going on through this interval, but the rationale for looking at the coccolithophore response is so broad that it loses meaning: "to evaluate this phytoplankton group's behaviour and gain a better understanding of its response to climate conditions during glacials, interglacials, deglaciations and the transition from interglacial to glacial conditions, at both orbital and sub-orbital time scales."

(line 85 & on). Or: "We aim to characterise long-term changes in coccolithophore productivity in such a system, where their behaviour in the past remains unknown."

(line 74) or: ". . .and evaluate the main factors influencing coccolithophore productivity."

(line 77) I would really like specifics of: 1) the dynamics / processes that you seek to investigate and 2) why these intervals.

I think part of your struggle is related to: 1) setting up the Sr/Ca as something that is a "better" measure of coccolithophore "productivity" than other approaches – e.g. NAR / alkenone accumulation; and 2) equating coccolith Sr/Ca with quite a loose concept of "productivity". Together these end up setting off your "productivity" records "against" one another, rather than being mutually informative about different components of the growth – export – accumulation system. This seems to lead to a discussion which is phrased in terms of "mismatch" rather than one that allows the complexity of the system response to be seen, because, you've got the advantage of multi-proxy data covering different aspects of the same system.

My recommendation is that you recast this introduction somewhere along these lines:

1) There are significant uncertainties about the complex interactions between coccolithophore growth rates, nutrient dynamics, seasonality, export (carbonate / organic carbon), dissolution and final accumulation / burial rates. These uncertainties make modelling the responses of this system to modern environmental change problematic.

2) These questions can only be addressed with: a) records that test the dynamic response of various components of the system over a reasonable range of change (i.e. palaeo records); and b) through multi-proxy studies of growth environment (Sr/Ca) and export of both organic (alkenones) and inorganic (NAR) carbon.

3) Then make the argument for the particular time period studied providing the chance to test a range of particular environmental conditions – and make your introduction to the time period outline what these might be – e.g. upwelling, seasonality, temperature…

With this set-up, hopefully you'll then be able to circle round in the discussion and answer these questions.

We thank the reviewer for this comment and we have changed the introduction accordingly. Please check the tracked changes document.

**Sr/Ca VARIATIONS AND ASSEMBLAGE CHANGES**

Line 68 – 69: I'd like some more justification for the assumption that "assemblage changes don't matter". My reading of the Fink et al. 2010 paper was that the abundance of the larger *Calcidiscus leptoporus* did have a significant impact on CF Sr/Ca.

I'm also suspicious of using the logic that in some instances in the modern oceans CF Sr/Ca changes coincide with productivity changes, therefore, it must be productivity, when coincident

assemblage changes haven't been properly considered. Given expertise of Baumann and Stoll, I would like to see some more justification of this point, that CF Sr/Ca are really dominated by changes in growth rate, rather than assemblages, especially with respect to Calcidiscus, which I suspect can contribute strongly to some records that show large Sr/Ca variations.

We thank the reviewer for this comment. In our study we focus on the Mid-Brunhes interval exactly because during that time the coccolithophore flora was dominated by *gephyrocapsids* (97 % average, 60 % of *Gephyrocapsa caribbeanica*) with very low abundances of *Calcidiscus leptoporus* and *Helicosphaera* sp.. So contrary to the modern assemblages studied by Fink et al. (2010), it seems unlikely that changes in the range of the latter two's average relative or absolute abundance would have a significant effect on the coccolith fraction Sr/Ca ratio in our study. Please also check the reply to the next comment and to a similar comment from reviewer #2. We also believe that indeed the introduction stated too firmly that changes in the coccolith assemblages do not influence the coccolith fraction Sr/Ca ratio based on the papers from Barker et al., 2006 and Stoll et al., 2002a (see manuscript for references). Fink et al. 2010 present divergent data and we added a sentence to highlight the importance of considering the influence of assemblages' changes on the coccolith fraction Sr/Ca ratio before interpreting it as a paleoproductivity reconstruction proxy. The respective paragraph has now been moved to method subchapter 3.3.

Later – Line 179 – you talk about being in the Gephyrocapsa acme and that this makes assemblage variability less of an issue for CF Sr/Ca, but this somewhat admits that assemblage change can be an issue under other circumstances. A clear delineation – even without hard and fast data / rules – between when assemblage changes are likely and not-likely a problem with CF Sr/Ca would be better than trying to imply there is no issue. Please be precise with the logic and transparent with the reader as to if and where assemblage change might be playing a significant role.

The reviewer is correct when stating that indeed changes in the coccolith assemblage might bias the interpretation of the coccolith fraction Sr/Ca proxy for coccolithophore productivity. However, we believe that to delineate a threshold above which changes in coccolith assemblage would significantly bias the coccolith fraction Sr/Ca ratio is out of the scope of this research. As mentioned previously, in our research it is very unlikely that the changes in *Calcidiscus leptoporus* abundance could bias the CF Sr/Ca results, given the gephyrocapsids dominance. Plus, statistically, we find that there is no significant relationship between the relative abundance of *Calcidiscus* and the coccolith fraction Sr/Ca ratio or with the coccolithophore productivity proxy (see correlation charts and Pearson's correlation results below – Figure 1 and Table 1).

[Figure]

Figure 1                                                    Figure 2

Figures 1 and 2 showing cross-plots of Calcidiscus leptoporus with coccolithophore productivity proxy (Fig. 1) and the coccolith fraction (CF) Sr/Ca ratio (mmol/mol).

Table 1

| Pearson's Correlation coef. | R | p-value | H0 (5%) |
|---|---|---|---|
| CF Sr/Ca ratio (mmol/mol) vs *C. leptoporus* (%) | -0.14 | 0.07 | *accepted* |
| Coccolithophore productivity proxy vs *C. leptoporus* (%) | -0.09 | 0.21 | *accepted* |

Line 184 – gephyrocapsids's – lower case I believe.
Yes, we thank the reviewer for this comment and have changed all terms accordingly.

Lines 255- bias from other carbonate phases – from Figure 5 it looks like the trend starts before the cut-off used, from more like 30 mmol/mol Mg/Ca. And could you please clarify which "cut off" you are using, whether this is Sr/Ca below 1.8 mmol/mol or higher values of Mg/Ca? If tracing contamination from other carbonates, would it make more sense to use the Mg/Ca values for the cut-off? For example you could cut off more stringently, at _30 mmol/mol Mg/Ca and yet maintain what look like more robust / primary signals of lower Sr/Ca within data that would pass this criteria. You would lose a few more data points in total, but I think this would be a more defensible cut-off point and rationale.
We thank the reviewer for this comment and have used values of Mg/Ca >30 mmol/mol and agree that this way we have a more defensible cut-off point.

Line – 272 – "This interval represents 39% of the whole sampling variation. . .", doesn't make sense to me, please clarify. Do you mean something like the clipped data represents 39% of the dynamic range in Sr/Ca of the full sample set?
We thank the reviewer for this comment and we have substituted that sentence with the following one: "The coccolith fraction Sr/Ca ratio results varied between 1.2 and 2.4 mmol/mol (sampling range of 1.16 mmol/mol) with 85 % of the samples falling between 1.8 and 2.3 mmol/mol."
This means that only 15 % of the samples fall on 70% of the sampling range. And, most of them coincide with the samples with higher bias likelihood from other carbonate phases.

Lines 288-289: is significance level of 80% really enough to be confident that the 6ka peak is real? I'm not really convinced. I don't think this spectral analysis reveals anything and is a distraction for the reader - you have a nice tight coupling to well-resolved climate records (Uk37) and a good age model for making 104-year correlations to N. Atlantic climate records, so I don't think there's a need to try to resolve periodicities independently within this record. I would leave this analysis out.
We agree that we should focus the purpose of this study in long-term processes and not so much on shorter scale changes. Hence, we agree that the spectral analysis might deviate the reader from the most important aspects of our research. Therefore, the spectral analysis was deleted from the revised manuscript.

Lines 355 – 350: are all these paragraphs part of one argument? If so combine.
We thank the reviewer for this comment and have taken it in mind while re-structuring the discussion.

Line 356: Si/Ca fraction. With the preparation methods and uptake by acetic acid digestion, can the authors please clarify which phase / sedimentary component they think the Si is coming from? Is it likely dominated by biogenic Si?
We thank the reviewer for this comment and decided to delete this assumption.

Lines 360 – 365: I think I get what you're trying to say, but this could be expressed more clearly.
We thank the reviewer for this comment and these lines were deleted as a consequence of the previous comment.

Lines 435 – 437 what has MIS12a got to do with MIS 2 and 6? Not clear what your point is here.

We thank the reviewer for this comment and highlight that just for the sake of comparison, we have mentioned other past glacial periods. However, given the re-structuring of the discussion, we expect that the context now is provided in a clearer way in line 450.

**DISCUSSION:**
In general, the discussion feels long and could have more focus. It feels like you are discussing every aspect of the record from a descriptive perspective, rather than focusing on what the data tell you about processes. I've got specific comments below, but I would focus on the broader longer-term trends and behaviour of the coccolithophore productivity and export system during times of distinct oceanographic conditions (i.e. between the stages) rather than the millennial scale lead and lags (dubious as to how robust these are). It would be great to use these different intervals to try frame clearly articulated conceptual models about how and why growth rate is *coupled* or *decoupled* from organic and inorganic carbon fluxes at different stages. Such conceptual models would have the potential to genuinely inform the thinking of biogeochemical modellers by providing clear patterns of change that should be reproducible by numerical biogeochemical models of these systems. But I think you need to clearly formulate these, in words and ideally schematics, in order for them to take notice of your data. This is also where the multi-proxy approach you take is a clear ADVANTAGE, it's not about "mismatches" in the data, it's about using multi-proxy data to represent the responses of different components of the primary production to export system.
We greatly appreciate these very useful comments from the reviewer and believe that the revised manuscript addresses all of these points.

Section 5.2 – looking at the records, I think it's a matter of scale at which they are interrogated. Yes, they are subtly different, but they also preserve some of the same features with reasonable fidelity – for example there are broad trends from MIS 12 to 11, to 10, to 9 that are conserved between the proxies. I would consider taking off the "Mismatch" from your title to this section, to give you space to consider both the agreements and the divergence. This would be more helpful in the communication of the key findings of the study – point out the agreements first and then suggest the mismatches.
We appreciate this comment and we have taken it into account while re-structuring the discussion. We have divided the discussion into two sections: the first discussed the surface ocean processes that have determined past coccolithophore productivity, and a second one where we have followed the reviewer's suggestions and analysed the coccolithophore productivity (surface signal) with the nannofossil and alkenone accumulation rates (that combine both surface and bottom signals) as linked components of the marine carbon cycle.

Following on from this Line 474 – alkenone and n-alkane fluxes mostly reflect conditions of increased export / preservation of organics. Maybe, but I'm not sure that this can, or should, then be decoupled from "rather than coccolithophore growth". . . in line 475. The first order coupling with some features of the coccolith accumulation rates (NAR) and the alkenones, and also your Sr/Ca measure – e.g. the transition from MIS 12 to 11 – would suggest a coupled system change, and this could be the case if increase coccolithophore production was part of the driver of increased general MAR and organic export and accumulation rates? I just wouldn't be so quick to decouple these components.
We appreciate this comment and we have taken it into account while re-structuring the discussion. The second section of the revised manuscript mentions the ballasting effect of calcite (line 475) and we build our argumentation differently from the original version of the manuscript (please read the argumentation given in lines 485-503. In the last lines of this paragraph we now conclude that both our record and the alkenone accumulation rate evidence indeed increased coccolithophore productivity.

Lines 477 – 485: leads and lags of the alkenone versus Sr/Ca records of <1000 ka – is this getting down to the resolution of sampling uncertainty / offset? Were the records based on the same sample set?

We appreciate this reviewer's comment. As previously mentioned, we deleted the spectral analyses and related text from the revised manuscript.

Figure 8 – useful to have the Uk37 temperatures on this figure as well for reference.

This has been changed accordingly; we agree that it offers a better analysis of the records.

Line 498 – "the NAR reveals large amplitude shifts during times of high coccolithophore productivity" – this seems to be missing the point. The Sr/Ca also shows large amplitude shifts within this period, arguably larger than the NAR. The point is that the NAR steps down substantially form MIS 11b to 11a, whereas Sr/Ca remains high (as you say in preceding sentence). This to me is the interesting system change, and there's a reverse trend in alkenone MAR (gentle rise in values into MIS11a). Could this be a seasonality thing between Sr/Ca and NAR? I.e. growing faster (higher Sr/Ca) but for a shorter growth season (less coccoliths?)? Seasonality could (maybe?) also be coupled with more efficient export and preservation of organics (alkenone MARs)? Ah, yes you come to this in Section 5.2.3. But, (see comment below), I think you could condense, simplify and make your interpretation of the key points more clearly in the discussion; including integrating 5.2.3. with these discussions of the data.

Similarly to the previous reviewer's comment on the coupling/decoupling of our coccolithophore productivity and alkenone data, we also analysed the coccolithophore productivity reconstruction and the nannofossil accumulation rate accordingly. We explain this further on the paragraph starting in line 538 of the revised manuscript.

Lines 505 – 509: again the question would be about sampling uncertainty between the records – can you confirm that these leads / lags are meaningful on the sub-millenial scale? I'm just pushing back, because, as a reader, they do not convey a strong argument about process or feature of the data that I would be confident in. This feeds into a general point about the discussion – I think this could be edited down quite considerably, so that your key points are more clearly and forcefully stated (and more easily digested by the reader!). This cross-spectral analysis doesn't add anything for me.

We greatly appreciated the reviewer for his comments and we have deleted the spectral analysis.

**-Reviewer 2**
The paper by Catarina Cavaleiro and collaborators entitled 'Coccolithophore productivity at the western Iberian Margin during the middle Pleistocene (310 – 455 ka) – evidence from coccolith Sr/Ca data' examines the geochemical response (coccolith Sr/Ca elemental data) across the MIS12 – MIS9 time slice offshore Portugal. Based on published coccolithophorid culture finding, the Authors use the abundance of strontium relative to calcite in fossil coccoliths measure by ICP-AES to derive a palaeoproductivity index during the rapid climatic oscillations of the Pleistocene. The region of interest typified by the Portugal Current System was previously documented in terms of changes in the courantology, sea surface temperatures (among other key climate-sensitive data) in a bunch of publications (cited in the paper). The authors used this well-established framework to interpret fluctuations in Sr/Ca ratios and productivity in the sunlit waters. They also discuss their data at the level of the phytoplanktonic ecosystem as they argue that coccolithophorid growth (and productivity) is dictated by macro and micronutrient availability and the competition with diatoms. They mainly focus their biogeochemical discussion on MIS 12 10 showing higher productivity at the beginning of these climate transitions. Playing at different timescales, they ultimately compare their coccolithophorid productivity indices to the available i) alkenone fluxes and ii) nannofossils accumulation rates in published literature and found some coherencies and discrepancies.
I am generally supportive of publication of this work in Climate of the Past. I have, however, a number of comments and questions, which I hope the Authors will find fair and useful to prepare

their revisions.

General comments
- It would be good to state what was measured exactly. 'Coccolith fraction' is not sufficient as the less that 20 micron filtrate may contain many non-coccolith particles. Some photos will be welcome from key samples to illustrate this.

We appreciate the reviewer's comment and we have changed the previous sentence to the following one "To obtain the coccolith fraction (CF) Sr/Ca record, ~250 mg of freeze-dried sample was collected and suspended in 2% ammonia (to avoid carbonate dissolution) and sieved through a 20 μm mesh. This sieving aimed to separate the coccoliths contained in the so-called coccolith fraction (<20 μm) from mostly foraminifera and their fragments, and other larger microfossils or sediment components."

Indeed, the coccolith fraction may contain non-coccolith particles. However, as explained in the methods section, all samples were treated to avoid Sr contamination from non-carbonate particles (assuming that foraminifera and foraminifera fragments were extracted during sieving). Plus, these samples were counted under s light microscope by the researchers Ornella Amore and Eliana Palumbo whose work is already published (Amore et al., 2012; Palumbo et al., 2013) with no mentioning to other non-particles present in the samples (which, by the way, were not sieved prior to the preparation for the observation under the microscope). According to the reviewer's request, we provide the only existing available picture (see below) from site MD03-2699, sample 1898 (referring to the depth in the core – 1898 cm) and with a corresponding age of 485 kyr (not covered in our research). Note that this photo was taken during a master class exercise and consequently it tried to gather as many different coccoliths in picture as possible.

[Figure]

*Figure 1 – SEM picture from site MD03-2699, at 1898 cm with a corresponding age of 485 kyr. The irregular surface background is the filter and several coccoliths from different species can be seen, namely* Calcidiscus leptoporus*,* Helicosphaera carteri *and gephirocapsids.*

- There was this nice paper by Omta et al.) that came out a few years back (On the potential role of marine calcifiers in glacial-interglacial dynamics - doi:10.1002/gbc.20060) in which an elegant model linking ocean alkalinity and the flourishment of coccolithophores at the inception of deglacial periods (with a possible role on the deglaciation). This paper has been omitted in the

present study. I urge the Authors to explore such a control on their productivity data. Even if the periods are not necessarily the same, another useful related paper is that by Duchamp-Alphonse developing the carbonate counter-pump aspects (Enhanced ocean-atmosphere carbon partitioning via the carbonate counter pump during the last deglacial – doi:10.1038/s41467-018- 04625-7). What I am trying to say is that the Authors did a pretty good job in integrating local and regional data but quantitatively understanding pelagic calcification requires a bigger biogeochemical picture. We greatly appreciate this reviewer's comment and suggested further reading. We have now included these references in the paper and have re-structured the discussion. The second section of the revised manuscript mentions carbon cycle components, such as the ballasting effect of calcite (line 475). Consequently, our argumentation now differs from the original version of the manuscript. Please check the major changes done in the manuscript, affecting all discussion chapters.

- Sentence line 283 'We would like to stress that our study focuses on the qualitative characteristics of the coccolithophore paleoproductivity record, rather than quantitatively estimating the productivity of coccolithophores.' is misleading and made me doubt about my understanding of the paper. If the Authors interpret Sr/Ca ratios, they intrinsically develop a quantitative approach pertaining productivity in the surface waters.
We thank the reviewer's comment and we would like to stress that this sentence was included in the manuscript to clarify the reader that the CF Sr/Ca ratio is not an absolute productivity proxy neither does it allow for the calculation of an absolute value of marine production of organic carbon or calcium carbonate by coccolithophores.

- Removing the temperature effect from Sr/Ca data to derive productivity component only. I am still debating with myself to be honest. When I read the paper for the first time, I found that it was a good idea. But the more I think, the more I believe that this is not. Both calcification rates and temperatures (and the control of the latter on the former) synergistically dictate Sr/Ca coccolith ratios. Thus dissecting the proxy may induce an artificial bias. I leave these thoughts to the Authors for their revisions…
- Emerging from the previous point, the heart of the Sr/Ca productivity proxy is poorly approached in this paper. The Authors mix cellular growth rate, primary productivity, and calcification rates. This is only calcification rates that control the substitution of Sr to Ca. Yet, culture data are unable to properly measure calcification rates, as they only document the bulk over the course of the batch experiments (See the Appendices in Stoll et al. 'Climate proxies from Sr/Ca of coccolith calcite: Calibrations from continuous culture of Emiliania huxleyi' published in 2002 in GCA). Thus, the generalisation of the proxy to productivity is far-fetched, as it implicitly means primary productivity in turn leading to the strength of the biological pump. I think that the Authors should clarify this.
We thank the reviewer's comment and we will further clarify these points in the revised manuscript. We further acknowledge the extent work already done in correlating coccolith Sr/Ca ratio with coccolithophore productivity. Stoll and Schrag (2000) initially suggested that the CF Sr/Ca ratios are strongly controlled by coccolithophorid growth and calcification rate. Stoll et al. (2002a; Potential and limitations of proxy) and Stoll et al. (2002b; E. huxleyi cultures), Stoll et al. (2002c; multi species cultures), Stoll et al. (2007a; Arabian and Sargassum seas) and Stoll et al. (2007b; Bay of Bengal) used culture records, sediment traps and sediment samples to confirm the relationship between coccolith Sr/Ca ratios and coccolithophore productivity (coccolithophore growth rate and coccosphere export). Furthermore, Stoll et al. (2002a) and Mejia et al. (2013) clearly stated that the temperature effect on the CF Sr/Ca must be addressed when reconstructing past coccolithophore productivity. Indeed, in our current study the extraction of the temperature effect does not represent a major change of the original curve. However, Cavaleiro et al. (2018) show a final coccolithophore productivity record notably different from the original coccolith fraction Sr/Ca curve due to the large influence of temperature in that area. Our temperature correction in the Iberian margin site reinforces that, contrary to the open mid-latitudinal North Atlantic, the temperature changes in the

Iberian margin do not seem to have affected the coccolith fraction Sr/Ca and consequently coccolithophore productivity. In addition, the possibility to use a proxy that is independent of accumulation rates allows comparison with commonly used "coccolithophore productivity proxies" such as nannofossil accumulation ratios and alkenone export from which coccolithophore productivity, in the ancient photic layer, is commonly inferred from. Finally, the term productivity is thus in this research used as a coccolithophore productivity proxy directly associated with coccolith calcification rate and generally associated with increased cell division and growth of coccolithophores that could lead to increased particulate organic matter and calcium carbonate export.

- The Authors spent considerable effort (and space in the manuscript) to try and find a good match between their coccolithophore productivity and the sedimentation of Point 1 coccolith-derived calcite (NAR) on one hand, and Point 2 coccolith-derived compound-specific organic matter (alkenones) on the other hand. Point 1 For the reasons outlined above, the Sr/Ca has not to scale with the bulk production (-ity) of calcite. This geochemical proxy has to do with intracellular processes why the production of calcite is also related ecologically with the density of cells in seawater and cellular division rates. Point 2 We know that alkenones are not only synthesized by the coccolithophores but also by other non-calcifying haptophytes (incl. naked coccolithophores). Furthermore, the export of calcite and organic matter from the top of the water column down to the seafloor obey to different processes (as their on the seafloor and during sedimentary burial diagenesis).
Therefore, I cannot see why all these parameters should scale. I am not aware of any sedimentary succession in which this is the case. I am happy to be wrong though.
We appreciate this reviewer's comment and believe that the re-structuring of the paper, namely of the second section, these point are addressed. The second section of the revised manuscript mentions carbon cycle components, such as the ballasting effect of calcite (line 475), and suggests conceptual models for the carbon cycle and its different components. Please check the major changes the manuscript suffered in the Discussion.

- I personally disagree with the fact the Si and Fe concentrations relative to Ca are meaningful in such a sedimentary study nor that they reflect the palaeoconcentrations of these elements. Si and Fe are very tricky to measure and it is unlikely that the measurements reflect the composition of coccolith calcite. Even if it was the case, by which means (proxy) the coccolith Si/Ca ratios would reflect the concentration of silicic acid in ambient waters?
We thank the reviewer's comment and we have decided to delete these assumptions from the manuscript.

- The Authors have managed to lose me with the concept of phenology they are trying to introduce. This is a black box concept and this is very misleading or at least not clear at all. Could they elaborate?
We thank the reviewer for this comment and we have decided to delete the coccolithophore phenology change discussion, which was partially linked to the spectral analyses results that have now been omitted from the manuscript. Coccolithophores' phenology (meaning the yearly timing at which they had higher productivity) most certainly changed throughout glacial-interglacial cycles. However, we have not focused on phenology changes in the paper because we believe it would distract the reader from the main findings.

- I found the statistics very poorly treated in the manuscript.
We thank the reviewer's comment and, where possible and according to what we assumed could be improved, we added more information to clarify for the reader the statistical analysis applied. If the reviewer refers to the spectral analysis presented in the previous version of the manuscript, we would like to clarify that, also based on reviewer 1 comment's, the spectral analysis was deleted

from the latest version of the manuscript because it distracted the reader from the most important aspects of this research.

Specific comments

Pg 1 Line 16. Perhaps use Carbonate Counter-Pump instead?
We thank the reviewer for this comment and we have changed the term accordingly.

Pg 1 Line 30. This what?
We thank the reviewer's comment and since the abstract was changed, we believe this issue was addressed.

Pg 1 Line 33. Not clear to me.
We thank the reviewer's comment and this is addressed in the body of the paper to clarify why changes in climate could influence changes in the coccolithophore phenology.

Pg 2 Line 40. Circumvoluted sentence. Consider splitting it.
We agree with the reviewer and since the introduction was also re-structured and re-written, we believe this issue has been addressed.

Pg 2 Line 59. I disagree with this statement (see General points).
We thank the reviewer's comment and following the answer already given to the General points, we only add that we have added references supporting that calcification rate is a function of growth rate in coccolithophores, namely Balch et al. (1996) and Daniels et al. (2018).

Pg 3 Line 68. I wonder whether the changes in size of gephyrocapsid coccoliths could influence the Sr/Ca ratio
We thank the reviewer's comment but believe this is out of the scope of our study since it is not our intention to better understand how coccolith Sr/Ca varies with the size of gephyrocapsa coccoliths.

Pg 4 Line 97. Poorly defined in terms of what?
We thank the reviewer's comment but believe this is out of the scope of our research since it is not our intention to better describe the Portugal current system. We can however substitute "poorly defined due to" by "with" for simplification.

Pg 5 Line 122. Minimum numbers. Do you mean absolute or relative abundances?
We thank the reviewer's comment and we will clarify the text by stating "absolute abundances" instead of "numbers", now in line146.

Pg 6 Line 156. The less than 20 micron fraction contain non coccolith particles. TheAuthors should do a better job in the characterization of the calcite / dolomite particles analysed. This is crucial.
We thank the reviewer's comment and believe we have already replied to this issue above, under the "general points" comments.

Section 3.4. I don't understand what is the relevance of this.
We agree that this spectral analysis might deviate the reader from the most important aspects of our research and for that reason we have deleted it from the revised version.

Section 4.1. belongs to the discussion. Section 4.2 should come first noting that the description if the results is extremality skinny.
We thank the reviewer for this comment but disagree. In our opinion, sections 4.1 and 4.2 should remain in the results section. We agree that some discussion of the results is done here, but the discussion mostly serves to validate our coccolithophore productivity reconstruction in order for it

to be discussed further in the discussion chapters, namely the reasons why coccolithophore productivity changed and why coupling/decoupling with other coccolithophore proxies might have occurred.

Figure 4. Please make the ages more legible.
We thank the reviewer's comment and the figure has been changed accordingly, as well as in figure 3.

Figure 5. What is the significance of the anti-correlation between Mg and Sr?
We thank the reviewer's comment and this information was added to the manuscript. The p-value of the Pearson correlation is 0, therefore the relationship between the coccolith fraction and the coccolithophore productivity and the coccolith fraction Mg/Ca is highly significant, as expected. It is the outliers, or higher values of Mg/Ca that are associated with very low CF Sr/Ca ratios and coccolithophore productivity results, mostly associated to abrupt and cold millennial-scale events.

Figure 6 is unnecessary in my opinion.
We agree that this spectral analysis might deviate the reader from the most important aspects of our research. Therefore, this analysis was left out of the revised manuscript.

Pg 14 Line 303. See my general comment on temperature and productivity on Sr/Ca ratios.
We thank the reviewer's comment and believe we have already replied to this issue above, under the "general points" comments.

Pg 16 Line 333. What do you refer to with 'opportunistic and fast growing species' here?
We appreciate the reviewer's comment and clarify that by "opportunistic and fast growing species" we refer to species with r-selected strategies, meaning that they react more rapidly with higher growth rates than k-selected species that are adapted to low nutrients and have lower growth rates (an example of a paper focusing on coccolithophore dynamic in the Iberian margin also refer these more biological terms: Guerreiro, C., Oliveira, A., De Stigter, H., Cachão, M., Sá, C., Borges, C., Cros, L., Santos, A., Fortuño, J. M. and Rodrigues, A.: Late winter coccolithophore bloom off central Portugal in response to river discharge and upwelling, Cont. Shelf Res., 59, 65–83, doi:10.1016/j.csr.2013.04.016, 2013.)

Pg 16 Line 355. Methodologically unjustified even using 'weak' acid.
We thank the reviewer for this comment and we have decided to delete this assumption that higher coccolith fraction Si/Ca and Fe/Ca could evidence higher competition with diatoms.

Pg 17 Line 361. Sentence not clear and too long.
We thank the reviewer for this comment and we have decided to delete this assumption that higher coccolith fraction Si/Ca and Fe/Ca could evidence higher competition with diatoms.

Pg 17 Line 371. Decrease of the SST.
We thank the reviewer's comment and it has been changed accordingly.

Pg 18 Line 421. I am not following the logic here. Are the Authors trying to say that the ice coverage reached the studied area?
We thank the reviewer's comment and clarify that we have not stated or suggested that ice coverage reached the Iberian margin. We are stating research that has found evidences of the presence of melting icebergs in the eastern mid-latitudinal North Atlantic and in western Iberian margin during rapid millennial-scale events, namely Alonso-Garcia et al. (2011), Hodell et al. (2008), McManus et al. (1999), Rodrigues et al. (2011, 2017). We believe that it is out of the scope of this study to infer sea-ice coverage in the Iberian margin during the analyzed interval.

Pg 19 Line 449. I don't understand the point that the Authors are trying to make here.
We thank the reviewer for this comment and we have decided to delete this assumption that higher coccolith fraction Si/Ca and Fe/Ca could evidence higher competition with diatoms.

Pg 19 Line 454. Visual comparison of what?
We thank the reviewer's comment and further clarify that we refer that the records of coccolithophore productivity, nannofossil accumulation rate and alkenone flux were compared visually.

Pg 19 Line 457. An illustration of the poor statistical approach here. ..
We thank the reviewer for this comment but given the re-structuring of the discussion we no longer present such data.

Pg 21 Lines 477- 492 and figure 9 are not necessary.
We agree that this spectral analysis might deviate the reader from the most important aspects of our research. Therefore, this analysis was left out of the revised manuscript.

**Coccolithophore productivity at the western Iberian Margin during the middle Pleistocene (310 – 455 ka) – evidence from coccolith Sr/Ca data**

Catarina Cavaleiro[1, 2, 3], Antje H. L. Voelker[2, 3], Heather Stoll[4*], Karl-Heinz Baumann[5] and Michal Kucera[1]

[1]University of Bremen, MARUM - Center for Marine and Environmental Sciences, Leobener Straße 8, 28359 Bremen, Germany.
[2]IPMA – Instituto Português do Mar e da Atmosfera, Divisão de Geologia Marinha e Georecursos Marinhos, Avenida Doutor Alfredo Magalhães Ramalho 6, 1495-165 Algés, Portugal.
[3]CCMAR, Centro de Ciências do Mar, Universidade do Algarve, Campus de Gambelas, 8005-139 Faro, Portugal.
[4]Geology Department, University of Oviedo, C/. Jesús Arias de Velasco s/n, 33005, Oviedo, Spain. *now at: Department of Earth Sciences, ETH Zürich, Sonneggstrasse 5, 8092 Zürich, Switzerland
[5]University of Bremen, Geosciences Department, Klagenfurter Straße 2-4, 28359 Bremen, Germany.

*Correspondence to*: Catarina Cavaleiro (cdcavaleiro@gmail.com)

**Abstract.** Coccolithophores contribute significantly to marine primary productivity and play a unique role in ocean biogeochemistry by using carbon for photosynthesis (soft tissue pump) and for calcification (carbonate counter pump). Despite the importance of including coccolithophores in Earth System Models to allow better predictions of the climate system's responses to planetary change, the reconstruction of coccolithophore productivity mostly relied on proxies dependent on accumulation and sedimentation rates, and preservation conditions. In this study we used an independent proxy, based on the coccolith fraction (CF) Sr/Ca ratio, to reconstruct coccolithophore productivity. We studied the marine sediment core MD03-2699 from the western Iberian margin (IbM), concentrating on glacial/interglacial cycles of Marine Isotopic Stage (MIS) 12 to MIS 9. We found that IbM coccolithophore productivity was controlled by changes in the oceanographic conditions, such as in sea surface temperature (SST) and nutrient availability, and by competition with other phytoplankton groups. Long-term coccolithophore productivity was primarily affected by variations in the dominant surface water mass. Polar and subpolar surface waters during glacial substages were associated with decreased coccolithophore productivity, with strongest productivity minima concomitant with Heinrich-type events (HtE). Subtropical, nutrient-poorer waters, increased terrigenous input and moderate to strong upwelling during the deglaciation and early MIS11 are hypothesized to have attributed a competitive advantage to diatoms in detriment of coccolithophores, resulting in intermediate coccolithophore productivity levels. During the progression towards full glacial conditions increasing presence of nutrient-richer waters, related to growing influence of transitional surface waters and/or intensified upwelling, probably stimulated coccolithophore productivity to maxima following the rapid depletion of silica by diatoms. We present conceptual models of the carbon and carbonate cycle components for the IbM in different time slices that might serve as a basis for further investigation and modeling experiments.

**1 Introduction**

Coccolithophores play a unique role in ocean biogeochemistry using carbon for both photosynthesis and calcification (e.g. Rost and Riebesell, 2004; Westbroek et al., 1993) and contributing up to 60 % to the total calcium carbonate in the ocean (Flores and Sierro, 2007). During the Mid Brunhes interval of Marine Isotope Stage (MIS) 14 to MIS 9, when the assemblages were vastly dominated by gephyrocapsids (Baumann and Freitag, 2004; Saavedra-Pellitero et al., 2017), coccolithophores provided >80 % of the total oceanic calcium carbonate. And, several studies hypothesized a relevant role of coccolithophores on glacial-interglacial dynamics (Duchamp-Alphonse et al., 2018; McClelland et al., 2016; Omta et al., 2013; Rickaby et al., 2007; Saavedra-Pellitero et al., 2017) but there are uncertainties arising from the complex interactions between coccolithophore productivity and growth rates, nutrient dynamics and competition with non-calcifiers, seasonality and export of both carbonate and organic carbon and the final accumulation/burial rates (Balch, 2018; Duchamp-Alphonse et al., 2018; McClelland et al., 2016; Omta et al., 2013; Rickaby et al., 2007; Ridgwell and Zeebe, 2005; Saavedra-Pellitero et al., 2017). These uncertainties make the inclusion of biogeochemical processes related to coccolithophores in Earth System models challenging. In order to provide useful information to modellers, it is thus fundamental to test the dynamic response of the various components of the system over different climatic scenarios through the use of multiproxy studies involving the reconstruction of coccolithophores growth rates and export of both their organic and inorganic compounds, namely alkenones and nannofossil accumulation rate (AlkAR and NAR, respectively).

Since coccolithophores are sensitive to rapid fluctuations in temperature, salinity, nutrients, and turbidity of surface waters (Baumann et al., 2005; McIntyre and Bé, 1967), their calcareous remains, the coccoliths, retrieved from deep-sea sediments have been used extensively to reconstruct paleoenvironmental conditions (Amore et al., 2012; Baumann et al., 2005; Beaufort et al., 2001; Flores et al., 1997; Maiorano et al., 2015; Marino et al., 2014; McIntyre and Molfino, 1996; Saavedra-Pellitero et al., 2017). However, coccolithophore paleoproductivity reconstruction has been tentative and mostly relied on proxies dependent not only on the extent of the supply but also on dilution by mineral matter, changes in sedimentation or accumulation rates, as well as preservation conditions (Rullkötter, 2006). Beaufort et al. (1997) proposed a proxy to quantitatively reconstruct coccolithophore productivity, but its applicability is unfortunately limited to latitudes between 30°N and 30°S (Hernández-Almeida et al., 2019). A widely used alternative proxy is the coccolith fraction Sr/Ca (CF Sr/Ca) ratio that is independent of accumulation rate (Cavaleiro et al., 2018; Mejía et al., 2014; Saavedra-Pellitero et al., 2017; Tangunan et al., 2017). Coccolithophores construct coccoliths internally within their cell and several studies show the direct and proportional relationship between the Sr/Ca ratio of the coccolith and the coccolith calcification rate. Calcification rate is a function of growth rate (e.g. Daniels et al., 2018) and therefore of coccolithophore productivity (Balch et al., 1996; Rickaby et al., 2007; Stoll and Schrag, 2000). The faster coccolithophores grow, the faster they calcify and more Sr is incorporated into the calcite lattice of their coccoliths (Stoll et al., 2002b, 2002a; Stoll and Schrag, 2000). With temperature and assemblage effects considered (see Material and methods section), the SST corrected CF Sr/Ca curve (or residual curve)

Catarina Cavaleiro 3/7/2020 16:57

Catarina Cavaleiro 3/7/2020 16:57

Catarina Cavaleiro 3/7/2020 16:57

Catarina Cavaleiro 3/7/2020 16:57

Catarina Cavaleiro 3/7/2020 16:57

Catarina Cavaleiro 3/7/2020 16:57

Catarina Cavaleiro 3/7/2020 16:57

Catarina Cavaleiro 3/7/2020 16:57

Catarina Cavaleiro 3/7/2020 16:57

Catarina Cavaleiro 3/7/2020 16:57

Catarina Cavaleiro 3/7/2020 16:57

Catarina Cavaleiro 3/7/2020 16:57

Catarina Cavaleiro 3/7/2020 16:57

Catarina Cavaleiro 3/7/2020 16:57

Catarina Cavaleiro 3/7/2020 16:57

Catarina Cavaleiro 3/7/2020 16:57
**Moved (insertion) [1]**

Catarina Cavaleiro 3/7/2020 16:57
**Moved (insertion) [2]**

[revised manuscript text omitted]

the

| Page 1: [1] Deleted | Catarina Cavaleiro | 03/07/2020 16:57 |

the

| Page 1: [1] Deleted | Catarina Cavaleiro | 03/07/2020 16:57 |

the

| Page 1: [1] Deleted | Catarina Cavaleiro | 03/07/2020 16:57 |

the

| Page 1: [1] Deleted | Catarina Cavaleiro | 03/07/2020 16:57 |

the

| Page 1: [1] Deleted | Catarina Cavaleiro | 03/07/2020 16:57 |

the

| Page 1: [1] Deleted | Catarina Cavaleiro | 03/07/2020 16:57 |

the

| Page 1: [1] Deleted | Catarina Cavaleiro | 03/07/2020 16:57 |

the

| Page 1: [1] Deleted | Catarina Cavaleiro | 03/07/2020 16:57 |

the

| Page 1: [1] Deleted | Catarina Cavaleiro | 03/07/2020 16:57 |

the

| Page 1: [1] Deleted | Catarina Cavaleiro | 03/07/2020 16:57 |

the

| Page 1: [1] Deleted | Catarina Cavaleiro | 03/07/2020 16:57 |

the

| Page 1: [1] Deleted | Catarina Cavaleiro | 03/07/2020 16:57 |

the

| Page 1: [1] Deleted | Catarina Cavaleiro | 03/07/2020 16:57 |

the

| Page 1: [1] Deleted | Catarina Cavaleiro | 03/07/2020 16:57 |

the

| Page 1: [1] Deleted | Catarina Cavaleiro | 03/07/2020 16:57 |

the

| Page 1: [1] Deleted | Catarina Cavaleiro | 03/07/2020 16:57 |

the

| Page 1: [1] Deleted | Catarina Cavaleiro | 03/07/2020 16:57 |

the

| Page 1: [1] Deleted | Catarina Cavaleiro | 03/07/2020 16:57 |

the

| Page 1: [1] Deleted | Catarina Cavaleiro | 03/07/2020 16:57 |

the

| Page 1: [1] Deleted | Catarina Cavaleiro | 03/07/2020 16:57 |

the

| Page 1: [1] Deleted | Catarina Cavaleiro | 03/07/2020 16:57 |

the

| Page 1: [1] Deleted | Catarina Cavaleiro | 03/07/2020 16:57 |

the

[revised manuscript text omitted]

defined

| Page 13: [8] Deleted | Catarina Cavaleiro | 03/07/2020 16:57 |

defined

| Page 13: [8] Deleted | Catarina Cavaleiro | 03/07/2020 16:57 |

defined

| **Page 13: [8] Deleted** | **Catarina Cavaleiro** | **03/07/2020 16:57** |

defined

| **Page 14: [9] Deleted** | **Catarina Cavaleiro** | **03/07/2020 16:57** |

and horizontal

| **Page 14: [9] Deleted** | **Catarina Cavaleiro** | **03/07/2020 16:57** |

and horizontal

| **Page 14: [9] Deleted** | **Catarina Cavaleiro** | **03/07/2020 16:57** |

and horizontal

| **Page 14: [9] Deleted** | **Catarina Cavaleiro** | **03/07/2020 16:57** |

and horizontal

| **Page 14: [9] Deleted** | **Catarina Cavaleiro** | **03/07/2020 16:57** |

and horizontal

| **Page 14: [9] Deleted** | **Catarina Cavaleiro** | **03/07/2020 16:57** |

and horizontal

| **Page 14: [9] Deleted** | **Catarina Cavaleiro** | **03/07/2020 16:57** |

and horizontal

| **Page 14: [9] Deleted** | **Catarina Cavaleiro** | **03/07/2020 16:57** |

and horizontal

| **Page 14: [10] Deleted** | **Catarina Cavaleiro** | **03/07/2020 16:57** |

ratios

| **Page 14: [10] Deleted** | **Catarina Cavaleiro** | **03/07/2020 16:57** |

ratios

| **Page 14: [10] Deleted** | **Catarina Cavaleiro** | **03/07/2020 16:57** |

ratios

| **Page 14: [10] Deleted** | **Catarina Cavaleiro** | **03/07/2020 16:57** |

ratios

| **Page 14: [10] Deleted** | **Catarina Cavaleiro** | **03/07/2020 16:57** |

ratios

| **Page 14: [10] Deleted** | **Catarina Cavaleiro** | **03/07/2020 16:57** |

ratios

| Page 14: [10] Deleted | Catarina Cavaleiro | 03/07/2020 16:57 |
|---|---|---|

ratios

| Page 14: [11] Deleted | Catarina Cavaleiro | 03/07/2020 16:57 |
|---|---|---|

coccolithophore productivity

| Page 14: [12] Deleted | Catarina Cavaleiro | 03/07/2020 16:57 |
|---|---|---|

12a

| Page 14: [13] Deleted | Catarina Cavaleiro | 03/07/2020 16:57 |
|---|---|---|

Coccolithophore productivity

**Page 14: [13] Deleted**          **Catarina Cavaleiro**          **03/07/2020 16:57**

Coccolithophore productivity

**Page 16: [14] Deleted**          **Catarina Cavaleiro**          **03/07/2020 16:57**

We therefore firstly evaluated the relationship between coccolithophore productivity and SST changes. The relationship between SST and coccolithophore productivity is characterised by a weak negative correlation ($r=-0.4$, $p<0.01$). However, the cross-plot seems to show an inflexion around 15.5 ºC (Fig. 7) and, by separating the different sets of samples under the different temperature ranges (below and above 15.5 ºC), we obtained very different correlation results. Increasing temperatures until 15.5 ºC seem to have a positive effect on coccolithophore productivity (though with a low positive correlation; $r=0.27$, $p<0.05$), whereas increasing temperatures above 15.5 ºC seem to have a negative effect on coccolithophore productivity (with a medium negative correlation; $r=-0.54$, $p<0.01$). Though it is accepted that increased temperature generally enhances coccolithophores growth (e.g. Sett et al., 2014), our results demonstrate that the relationship between coccolithophores productivity and SST is not straightforward and should be explored further in the future and that other factors, besides temperature, must have affected coccolithophore productivity in the western IbM. In an upwelling region like the IbM a potential factor could be nutrient concentrations because the upwelled waters in general have temperatures between 13 and 16 °C (e.g., Fiúza et al., 1998).

[Figure]

**Correlation Coefficients - Coccolithophore productivity (CF Sr/Ca residual) vs SST (ºC)**

|  | *Sr/Ca residual vs. SST (ºC)* | *Sr/Ca residual vs. SST (ºC)* | *Sr/Ca residual vs. SST (ºC)* |
|---|---|---|---|
| **R** | **-0.4** | **0.27** | **-0.54** |
| *R Standard Error* | 5.17E-3 | 0.01 | 7.3E-3 |
| *t* | -5.52 | 2.22 | -6.32 |
| *p-value* | 1.29E-7 | 0.03 | 7.8E-9 |
| *H0 (5%)* | rejected | rejected | rejected |
| *No# of valid cases* | 165 (all valid samples) | 66 (samples SST < 16 ºC) | 99 (samples > 16 ºC) |

**Figure 7** Coccolithophore paleoproductivity and SST Pearson correlation results: all valid samples (green), samples below 15.5 ºC (blue) and samples above 15.5 ºC (orange). Person correlation computed with StatPlus:mac, AnalystSoft Inc. - statistical analysis program for macOS®. Version v7. See http://www.analystsoft.com/en/

**5.1 Causes for coccolithophore productivity change**

5.1.1

| Page 17: [15] Deleted | Catarina Cavaleiro | 03/07/2020 16:57 |

**terminations and interglacial periods**

**The abrupt SST rise during the**
* * *
| Page 17: [16] Deleted | Catarina Cavaleiro | 03/07/2020 16:57 |

Accordingly, during MIS 9e and MIS 11c, the abundance of warm coccolith taxa increased (Amore et al., 2012; Maiorano et al., 2015; Marino et al., 2014; Palumbo et al., 2013a) as well as the abundance of tropical planktonic foraminifera species in cores MD01-2443 (de Abreu et al., 2005) and MD03-2699 (presence of *Globorotalia menardii* and *Sphaeroidinella dehiscens;* Voelker et al., 2010). These evidences point to the prevailing presence of subtropical oligotrophic waters during the later phase of the deglaciation and across interglacial MIS 11c. As a consequence of the limited nutrient supply, coccolithophore productivity, especially the opportunistic and fast-growing species, could have been limited to intermediate levels during MIS 11c.
* * *
| Page 17: [17] Deleted | Catarina Cavaleiro | 03/07/2020 16:57 |

Besides changes in the nutrient availability

| Page 17: [18] Deleted | Catarina Cavaleiro | 03/07/2020 16:57 |

coccolithophore productivity.

The

| Page 18: [19] Deleted | Catarina Cavaleiro | 03/07/2020 16:57 |

iron

| Page 18: [19] Deleted | Catarina Cavaleiro | 03/07/2020 16:57 |

iron

| Page 18: [19] Deleted | Catarina Cavaleiro | 03/07/2020 16:57 |

iron

| Page 18: [19] Deleted | Catarina Cavaleiro | 03/07/2020 16:57 |

iron

| Page 18: [19] Deleted | Catarina Cavaleiro | 03/07/2020 16:57 |

iron

| Page 18: [19] Deleted | Catarina Cavaleiro | 03/07/2020 16:57 |

iron

| Page 18: [19] Deleted | Catarina Cavaleiro | 03/07/2020 16:57 |

iron

| Page 18: [19] Deleted | Catarina Cavaleiro | 03/07/2020 16:57 |

iron

| Page 18: [19] Deleted | Catarina Cavaleiro | 03/07/2020 16:57 |

iron

| Page 18: [19] Deleted | Catarina Cavaleiro | 03/07/2020 16:57 |
| --- | --- | --- |

iron

| Page 18: [19] Deleted | Catarina Cavaleiro | 03/07/2020 16:57 |
| --- | --- | --- |

iron

| Page 18: [20] Deleted | Catarina Cavaleiro | 03/07/2020 16:57 |
| --- | --- | --- |

Also, based on evidence from the late Pleistocene (e.g. Margari et al., 2014), the transition from MIS 12 to MIS 11 is likely also marked by a transition from very cold winters and windy conditions in the western and southern Iberian margin to warmer conditions with increased winter precipitation (Desprat et al., 2009; Tzedakis et al., 2009). Higher humidity and precipitation would increase continental weathering and transport of silica and iron to coastal areas. And indeed the coccolith fraction, in which the Sr/Ca ratio was measured, also shows increased silica and iron contents when compared to calcium (CF Si/Ca and CF Fe/Ca; Fig. 4), especially during deglaciation and the beginning of MIS 11c. Also, CF Si/Ca and CF are intricately connected (r=0.95, p<0.01), and are both negatively correlated with coccolithophore productivity (r=-0.44, p<0.01 and r= -0.49, p<0.01, respectively). This could indicate that even if iron, likely brought by dust originated in the Sahara desert, might have played an important fertilizing effect for the overall phytoplankton community offshore the IbM (Blain et al., 2004) in the past, diatoms most likely outcompeted coccolithophores, limiting their productivity due to increased competition for nutrients, when silica was in sufficient amount not to limit diatom blooms. All of these evidences suggest that irrespective of a moderate to strong upwelling regime, which increased the overall nutrients availability and phytoplanktonic productivity, only sufficiently higher the amounts of silica and iron introduced by increased wind and aridity or by increased precipitation, would lead to diatom blooms and coccolithophores to be outcompeted, most likely explaining the intermediate level of coccolithophore productivity.

**5.1.2 High coccolithophore productivity levels associated with the transition from interglacial to glacial periods**

Coccolithophore productivity increases steadily from mid MIS 11c until the end of MIS 11b and remained generally high from late MIS 11c to MIS 10c. Although the increasing trend in coccolithophore productivity coincides with a decreasing trend in SST from 403 to 390 kyr (which returned a high negative correlation, r=-0.88, p<0.01), the high productivity interval, from 398 kyr to 354 kyr, seems disconnected from SST influence (r=0.16, p=0.2).

The cooling of the SST during the transition from interglacial substages to glacial substages is related to the build-up of ice sheets on the continents and the subsequent changes in atmospheric and oceanic circulation, which subsequently resulted in a strength decline of the AMOC (e.g., McManus et al., 1999; Voelker et al., 2010), also evidenced by the decreasing trend in bottom water ventilation (e.g., Martrat et al., 2007). During interglacial substages, such as MIS 11c and MIS 9e, the wind stress associated to the upwelling events is hypothesized to be lower than during glacial substages (e.g.

**Page 18: [21] Deleted**        **Catarina Cavaleiro**        **03/07/2020 16:57**

Whereas, during the transitions from interglacial to glacial substages, the narrower latitudinal temperature gradient caused by the expansion of the northern continental ice sheets and the associated location shifts in the North Atlantic frontal system, is hypothesized to have increased the northerly winds and lead to more intense upwelling or wind related mixing of the upper water column. As a consequence, higher turbulence would replenish the ocean surface with nutrients.

Because of the differences in the planktonic oxygen isotope records between cores MD03-2699 and MD01-2446 (more offshore) Voelker et al. (2010) suggested that the high variability in the closer to shore core MD03-2699 could reflect variations in upwelling of deeper waters into the thermocline. Indeed, Fiúza (1984) had already proposed that variations in wind stress could lead to the upwelling of different water masses.

**Page 18: [22] Deleted**        **Catarina Cavaleiro**        **03/07/2020 16:57**

Enhanced northerly wind stress could have intensified the upwelling, both in strength (upwelling deeper and nutrient-richer waters) and in distance to shore (reaching further offshore than today). Increased nutrient availability would thus support the whole phytoplankton community and decrease diatoms' and coccolithophores' competition for nutrients, especially for those coccolithophores species capable of rapid growth in a nutrient-rich environment, such as the *Gephyrocapsa caribbeanica* is thought to have been, given its cosmopolitan distribution and dominance in the sediments (Baumann and Freitag, 2004; Bollmann et al., 1998). Note that this would only be possible given that bioavailable silica and iron were not sufficiently high to allow diatoms to outcompete coccolithophores, as seen before, and as the CF Si/Ca seem to show, with quite low and constant levels during the coccolithophore productivity maxima, only interrupted by a period of increased CF Si/Ca and abrupt decrease in coccolithophore productivity, from ~375 to 365 ka (Fig. 4). The coccolithophore productivity maxima during the transition from interglacial MIS 11c to glacial MIS 10a conditions indicate the fastest growth and calcification rates of the record, only comparable to mid-MIS 12 levels, in the beginning of our record, and preceding full glacial conditions. This supports the idea that the coccolithophore community was able to better perform under these transitional conditions, despite colder SST, and under windier and more turbulent settings and more intense upwelling.

**Page 18: [23] Deleted**        **Catarina Cavaleiro**        **03/07/2020 16:57**

Coccolithophore productivity

**Page 18: [23] Deleted**        **Catarina Cavaleiro**        **03/07/2020 16:57**

Coccolithophore productivity

**Page 18: [23] Deleted**        **Catarina Cavaleiro**        **03/07/2020 16:57**

Coccolithophore productivity

**Page 18: [23] Deleted**        **Catarina Cavaleiro**        **03/07/2020 16:57**

Coccolithophore productivity

| Page 18: [23] Deleted | Catarina Cavaleiro | 03/07/2020 16:57 |
|---|---|---|

Coccolithophore productivity

| Page 18: [23] Deleted | Catarina Cavaleiro | 03/07/2020 16:57 |
|---|---|---|

Coccolithophore productivity

| Page 18: [23] Deleted | Catarina Cavaleiro | 03/07/2020 16:57 |
|---|---|---|

Coccolithophore productivity

| Page 18: [23] Deleted | Catarina Cavaleiro | 03/07/2020 16:57 |
|---|---|---|

Coccolithophore productivity

| Page 18: [23] Deleted | Catarina Cavaleiro | 03/07/2020 16:57 |
|---|---|---|

Coccolithophore productivity

| Page 19: [24] Deleted | Catarina Cavaleiro | 03/07/2020 16:57 |
|---|---|---|

AMOC

| Page 19: [24] Deleted | Catarina Cavaleiro | 03/07/2020 16:57 |
|---|---|---|

AMOC

| Page 19: [24] Deleted | Catarina Cavaleiro | 03/07/2020 16:57 |
|---|---|---|

AMOC

| Page 19: [24] Deleted | Catarina Cavaleiro | 03/07/2020 16:57 |
|---|---|---|

AMOC

| Page 19: [24] Deleted | Catarina Cavaleiro | 03/07/2020 16:57 |
|---|---|---|

AMOC

| Page 19: [25] Deleted | Catarina Cavaleiro | 03/07/2020 16:57 |
|---|---|---|

coccolithophore productivity

| Page 19: [25] Deleted | Catarina Cavaleiro | 03/07/2020 16:57 |
|---|---|---|

coccolithophore productivity

| Page 19: [25] Deleted | Catarina Cavaleiro | 03/07/2020 16:57 |
|---|---|---|

coccolithophore productivity

| Page 19: [25] Deleted | Catarina Cavaleiro | 03/07/2020 16:57 |
|---|---|---|

coccolithophore productivity

| Page 19: [25] Deleted | Catarina Cavaleiro | 03/07/2020 16:57 |
|---|---|---|

coccolithophore productivity

**Page 19: [26] Deleted**       **Catarina Cavaleiro**       **03/07/2020 16:57**

coccolithophore productivity

**Page 19: [27] Deleted**       **Catarina Cavaleiro**       **03/07/2020 16:57**

[revised manuscript text omitted]

---

## Author Response (AR3)

Lisbon, 25th September 2020

Dear Editor Luc Beaufort,

We are submitting the revised version of the manuscript " Coccolithophore productivity at the western Iberian Margin during the middle Pleistocene (310 – 455 ka) – evidence from coccolith Sr/Ca data " [cp-2019-131] in which we address the three minor revisions asked from you, all changed accordingly.

I also added the doi link, under **Data availability**, of the Pangea webpage from where the data will be available and freely downloadable.

Once more, thank you very much for your understanding throughout the revision process and under such a challenging pandemic situation.

Best regards, on behalf of all co-authors,
Catarina Cavaleiro